# Pointwise Information Measures as Confidence Estimators in Deep Neural Networks: A Comparative Study

**Shelvia Wongso** [1]  **Rohan Ghosh** [1]  **Mehul Motani** [1 2]

## Abstract

Estimating the confidence of deep neural network predictions is crucial for safe deployment in high-stakes applications. While softmax probabilities are commonly used, they are often poorly calibrated, and existing calibration methods have been shown to be detrimental to failure prediction. In this paper, we propose using information-theoretic measures to estimate prediction confidence in a post-hoc manner, without modifying network architecture or training. Specifically, we compare three pointwise information (PI) measures: pointwise mutual information (PMI), pointwise $\mathcal{V}$-information (PVI), and the recently proposed pointwise sliced mutual information (PSI). These measures are theoretically grounded in their relevance to predictive uncertainty, with properties such as invariance, convergence rates, and sensitivity to geometric attributes like margin and intrinsic dimensionality. Through extensive experiments on benchmark computer vision models and datasets, we find that PVI consistently outperforms PMI, PSI and existing post-hoc baselines in failure prediction across metrics. For confidence calibration, PVI matches the performance of temperature-scaled softmax, which is already regarded as a highly effective baseline. This indicates that PVI achieves superior failure prediction without compromising its calibration performance. This aligns with our theoretical insights, which suggest that PVI offers the most balanced trade-offs.

[1]Department of Electrical and Computer Engineering, National University of Singapore, Singapore. [2]N.1 Institute for Health, Institute for Digital Medicine, Institute of Data Science, National University of Singapore, Singapore. Correspondence to: Shevia Wongso <shelvia.w@u.nus.edu>, Rohan Ghosh <rghosh92@gmail.com>, Mehul Motani <motani@nus.edu.sg>.

*Proceedings of the $42^{nd}$ International Conference on Machine Learning*, Vancouver, Canada. PMLR 267, 2025. Copyright 2025 by the author(s).

## 1. Introduction

As deep neural networks (DNNs) are increasingly being deployed in high-stakes areas like healthcare and autonomous driving, the focus has shifted from achieving good accuracy to also ensuring trustworthiness for safe deployment (Kaur et al., 2023). An important aspect of a trustworthy model is its ability to accurately quantify prediction confidence, enabling users to judge when to trust its outputs (Jiang et al., 2018). The standard way to estimate confidence is via softmax outputs, but they can be overconfident on misclassified samples (Guo et al., 2017). Many existing approaches that address this issue involve modifying the network architecture (Corbière et al., 2019) or training procedure (Gal & Ghahramani, 2016), which may not always be feasible in practice. Meanwhile, popular confidence calibration methods have been shown to be useless or harmful for failure prediction tasks (Zhu et al., 2022). This study addresses both failure prediction and confidence calibration by analyzing information-theoretic measures to estimate the confidence of predictions from trained networks in a post-hoc manner, without altering their architecture or training process.

Mutual Information (MI) is the conventional information measure used to capture statistical dependence between two random variables (Cover & Thomas, 2001). However, accurately estimating MI in high-dimensional spaces, typically encountered in the context of DNNs, is challenging due to an exponentially large sample complexity (Battiti, 1994). In recent years, there have been proposals for alternative measures of informativeness that scale well with dimensions. The first is the $\mathcal{V}$-*information* ($\mathcal{V}$I) which measures the amount of usable information under computational constraints (Xu et al., 2020). The second is *sliced mutual information* (SMI) which is the average of the MI between one-dimensional projections of the random variables (Goldfeld & Greenewald, 2021). Unlike MI, both $\mathcal{V}$I and SMI can be estimated reliably from data, even in high dimensions.

To apply the three information-theoretic measures (MI, $\mathcal{V}$I, and SMI) for confidence estimation, we use their pointwise variants: pointwise MI (PMI), pointwise $\mathcal{V}$I (PVI), and pointwise SMI (PSI). Specifically, we use these pointwise information (PI) measures to quantify the degree of relevance between feature representation and predicted output

of a model for each individual sample. We analyze their theoretical properties, including invariance, convergence rates, dependence on margin and intrinsic dimensionality, and discuss their relevance to confidence estimation problem. Empirically, we compare their effectiveness in estimating confidence scores on benchmark image classification tasks for failure prediction and calibration. For a review of related work on confidence estimation and the three PI measures, please refer to Appendix A.1.

**Motivation.** We provide four factors that motivate the use of PI measures for confidence estimation:

1. **Recent Applications of PI Measures:** PI measures have recently found application in diverse domains of DNNs, showcasing their versatility and effectiveness. For instance, the significant work by (Ethayarajh et al., 2022) showcases the applicability of PVI to the problem of dataset difficulty, which relates to confidence as networks are naturally less confident in their predictions when the datasets are harder. While PI measures are more commonly applied in natural language, we focus on their potential in computer vision, an area still relatively under-explored from an information-theoretic perspective. More closely aligned with our work is the study by (Wongso et al., 2023b) which proposed PSI for confidence estimation and explainability. We extend their research by comparing the performance of PSI with PMI and PVI, providing deeper theoretical insights, and conducting extensive evaluations.

2. **Theoretical Foundations:** Despite the increasing applications of the PI measures, there has been a notable lack of research devoted to exploring their theoretical properties. To the best of our knowledge, only PMI has been theoretically studied (Fano & Hawkins, 1961b), though largely in a more general context. In this work, we derive and compare theoretical properties of PMI, PVI, and PSI that are relevant to this setting, including invariance, convergence rates, margin and intrinsic dimensionality. We find that these measures exhibit some properties that can be relevant in the context of confidence estimation.

3. **Information Theoretic Connection:** Another interpretation of the PI measures comes from the notion of information gain in information theory. In its standard (aggregate) form, information gain measures how much knowing one variable $X$ reduces uncertainty about another variable $Y$, i.e., $H(Y) - H(Y|X)$. PI measures can be seen as a pointwise version of this information gain, computing the logarithm of the probability density ratio $p(\hat{y}|x)/p(\hat{y})$ (and its variant), where $x$ is the feature and $\hat{y}$ is the predicted output. In contrast, a typical neural network output approximates the conditional probabilities of each class $p(\hat{y}|x)$ from logits, which themselves lack direct probabilistic meaning. PI measures, being grounded in information theory, offer a more principled alternative to the softmax output computed from logits for confidence estimation. They quantify the relative increase in confidence for the predicted class compared to its prior occurrence probability. By focusing on this relative change, PI measures can help mitigate biases in $p(\hat{y}|x)$, which can arise due to underrepresentation of certain classes in the data.

4. **Relationship to Probabilistic Causation:** We find that PI measures can be also interpreted via the lens of probabilistic causation. This perspective on causality, as outlined by (Hitchcock, 1997), argues that $X$ causes $Y$ if $P(Y|X) > P(Y)$. In the context of confidence estimation, the goal is to quantify how strongly a feature $x$ supports the predicted class $\hat{y}$: the model should naturally assign higher confidence to predictions where $x$ has a strong influence on $y$. We argue that this problem can be mathematically formulated by measuring the quantity $p(\hat{y}|x)/p(\hat{y})$ which indicates the degree to which a certain feature $x$ influences the decision made for a single instance. This directly connects to PI measures which compute the logarithm of the probability density ratio.

**Contributions.** The specific contributions of this paper are as follows:

1. **Information-Theory Grounded Measures**. We propose to use information-theoretic pointwise measures (PMI, PSI and PVI) for confidence estimation. These measures capture different aspects of the relationship between model predictions and true labels, providing alternative perspectives beyond conventional softmax-based confidence scores.

2. **Theoretical Analysis of PI Measures.** We perform an in-depth study of the theoretical properties of the three PI measures, including invariance properties, convergence rates, and geometric characteristics such as margin and intrinsic dimensionality (Section 3 and Appendix B.3). We explain how each of these properties relates to confidence in neural networks and highlight several noteworthy caveats. In particular, we observe that margin sensitivity is a less critical property for confidence estimation tasks (failure prediction and confidence calibration) compared to invariance and convergence rates.

3. **Empirical Evaluation of PI Measures**. We evaluate the three PI measures for both failure prediction and confidence estimation tasks across a diverse range of datasets and architectures. (Section 4). We find that PVI consistently outperforms both PMI and PSI, as well as standard post-hoc methods, in failure prediction across all metrics, with particularly notable gains on larger models and more complex datasets. For confidence calibration, PVI matches temperature-scaled softmax, which is already regarded as a highly effective baseline, indicating that PVI achieves superior failure prediction without compromising its calibration performance.

## 2. Information-Theoretic Measures

**Notation.** We use uppercase letters for random variables (e.g., $X$), corresponding lowercase letters for their values/outcomes (e.g., $x$), and calligraphic letters for their domains (e.g., $\mathcal{X}$). The joint probability distribution of $X, Y$ is denoted by $P_{XY} = P(X, Y)$ and their marginal distributions are denoted by $P_X = P(X)$ and $P_Y = P(Y)$. For specific outcomes $x$ and $y$, we have $p(x, y) = P(X = x, Y = y)$, $p(x) = P(X = x)$, and $p(y) = P(Y = y)$. Here, we provide the formal definitions of the three PI measures (more details on their properties and estimators are given in the Appendix A.2 and Appendix A.3).

**MI and Pointwise MI.** MI measures the statistical dependence between two random variables (Cover & Thomas, 2001), while PMI measures the association between specific realizations of these random variables (Fano & Hawkins, 1961a). They are defined as follows:

**Definition 1** (**MI and PMI**). *Let* $(x, y) \sim P_{XY}$. *The MI and PMI are defined as follows:*

$$I(X; Y) := \mathbb{E}_{X,Y}\left[\log \frac{P(X, Y)}{P(X)P(Y)}\right], \qquad (1)$$

$$pmi(x; y) := \log \frac{p(x, y)}{p(x)p(y)}. \qquad (2)$$

**PMI Estimator:** (Tsai et al., 2020) proposed three methods to compute the probability density ratio $p(x, y)/p(x)p(y)$ using neural networks: the probabilistic classifier method, the density-ratio fitting method and the variational JS bound method. We compare the three methods in the Appendix D.2.1 and choose the variational JS bound method as the default estimator. This estimator relies on the variational form of MI, and in particular the Jensen-Shannon divergence between $P_{XY}$ and $P_X P_Y$ (Poole et al., 2019). We note that although our method for estimating PMI incorporates neural networks, we utilize only a shallow 2-layer neural network, which is often considered well-calibrated (Guo et al., 2017).

**SMI and pointwise SMI.** SMI was proposed by (Goldfeld & Greenewald, 2021) as an alternative measure to MI, which can be hard to estimate in high dimensions. Similarly, its pointwise variant, PSI, was proposed as an alternative measure to PMI (Wongso et al., 2023b). Both SMI and PSI can easily scale to high dimensions by taking one-dimensional projections.

**Definition 2** (**SMI and PSI**). *Let* $(x, y) \sim P_{XY} \in \mathcal{P}(\mathbb{R}^{d_x} \times \mathbb{R}^{d_y})$. *Let* $\Theta \sim \text{Unif}(\mathbb{S}^{d_x-1})$ *and* $\Phi \sim \text{Unif}(\mathbb{S}^{d_y-1})$ *be independent of each other and* $(X, Y)$. *The SMI and PSI are defined as follows:*

$$SI(X; Y) := \mathbb{E}_{\substack{\theta \in \Theta \\ \phi \in \Phi}}[I(\theta^T X; \phi^T Y)], \qquad (3)$$

$$psi(x; y) := \mathbb{E}_{\substack{\theta \in \Theta \\ \phi \in \Phi}}\left[pmi(\theta^T x; \phi^T y)\right]. \qquad (4)$$

**PSI Estimator.** The estimation of PSI for supervised learning tasks requires projecting only the feature vector $x$ to one dimension, while labels $y$ are typically discrete and therefore not projected. Using Bayes' Theorem, it can be rewritten as follows: $psi(x; y) := \mathbb{E}_{\theta \in \Theta}\left[\log \frac{p(\theta^T x | y)}{p(\theta^T x)}\right]$. To estimate $p(\theta^T x | y)$, we use a binning method or assume a Gaussian distribution. We compare the two estimators in the Appendix D.2.2 and use the Gaussian-based estimator (with 500 projections) in our experiments.

**$\mathcal{V}$I and Pointwise $\mathcal{V}$I.** $\mathcal{V}$I was introduced to relax the unbounded computation assumption of Shannon information, which may not be realistic in practice (Xu et al., 2020). It was later extended to its pointwise version, PVI, in (Ethayarajh et al., 2022), for individual samples.

**Definition 3** (**$\mathcal{V}$I and PVI**). *Let* $(x, y) \sim P_{XY} \in \mathcal{P}(\mathcal{X} \times \mathcal{Y})$ *and* $\varnothing$ *represent a null input that provides no information about* $Y$. *We are given predictive family* $\mathcal{V} \subseteq \Omega = \{f : \mathcal{X} \cup \varnothing \to P(\mathcal{Y})\}$. *We first define the $\mathcal{V}$-entropy and conditional $\mathcal{V}$-entropy as follows:*

$$H_{\mathcal{V}}(Y) := \inf_{f \in \mathcal{V}} \mathbb{E}_Y[-\log f[\varnothing](Y)], \qquad (5)$$

$$H_{\mathcal{V}}(Y|X) := \inf_{f \in \mathcal{V}} \mathbb{E}_{X,Y}[-\log f[X](Y)] \qquad (6)$$

*Let* $g = \arg\min_{f \in \mathcal{V}} \mathbb{E}_Y[-\log f[\varnothing](Y)]$ *and* $g' = \arg\min_{f \in \mathcal{V}} \mathbb{E}_{X,Y}[-\log f[X](Y)]$. *The $\mathcal{V}$I and PVI are defined as follows:*

$$I_{\mathcal{V}}(X \to Y) := H_{\mathcal{V}}(Y) - H_{\mathcal{V}}(Y|X), \qquad (7)$$

$$pvi(x \to y) := -\log g[\varnothing](y) + \log g'[x](y) \qquad (8)$$

**PVI Estimator.** The estimation of PVI requires training two neural networks: $f$ for estimating $H_{\mathcal{V}}(Y)$ and $f'$ for estimating the conditional $H_{\mathcal{V}}(Y|X)$ (Ethayarajh et al., 2022). $f'$ is trained with the input-label pairs from the training data $(x_{\text{train}}, y_{\text{train}})$ while $f$ is trained with the null input-label pairs from the training data $(x_{\text{null}}, y_{\text{train}})$. For computer vision tasks, images composed entirely of zeros can be treated as null inputs. The PVI can then be computed as: $pvi(x \to y) = -\log f[\varnothing](y) + \log f'[x](y)$ where $(x, y)$ is an input-label pair from a held-out set. To ensure that the probabilities for computing PVI are properly calibrated, we consider using temperature scaling. We consider three different approaches: using the original trained network as $f$, using the same architecture but trained with different initialization as $f$ and using a one-hidden layer neural network as $f$ trained using penultimate features as inputs. We compare the three approaches in the Appendix D.2.3 and use the second approach (same architecture, different initialization) as the default estimator for PVI.

# 3. Theoretical Properties

In this section, we analyze the theoretical properties of the three PI measures, focusing on their invariance and convergence rate analyses. Results on their geometric properties are provided in Appendix B.3. Proofs and additional remarks are given in Appendix B.

## 3.1. Invariance Properties of PI measures

We argue that confidence estimates should remain stable under certain transformations of the data distribution, e.g., translation or rotation. Intuitively, if a distribution is transformed and a point $x$ is mapped accordingly, the confidence assigned to $x$ should be preserved in the transformed space. This is essential for ensuring that confidence measures reflect intrinsic uncertainty and are not confounded by bijective transformations of the input space.

In what follows, we consider the case where $X \in \mathbb{R}^{d_x}$ are the features and $Y \in \{0, 1\}$ are the labels, and $(x, y)$ is a feature-label instance sampled from $P_{XY}$. Note that in what follows, other than Theorem 1, all other theoretical results can be trivially extended to the multi-label setting.

For convenience of notation, when $(x, y) \sim P_{XY}$, we denote $pmi(x; y)$, $psi(x; y)$ and $pvi(x \rightarrow y)$ by $pmi_P(x, y)$, $psi_P(x, y)$ and $pvi_P(x, y)$ respectively. For estimating $pvi_P(x, y)$, we assume that $\mathcal{V}$ refers to a fully connected neural network of arbitrary depth and fixed architecture, where each layer contains weights and biases. For any transformation $\mathcal{T} : \mathbb{R}^d \rightarrow \mathbb{R}^d$, we denote the probability distribution $P(\mathcal{T}X, Y)$ by $\mathcal{T}P$. We have the following results.

**Proposition 1** (**Invariance to shift, scale, and rotation**). *Let $\mathcal{T}x = \alpha \boldsymbol{R}x + \boldsymbol{p}$, where $\boldsymbol{p} \in \mathbb{R}^{d_x}$ represents the extent to which the distribution is shifted, and $\alpha \in \mathbb{R}$ is a scalar that represents how much the distribution is scaled. Furthermore, $\boldsymbol{R} \sim \mathbb{R}^{d_x \times d_x}$ is a rotation matrix, such that we have $\boldsymbol{R}\boldsymbol{R}^T = I$ and $\det(\boldsymbol{R}) = 1$, where $I$ is the identity matrix and $\det$ represents the determinant operator. Then, we have: $pmi_P(x, y) = pmi_{\mathcal{T}P}(\alpha \boldsymbol{R}x + \boldsymbol{p}, y)$, $psi_P(x, y) = psi_{\mathcal{T}P}(\alpha \boldsymbol{R}x + \boldsymbol{p}, y)$ and $pvi_P(x, y) = pvi_{\mathcal{T}P}(\alpha \boldsymbol{R}x + \boldsymbol{p}, y)$.*

Next, we have the following results for more general linear transformations and homeomorphic (continuous and invertible) transformations.

**Proposition 2** (**Invariance to general linear transformations**). *Let $\mathcal{T}x = \boldsymbol{M}x$, where $\boldsymbol{M} \sim \mathbb{R}^{d_x \times d_x}$ is invertible. Then, we have: $pvi_P(x, y) = pvi_{\mathcal{T}P}(\boldsymbol{M}x, y)$ and $pmi_P(x, y) = pmi_{\mathcal{T}P}(\boldsymbol{M}x, y)$.*

**Proposition 3** (**Invariance to homeomorphic transformations**). *Let $\mathcal{T}x = f(x)$, where $f : \mathbb{R}^{d_x} \rightarrow \mathbb{R}^{d_x}$ represents any homeomorphism. Then, we have: $pmi_P(x, y) = pmi_{\mathcal{T}P}(f(x), y)$.*

**Remark 1** (**Invariance and confidence estimation**). *We note that invariance to bijective transformations $\mathcal{T}$ is important in the context of confidence estimation; otherwise, the PI measures will confound $\mathcal{T}$ in their estimates. Using the terminology from (1) (Mukhoti et al., 2023), this can be expressed as: $H[Y|x, D] = H[Y|\mathcal{T}x, \mathcal{T}D]$, where $H[Y|\mathcal{T}x, \mathcal{T}D]$ denotes the conditional entropy of the output labels given both the transformed input $\mathcal{T}x$ and the transformed dataset $\mathcal{T}D = \{(\mathcal{T}x_1, y_1), ..., (\mathcal{T}x_n, y_n)\}$, which implies that the underlying distribution has been transformed as well. Ideally, this property should hold for any invertible, and thus information-preserving, transformation $\mathcal{T}$. However, as we cannot ignore the constraints of the model involved in the decision-making process, we restrict $\mathcal{T}$ to invertible linear transformations on $x$. In the following remark, we argue that invariance to the broader class of homeomorphic transformations may be counterproductive.*

**Remark 2** (**Caveat on invariance to homeomorphic transformations**). *In neural network architectures, feature representations at a given layer typically undergo complex transformations through multiple nonlinear activations, making them highly unlikely to remain invertible. Consider the case where features at a given layer appear as transformed versions of themselves across different training iterations. This can occur due to random weight initializations, where features $T$ from one run may correspond to $WT'$ from another, with $T'$ being the feature from a different initialization and $W$ being an invertible matrix. Since all such versions of $T$ encode the same underlying information up to a linear map, pointwise measures between $T$ and the output labels should remain unchanged, thereby emphasizing the need for invariance to rotational and random matrix transformations. However, extending this invariance to general invertible transformations, as seen in the case of PMI, can be counterproductive. The degree of non-linearity between $T$ and $Y$ can be indicative of the network's confidence in estimating $Y$ from $T$. Intuitively, more complex, non-linear maps are less likely to yield confident predictions and vice-versa. Consequently, while PMI's invariance holds under broader homeomorphic transformations, we hypothesize that this property may not always be beneficial, which is also seen in our experiments.*

## 3.2. Convergence Rates

We provide convergence rates analysis for the estimators studied here. We note that PMI's convergence rates will depend on the choice of probability estimator, as different estimators give different convergence rates. Here, we mainly focus on the Kernel Density Estimator (KDE) for estimating the densities $p(x|y)$ and $p(x)$. We consider the case of binary classification, thus, $Y \in \{0, 1\}$. For the KDE estimator studied in (Jiang, 2017), we have the following convergence bound for PMI.

**Proposition 4** (**PMI convergence rate**). *Let $P(X)$ be $\alpha$-Holder continuous and let $(x, y) \sim P_{XY}$ where $X \in \mathbb{R}^{d_x}$ and $Y \in \{0, 1\}$. Let $\widehat{pmi}_n$ represent the KDE estimate of PMI using $n$ samples. Assuming $\min \{P(Y = 0), P(Y = 1)\} \neq 0$ when the probabilities are estimated on the training data, for large enough $n$, we can bound the estimation error as*

$$\left| pmi(x; y) - \widehat{pmi}_n \right| \leq \mathcal{O} \left( \frac{n^{-\alpha/(2\alpha+d_x)}}{\min \{p(x), p(x|y)\}} \right) \quad (9)$$

In the following result, we provide convergence rate for PSI, when the KDE approach (Jiang, 2017) is used to estimate $p(\theta^T x)$ and $p(\theta^T x|y)$.

**Proposition 5** (**PSI convergence rate**). *Let $P(\theta^T X)$ be $\alpha$-Holder continuous for all $\theta$ and let $(x, y) \sim P_{XY}$ where $X \in \mathbb{R}^{d_x}$ and $Y \in \{0, 1\}$. Let $\widehat{psi}_{n,m}$ represent the KDE estimate of PSI using $n$ samples and $m$ projections. Furthermore, let $\min_\theta pmi(\theta^T x; y) \geq \rho$. Assuming $\min \{P(Y = 0), P(Y = 1)\} \neq 0$ when the probabilities are estimated on the training data, for large enough $n$, we can bound the estimation error as*

$$\mathbb{E}_{X,Y} \left[ \left| \psi(x; y) - \widehat{\psi}_{n,m} \right| \right] \leq \frac{1-\rho}{2\sqrt{m}}$$
$$+ \mathcal{O} \left( \frac{n^{-\alpha/(2\alpha+1)}}{\min_\theta \min \{p(\theta^\top x),\; p(\theta^\top x \mid y)\}} \right) \quad (10)$$

For PVI, we have the following bound on the expected deviation of the PVI estimates.

**Theorem 1** (**PVI convergence rate**). *Given $(x, y) \sim P_{XY}$ where $X \in \mathbb{R}^{d_x}$ and $Y \in \{0, 1\}$, we assume that $P(Y = 0) = P(Y = 1) = 0.5$. Assume $\mathcal{V}$ represents the set of all possible functions modelled by a neural network having some fixed architecture. Assume $\forall f \in \mathcal{V}$, $\log f[x](y) \in [-B, B]$. Also, let $f^* = \arg\min_{f \in \mathcal{V}} \mathbb{E}_{X,Y}[-\log f[X](Y)]$ represent the ground truth function for estimating conditional $\mathcal{V}$-entropy, and $\widehat{f}$ represent the trained function given $n$ datapoints $(x_1, y_1), ..., (x_n, y_n)$ sampled from $P_{XY}^n$. Let $M = \max\{var(f^*[x](y)), var(\widehat{f}[x](y))\}$ where $var$ denotes the variance. Let $\widehat{pvi}_n$ represent the PVI estimated using this neural network with $n$ samples. Then, for any $\delta \in (0, 0.5)$, with probability $p \geq 1 - 2\delta$, we have*

$$\mathbb{E}_{X,Y} \left[ \left| pvi(x \to y) - \widehat{pvi}_n \right| \right] \leq 2\mathcal{R}_n(\mathcal{G}_\mathcal{V}) + 2\sqrt{M}$$
$$+ 2B\sqrt{\frac{2\log(1/\delta)}{n}} \quad (11)$$

*where the function family $\mathcal{G}_\mathcal{V} = \{g | g(x, y) = \log f[x](y), f \in \mathcal{V}\}$ and $\mathcal{R}_n$ denotes the Rademacher complexity with $n$ sampled points.*

**Remark 3** (**Convergence rates and confidence estimation**). *Note that convergence rates are a critical factor in determining the stability and accuracy of confidence estimates using these measures. If a measure has slower convergence, then even if it has other desirable invariance and geometric properties, the slower convergence implies more estimation error, thereby decreasing the accuracy of the confidence estimates.*

### 3.3. Theoretical Takeaways

We present a summary of the key takeaways from the theoretical results and their implications for our subsequent experiments. These takeaways will also be referenced in our discussion of the experimental results later.

**T1** We find that the different PI measures each have their own strengths and weaknesses, with no single measure outperforming the others across all scenarios.

**T2 Invariance:** PMI is the most invariant among the three, as it exhibits invariance to any homeomorphic transformation, and thus is the most structure-preserving. However, we note (in Remark 2) that this may not be a boon in the context of confidence estimation, as the model's constraints matter significantly. PSI, on the other hand, is not invariant to general invertible linear transformations, which can hinder performance as neural networks can preserve output function in response to invertible linear transformation on the input, and thereby preserve the confidence as well. Thus, since PVI is invariant to linear invertible transformations but not to homeomorphic transformations, it appears to be the most suitable measure for confidence estimation with respect to invariance properties.

**T3 Convergence Rates:** When comparing PMI and PSI, our theoretical results concretely find that PSI is likely to have better convergence behaviour compared to PMI, following the differences in the order of the sample complexity $n$. We find that PVI's convergence rate depends on the complexity of the predictive family $\mathcal{V}$, thereby making direct comparison with PSI and PMI more challenging. However, in absolute terms, we find that the convergence of PMI and PSI also depends on the spread of the distribution $P(x)$ and the amount of overlap between the class-wise distributions $P(x|y = 1)$ and $P(x|y = 0)$, primarily due to the denominators in (9) and (10). In complex datasets, these distributions are typically broader and exhibit greater overlap, resulting in slower convergence. In contrast, for simpler, more separable datasets such as MNIST, PSI and PMI converge more quickly, as reflected in PMI outperforming the other methods on MNIST for failure prediction (Table 1).

**T4 Margin Sensitivity:** We provide a detailed analysis of the three PI measures with respect to their sensitivity to the classifier's sample-wise margin in Appendix B.3. Ideally, samples closer to the decision boundary should be assigned lower confidence scores. Overall, our results point to PSI

potentially being the most sensitive to sample-wise margin (hard and soft), although we found direct comparisons are not very straightforward. However, we do note that PMI is invariant to hard margin (when the classes are clearly separated), whereas PSI remains sensitive to it. Empirically, we found that PSI exhibits the strongest correlation with margin, as shown in Table 4. However, since PSI performs poorly in failure prediction experiments, this raises an important observation: margin sensitivity alone may not be a sufficient criterion for identifying good confidence measures.

## 4. Experiments

We performed two types of experiments related to confidence estimation: (1) failure prediction and (2) confidence calibration. In all experiments, the PI measures are trained with true labels of the training dataset and evaluated with predicted labels of the test dataset. We also normalize these PI measures with a softmax function to ensure their values lie between 0 and 1. For all experiments, we calibrate all methods using temperature scaling (including the PI measures), except for max logits and logits margin since their values are not in the [0,1] range, to ensure a fair comparison. All experiments are conducted using benchmark datasets and architectures readily available in TensorFlow. More details on the datasets, architectures and training algorithms used in all experiments are provided in Appendix C.

For PVI, we compute it between the input features and the predicted labels, following the approach from in (Ethayarajh et al., 2022), which is to estimate the PVI between $X$ and $Y$ by training another model with the same architecture. It measures how easily we can predict $Y$ from $X$ using $\mathcal{V}$. Thus, it can capture the confidence of the predictive family $\mathcal{V}$. While the way PVI is defined is architecture-dependent, the definitions of PMI and PSI are not. For PMI and PSI, it is more natural to use the features of the model directly and the layers closest to the output should capture the model's confidence about the network the most. For PMI and PSI, we compute them between the output layer features and the predicted labels. Furthermore, instead of computing the measures with just the predicted class, we compute them for all classes and apply softmax function along with temperature scaling. More discussion on this can be found in Appendix D.1. In this way, the PI values are normalized to a range between 0 and 1.

### 4.1. Failure Prediction

**Goal:** The goal of this experiment is to compare the effectiveness of different confidence estimates for failure prediction. Failure prediction typically involves three tasks: misclassification detection, selective prediction, and out-of-distribution detection (Jaeger et al., 2023). This work focuses on the first two tasks. In misclassification detection,

the objective is to identify incorrect predictions made by trained networks. Ideally, confidence scores should be high for correct predictions and low for incorrect ones. In selective prediction, the aim is to evaluate the improvement in classification performance after excluding a certain percentage of low-confidence predictions.

**Methodology:** For misclassification detection, we evaluate the effectiveness of confidence estimates in distinguishing between positive (incorrect predictions) and negative (correct predictions) samples. We use a threshold-independent metric, $\text{AUROC}_f$ (Area Under the ROC Curve, with $f$ denoting failure), which is widely adopted in the literature (Hendrycks & Gimpel, 2017; Jaeger et al., 2023). Since AUROC is less informative when the positive and negative classes have significantly different base rates, we also consider another metric called AUPR (Area Under the Precision-Recall Curve). Given that the base rate of the positive class greatly influences AUPR, we examine both scenarios: treating success classes as positive samples ($\text{AUPR}_f$, success) and treating error classes as positive samples ($\text{AUPR}_f$, error). We also compute the FPR95, which measures the false positive rate when the true positive rate is at 95%. For selective prediction, we examine the improvement in classification error rates by filtering out low-confidence samples. In this context, we define risk as the error rate on the remaining samples, and coverage as the proportion of remaining samples relative to the total samples. We employ a threshold-independent metric, E-AURC (Area Under the Risk-Coverage Curve), as described in the literature (Geifman et al., 2019; Jaeger et al., 2023). We compare our results against six benchmark methods: maximum softmax probability (MSP) (Geifman & El-Yaniv, 2017), softmax margin (SM) (Tagasovska & Lopez-Paz, 2019), max logit (ML) (Hendrycks et al., 2022), logits margin (LM) (Streeter, 2018), negative entropy (NE) (Belghazi & Lopez-Paz, 2021) and negative Gini index (NG) (Granese et al., 2021). We report the results in Table 1. Note that we omit PMI computation for datasets with a very large number of classes (CIFAR-100, Stanford Dogs, and TinyImageNet) due to its high computational cost.

**Results:** We observe that PVI generally outperforms the other two PI measures, as well as other benchmark post-hoc methods, across a range of metrics. The performance gains are particularly pronounced for $\text{AUPR}_{f,\text{failure}}$, $\text{FPR95}_f$, and AURC. A higher $\text{AUPR}_{f,\text{failure}}$ and a lower $\text{FPR95}_f$ indicate that PVI can reliably assign lower confidence to misclassified examples, while a lower E-AURC indicates that PVI can reliably defer uncertain predictions. The superior performance of PVI is especially true for larger models and datasets. We argue that this is due to PVI being generally well-rounded in terms of its theoretical properties for confidence estimation, as discussed in Section 3.3.

Table 1: Comparison of Confidence Estimation Methods for Failure Prediction (Averaged over 10 Runs).

| Model, Dataset | Method | $\text{AUROC}_f \times 10^2 \uparrow$ | $\text{AUPR}_{f,\text{success}} \times 10^2 \uparrow$ | $\text{AUPR}_{f,\text{error}} \times 10^2 \uparrow$ | $\text{FPR95}_f \times 10^2 \downarrow$ | $\text{E-AURC} \times 10^3 \downarrow$ |
|---|---|---|---|---|---|---|
| MLP, MNIST | MSP | 96.80 ± 0.17 | 99.93 ± 0.00 | 43.17 ± 2.49 | 16.25 ± 1.64 | 0.69 ± 0.05 |
| | SM | 96.75 ± 0.17 | 99.93 ± 0.00 | 40.45 ± 1.94 | 16.21 ± 1.65 | 0.70 ± 0.05 |
| | ML | 90.89 ± 0.32 | 99.78 ± 0.01 | 28.85 ± 1.49 | 40.67 ± 1.44 | 2.12 ± 0.10 |
| | LM | 96.67 ± 0.18 | 99.93 ± 0.01 | 39.40 ± 1.89 | 16.87 ± 1.66 | 0.71 ± 0.05 |
| | NE | 96.76 ± 0.17 | 99.93 ± 0.01 | 42.41 ± 2.02 | 16.09 ± 1.65 | 0.70 ± 0.05 |
| | NG | 96.80 ± 0.17 | 99.93 ± 0.00 | 43.31 ± 2.30 | 16.20 ± 1.64 | 0.69 ± 0.05 |
| | PMI | **97.13 ± 0.14** | **99.94 ± 0.00** | 45.94 ± 2.25 | 14.82 ± 1.39 | **0.61 ± 0.04** |
| | PSI | 94.77 ± 0.46 | 99.88 ± 0.01 | 31.01 ± 1.99 | 27.89 ± 1.54 | 1.19 ± 0.14 |
| | PVI | 96.94 ± 0.22 | 99.93 ± 0.01 | **48.28 ± 3.20** | **15.59 ± 1.46** | 0.66 ± 0.05 |
| CNN, F-MNIST | MSP | 92.53 ± 0.17 | 99.39 ± 0.03 | 44.71 ± 1.21 | 46.94 ± 1.73 | 5.72 ± 0.29 |
| | SM | 92.42 ± 0.17 | 99.39 ± 0.03 | 42.77 ± 1.50 | 47.62 ± 1.66 | 5.80 ± 0.29 |
| | ML | 86.76 ± 0.50 | 98.85 ± 0.08 | 32.38 ± 1.01 | 63.26 ± 1.73 | 10.91 ± 0.76 |
| | LM | 92.29 ± 0.17 | 99.37 ± 0.03 | 42.12 ± 1.52 | 48.21 ± 1.67 | 5.91 ± 0.29 |
| | NE | 92.40 ± 0.17 | 99.38 ± 0.03 | 43.38 ± 0.93 | 48.85 ± 1.87 | 5.82 ± 0.31 |
| | NG | 92.52 ± 0.16 | 99.39 ± 0.03 | 44.51 ± 1.04 | 46.30 ± 1.64 | 5.73 ± 0.29 |
| | PMI | 91.38 ± 0.40 | 99.26 ± 0.03 | 42.37 ± 1.62 | 48.95 ± 1.90 | 6.99 ± 0.30 |
| | PSI | 89.95 ± 0.38 | 99.15 ± 0.07 | 36.28 ± 0.96 | 56.76 ± 2.03 | 8.02 ± 0.61 |
| | PVI | **92.98 ± 0.52** | **99.43 ± 0.03** | **50.01 ± 4.22** | **43.26 ± 3.36** | **5.38 ± 0.33** |
| VGG16, STL-10 | MSP | 92.04 ± 0.25 | 99.15 ± 0.05 | 50.75 ± 1.24 | 48.21 ± 1.69 | 7.93 ± 0.48 |
| | SM | 91.97 ± 0.25 | 99.14 ± 0.05 | 48.61 ± 1.64 | 49.12 ± 1.95 | 7.97 ± 0.48 |
| | ML | 89.50 ± 0.40 | 98.82 ± 0.07 | 45.70 ± 1.44 | 53.52 ± 1.30 | 10.94 ± 0.67 |
| | LM | 91.82 ± 0.25 | 99.13 ± 0.05 | 47.51 ± 1.64 | 50.81 ± 1.77 | 8.11 ± 0.48 |
| | NE | 91.77 ± 0.26 | 99.12 ± 0.06 | 48.95 ± 1.04 | 49.93 ± 1.48 | 8.19 ± 0.50 |
| | NG | 92.01 ± 0.25 | 99.14 ± 0.05 | 50.25 ± 1.12 | 48.04 ± 1.23 | 7.95 ± 0.49 |
| | PMI | 91.46 ± 0.37 | 99.07 ± 0.06 | 49.48 ± 1.95 | 49.12 ± 1.92 | 8.66 ± 0.56 |
| | PSI | 91.79 ± 0.32 | 99.11 ± 0.06 | 50.78 ± 1.74 | 47.70 ± 1.56 | 8.29 ± 0.55 |
| | PVI | **92.79 ± 0.62** | **99.23 ± 0.06** | **55.88 ± 3.50** | **41.97 ± 3.61** | **7.18 ± 0.53** |
| ResNet50, CIFAR-10 | MSP | 93.59 ± 0.24 | 99.63 ± 0.02 | 43.42 ± 1.20 | 36.29 ± 1.23 | 3.52 ± 0.22 |
| | SM | 93.71 ± 0.24 | 99.64 ± 0.02 | 43.05 ± 1.23 | 35.91 ± 1.10 | 3.44 ± 0.22 |
| | ML | 91.80 ± 0.30 | 99.52 ± 0.02 | 37.54 ± 1.13 | 45.35 ± 1.38 | 4.58 ± 0.23 |
| | LM | 93.80 ± 0.23 | 99.65 ± 0.02 | 42.74 ± 1.23 | 35.85 ± 1.23 | 3.38 ± 0.21 |
| | NE | 93.29 ± 0.23 | 99.62 ± 0.02 | 40.49 ± 1.16 | 38.57 ± 1.64 | 3.67 ± 0.21 |
| | NG | 93.55 ± 0.24 | 99.63 ± 0.02 | 42.44 ± 1.17 | 36.56 ± 1.35 | 3.54 ± 0.22 |
| | PMI | 93.11 ± 0.30 | 99.60 ± 0.02 | 43.51 ± 1.53 | 37.43 ± 1.71 | 3.87 ± 0.23 |
| | PSI | 93.51 ± 0.22 | 99.63 ± 0.02 | 43.27 ± 1.34 | 37.44 ± 1.64 | 3.57 ± 0.19 |
| | PVI | **95.14 ± 0.30** | **99.72 ± 0.02** | **60.99 ± 1.60** | **25.77 ± 1.56** | **2.72 ± 0.17** |
| ResNet101, CIFAR-100 | MSP | 85.54 ± 0.31 | 95.25 ± 0.15 | 60.83 ± 0.82 | 63.47 ± 0.96 | 39.17 ± 1.20 |
| | SM | 85.92 ± 0.29 | 95.41 ± 0.15 | 60.78 ± 0.62 | 63.14 ± 0.58 | 37.85 ± 1.14 |
| | ML | 82.48 ± 0.48 | 93.92 ± 0.22 | 57.20 ± 0.94 | 65.98 ± 1.13 | 50.15 ± 1.78 |
| | LM | 86.27 ± 0.28 | 95.65 ± 0.14 | 59.40 ± 0.55 | 65.23 ± 0.94 | 35.89 ± 1.07 |
| | NE | 84.33 ± 0.34 | 94.88 ± 0.16 | 57.57 ± 0.87 | 66.75 ± 1.11 | 42.32 ± 1.29 |
| | NG | 85.32 ± 0.31 | 95.20 ± 0.15 | 59.75 ± 0.86 | 64.61 ± 1.15 | 39.62 ± 1.21 |
| | PSI | 84.30 ± 0.26 | 94.72 ± 0.15 | 57.40 ± 1.01 | 66.44 ± 0.76 | 43.57 ± 1.11 |
| | PVI | **89.02 ± 0.38** | **96.27 ± 0.11** | **71.54 ± 1.42** | **47.17 ± 1.73** | **30.44 ± 0.85** |
| InceptionV3, Stanford Dogs | MSP | 83.52 ± 0.34 | 93.80 ± 0.31 | 59.85 ± 0.71 | 66.72 ± 1.03 | 50.16 ± 2.31 |
| | SM | 83.63 ± 0.37 | 93.91 ± 0.32 | 58.58 ± 0.70 | 67.78 ± 1.08 | 49.36 ± 2.36 |
| | ML | 81.18 ± 0.34 | 92.71 ± 0.31 | 56.87 ± 0.70 | 69.03 ± 0.93 | 59.16 ± 2.25 |
| | LM | 83.51 ± 0.44 | 94.05 ± 0.32 | 56.73 ± 0.69 | 70.46 ± 1.44 | 48.33 ± 2.40 |
| | NE | 82.58 ± 0.32 | 93.48 ± 0.32 | 57.49 ± 0.66 | 69.14 ± 0.93 | 52.91 ± 2.32 |
| | NG | 83.38 ± 0.33 | 93.76 ± 0.31 | 59.20 ± 0.70 | 67.48 ± 0.87 | 50.50 ± 2.30 |
| | PSI | 81.80 ± 0.25 | 93.32 ± 0.23 | 54.36 ± 0.86 | 73.73 ± 0.83 | 54.41 ± 1.67 |
| | PVI | **85.73 ± 0.73** | **94.50 ± 0.25** | **66.59 ± 2.30** | **57.05 ± 2.60** | **44.25 ± 2.20** |
| DenseNet121, TinyImageNet | MSP | 87.01 ± 0.34 | 94.28 ± 0.18 | 70.87 ± 0.83 | 58.29 ± 1.21 | 43.92 ± 1.36 |
| | SM | 86.95 ± 0.35 | 94.35 ± 0.18 | 69.22 ± 0.88 | 60.79 ± 1.42 | 43.50 ± 1.38 |
| | ML | 84.37 ± 0.45 | 92.58 ± 0.31 | 67.80 ± 0.89 | 61.42 ± 1.21 | 56.97 ± 2.29 |
| | LM | 86.50 ± 0.35 | 94.37 ± 0.18 | 66.60 ± 0.83 | 65.44 ± 1.33 | 43.58 ± 1.38 |
| | NE | 86.02 ± 0.37 | 93.86 ± 0.21 | 68.41 ± 0.86 | 61.82 ± 1.19 | 47.34 ± 1.56 |
| | NG | 86.87 ± 0.34 | 94.24 ± 0.18 | 70.23 ± 0.83 | 59.28 ± 1.12 | 44.29 ± 1.36 |
| | PSI | 84.78 ± 0.42 | 93.18 ± 0.23 | 64.73 ± 0.83 | 67.42 ± 1.45 | 52.65 ± 1.75 |
| | PVI | **89.66 ± 0.27** | **95.32 ± 0.14** | **77.82 ± 0.69** | **46.89 ± 0.90** | **35.59 ± 1.10** |

Table 2: Comparison of Confidence Estimation Methods for Confidence Calibration (Averaged over 10 Runs).

| Model, Dataset | Method | SCE ↓ | CC-SCE ↓ | Ada-SCE ↓ | RMS-SCE ↓ |
|---|---|---|---|---|---|
| MLP, MNIST | MSP | 0.06 ± 0.01 | 0.18 ± 0.01 | **0.01 ± 0.00** | 0.44 ± 0.09 |
| | PMI | **0.05 ± 0.01** | **0.17 ± 0.01** | **0.01 ± 0.00** | 0.42 ± 0.07 |
| | PSI | 0.14 ± 0.02 | 0.30 ± 0.02 | 0.03 ± 0.00 | 0.75 ± 0.10 |
| | PVI | **0.05 ± 0.01** | 0.18 ± 0.01 | **0.01 ± 0.00** | **0.41 ± 0.10** |
| CNN, F-MNIST | MSP | 0.15 ± 0.02 | 0.35 ± 0.02 | 0.07 ± 0.01 | 0.52 ± 0.05 |
| | PMI | 0.23 ± 0.02 | 0.58 ± 0.05 | **0.06 ± 0.02** | 0.93 ± 0.11 |
| | PSI | 0.15 ± 0.01 | 0.57 ± 0.05 | 0.08 ± 0.02 | 0.52 ± 0.04 |
| | PVI | **0.14 ± 0.02** | **0.34 ± 0.02** | 0.07 ± 0.01 | **0.50 ± 0.06** |
| VGG16, STL-10 | MSP | 0.28 ± 0.06 | **0.59 ± 0.03** | **0.12 ± 0.03** | 0.94 ± 0.14 |
| | PMI | 0.23 ± 0.03 | 0.65 ± 0.04 | 0.16 ± 0.02 | 0.82 ± 0.10 |
| | PSI | 0.22 ± 0.05 | 0.67 ± 0.05 | 0.15 ± 0.04 | 0.72 ± 0.11 |
| | PVI | **0.20 ± 0.03** | 0.61 ± 0.04 | **0.12 ± 0.03** | **0.68 ± 0.09** |
| ResNet50, CIFAR-10 | MSP | 0.19 ± 0.01 | 0.35 ± 0.02 | **0.03 ± 0.01** | 0.88 ± 0.07 |
| | PMI | **0.15 ± 0.01** | 0.39 ± 0.02 | 0.04 ± 0.01 | **0.74 ± 0.08** |
| | PSI | 0.17 ± 0.01 | 0.35 ± 0.02 | 0.04 ± 0.01 | 0.85 ± 0.08 |
| | PVI | 0.17 ± 0.00 | **0.33 ± 0.00** | **0.03 ± 0.00** | 0.85 ± 0.01 |
| ResNet101, CIFAR-100 | MSP | 0.08 ± 0.00 | **0.20 ± 0.00** | 0.04 ± 0.00 | 0.68 ± 0.04 |
| | PSI | **0.07 ± 0.01** | 0.31 ± 0.01 | **0.02 ± 0.00** | **0.49 ± 0.05** |
| | PVI | 0.08 ± 0.00 | **0.20 ± 0.00** | 0.04 ± 0.00 | 0.68 ± 0.04 |
| InceptionV3, Stanford Dogs | MSP | 0.07 ± 0.00 | **0.20 ± 0.01** | 0.03 ± 0.00 | 0.63 ± 0.03 |
| | PSI | **0.05 ± 0.01** | 0.28 ± 0.01 | **0.02 ± 0.00** | **0.47 ± 0.07** |
| | PVI | 0.07 ± 0.00 | **0.20 ± 0.01** | 0.03 ± 0.00 | 0.63 ± 0.03 |
| DenseNet121, TinyImageNet | MSP | 0.03 ± 0.00 | **0.12 ± 0.00** | **0.01 ± 0.00** | 0.32 ± 0.03 |
| | PSI | 0.03 ± 0.00 | 0.19 ± 0.00 | 0.02 ± 0.00 | **0.29 ± 0.03** |
| | PVI | **0.02 ± 0.00** | **0.12 ± 0.00** | **0.01 ± 0.00** | 0.29 ± 0.02 |

## 4.2. Confidence Calibration

**Goal:** The goal of confidence calibration is to determine whether the confidence scores reflect the true correctness likelihood (Guo et al., 2017). Perfect calibration is defined as follows: $\mathbb{P}(\hat{Y} = Y | \hat{P} = p) = p, \quad \forall p \in [0, 1]$, where $Y$ denotes the ground-truth labels, $\hat{Y}$ denotes the predicted labels and $\hat{P}$ is the associated probability.

**Methodology:** Expected Calibration Error (ECE), a widely used calibration metric, has known limitations: it relies on a fixed length binning approach that introduces bias–variance trade-offs (fewer bins increase bias, more bins escalate variance), considers only the maximum predicted probabilities (ignoring potentially meaningful information in lower-confidence estimates), and may mislead by yielding low calibration error even in poorly accurate models (Pavlovic, 2025; Nixon et al., 2019). Instead, we use the variants proposed in (Nixon et al., 2019). Static Calibration Error (SCE) is the multiclass extension of ECE that evaluates calibration over all predicted class probabilities, not just the top prediction. Class-Conditional SCE (CC-SCE) computes SCE separately for each class (binning and accuracy calculation per class), then averages the errors across classes. Adaptive SCE (Ada-SCE) uses adaptive (quantile-based) binning so that each bin contains approximately the same number of predictions, improving stability in accuracy estimates. RMS-SCE replaces the L1 norm in the error term with the L2 norm before averaging, thereby penalizing larger miscalibrations more strongly. The results are shown in Table 2.

We also include the results for other calibration metrics in Table 8 in the Appendix. Similar to the failure prediction experiment, we omit PMI computation for datasets with a very large number of classes (CIFAR-100, Stanford Dogs, and TinyImageNet) due to its high computational cost.

**Results:** The results in Table 2 show that MSP and PVI consistently perform well across most datasets and architectures, reinforcing the observation in (Guo et al., 2017) that simple temperature scaling can already be highly effective for confidence calibration. PVI, in particular, often matches or slightly outperforms MSP, especially in SCE and RMS-SCE, but the differences are typically small. PMI and PSI each show sporadic strengths in specific metrics, suggesting they may offer situational advantages depending on the dataset and calibration criterion. Overall, we find that PVI demonstrates more consistent calibration than MSP across all four metrics, with notable cases such as the substantially lower RMS-SCE for the VGG16–STL-10 pair.

## 5. Reflections

We performed a comparative analysis of using three PI measures, namely PMI, PVI, and PSI, for confidence estimation in DNNs. We studied several theoretical properties which we believe can be relevant to confidence estimation, including how well a measure behaves in response to data transformations (invariance properties), how well a measure converges with data (convergence rates) and how well

a measure tracks the geometric difficulty of classifying a feature point (sample-wise margin). We performed a series of experiments on confidence estimation (failure prediction and confidence calibration) to test and verify our theoretical hypothesis. Our findings show that PVI outperforms PMI, PSI, and benchmark post-hoc methods in failure prediction, while remaining competitive in confidence calibration. This highlights PVI's versatility, especially given that popular confidence calibration methods have been shown to be ineffective or even detrimental for failure prediction tasks (Zhu et al., 2022). This is consistent with our theoretical findings which suggest that PVI is generally well-rounded when considering the three theoretical properties analyzed here.

One of our findings in this work has been that better sensitivity to margin does not necessarily imply better performance in the confidence prediction problem. We note that for the correlation to margin experiment (Table 4), the focus is on whether the model assigns higher confidence to samples with a larger margin (and vice versa), regardless of whether the prediction is correct. On the other hand, for the misclassification detection, selective prediction, and calibration analysis, the focus is more on the correctness of predictions (directly linked to accuracy). The contrast lies in the interpretation of confidence: margin experiments treat confidence as a measure of sensitivity to decision boundaries, while the other tasks treat it as a measure of predictive reliability. Therefore, our results highlight that the two interpretations of confidence may not be consistent with each other, i.e., sample-wise margins may not be reflective of true confidence.

For future work, these PI measures could be explored for confidence estimation in other modalities (e.g., image, audio, tabular, etc.). In addition, one could explore the potential of using PI measures for other aspects of trustworthy machine learning such as explainability and privacy. Furthermore, we could study other scaling and normalization techniques to improve the performance of the PI measures. We hope that both our theoretical and empirical findings will motivate more work in the direction of information-theoretic approaches for confidence estimation in the context of DNNs.

Another promising direction for future work is to integrate PVI into existing confidence estimation methods (not necessarily post-hoc) that rely on logit-based softmax outputs for generating confidence scores, to evaluate whether it can further improve performance. This includes popular techniques such as Monte Carlo dropout, deep ensembles, focal loss, and various temperature scaling methods. The integration is straightforward: replace the logit-based softmax outputs with softmax-scaled PVI values when computing confidence scores.

**Limitations.** Our PI measures require training additional models to learn the probability distribution. Since estima-tors for PI measures are less common compared to their aggregate counterparts, our work, which clearly demonstrates their applications in confidence estimation, could motivate further research towards more accurate and efficient estimation of these measures. In addition, the PI measures are the optimal choice of explainability if we assume the probability-raising based causal model for the problem. Exploring more general structural causal models from an information-theoretic perspective in the context of explainability is an interesting avenue for future work.

## Impact Statement

The use of PI measures for confidence estimation in deep learning models serves as an additional safeguard against wrong predictions before deploying them in real-world applications. Our approach could be combined with explainability methods to ensure consistency of results. Caution should be taken when interpreting these results when they are used for critical decision-making, such as in being aware of any probabilistic biases in the data that may yield unfair outcomes. Finally, these PI measures can be used to explore other aspects of trustworthy AI including adversarial robustness and privacy.

## Acknowledgements

This research/project is supported by A*STAR, CISCO Systems (USA) Pte. Ltd and National University of Singapore under its Cisco-NUS Accelerated Digital Economy Corporate Laboratory (Award I21001E0002).

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

# Supplementary Materials

To allow ease of access and improve readability, we include the table of contents for our supplementary materials.

## Table of Contents

# A. Related Work & Information Measures Details

## A.1. Related Work

**Pointwise Mutual Information (PMI).** PMI compares the probability of two outcomes occurring together to what the probability would be if they are independent. It is commonly in natural language processing to measure the association between words (Church & Hanks, 1990; Turney, 2001; Su et al., 2006; Padmakumar & He, 2021). In this setting, $p(x)$ and $p(x, y)$ can be obtained by counting the occurrences and co-occurrences of words in the corpus. However, PMI can be sensitive to the size of the corpus and may not perform well with very rare words or when the data is sparse. Other variants of PMI have also been introduced, including positive PMI measure, which sets all the negative values to zero (Dagan et al., 1993), and normalized PMI measure, which scales the values to fall within the range $[-1, 1]$ (Bouma, 2009). PMI has found applications in a wide range of areas, including sentiment analysis (Ahanin & Ismail, 2022; Bandhakavi et al., 2017; Sintsova & Pu, 2016), community detection (Luo et al., 2021), response generation (Nandwani et al., 2023; Ren et al., 2023), truthful data acquisition (Zheng et al., 2024; Chen et al., 2020), and boundary detection (Isola et al., 2014). In this study, we use PMI to obtain both confidence scores and saliency maps for image classification tasks.

**Pointwise Sliced Mutual Information (PSI).** (Wongso et al., 2023b) introduced PSI as a measure for generating confidence scores and saliency maps for deep neural networks. For confidence scores, they compute the PSI between features of the penultimate layer of a neural network and predicted label for each sample and refer to this as the sample-wise PSI. For saliency maps, they compute the PSI between feature fiber of the last convolutional layer of a convolutional neural network and predicted label for each sample and refer to this as the fiber-wise PSI. In addition, they show that PSI, in contrast to PMI, exhibits sensitivity to sample-wise margin. Even though their findings demonstrate that PSI can produce sensible confidence scores and saliency maps the paper lacks a profound perspective and the essential quality assessment of PSI as a metric for model uncertainty and explainability. In this work, we provide a more comprehensive evaluation of PSI, comparing it to the other two pointwise measures, namely PMI and PVI, to determine the relevance between features and predicted labels. Additionally, we present a set of theoretical results that explore various properties of pointwise information measures, providing deeper insights into what they may represent.

**Pointwise $\mathcal{V}$-Information (PVI).** PVI was introduced to measure sample difficulty with respect to a given distribution (Ethayarajh et al., 2022). In their research, they investigate natural language inference tasks and observe that samples with high PVI are often predicted correctly, while those with low PVI are more likely to be predicted incorrectly. It is important to note that in their paper, they assess PVI in relation to the true label (also referred to as the gold label), making it a measure of sample difficulty rather than a measure of the network's confidence. They also show that PVI can be used to identify which subsets of each class are more difficult than others. PVI has recently been employed in a variety of NLP tasks (Lin et al., 2023; Lu et al., 2023; Prasad et al., 2023; Kulmizev & Nivre, 2023). In this study, we compute PVI to obtain both confidence scores and saliency maps for image classification tasks.

**Predictive Confidence.** The idea behind confidence estimation is closely connected to uncertainty quantification. Simply put, when we are more confident in a prediction made by a model, it means there is less uncertainty about that prediction. For a comprehensive survey/review on uncertainty quantification in deep learning, we refer the readers to (Gawlikowski et al., 2023; He & Jiang, 2023; Mena et al., 2022; Abdar et al., 2021). There are two common lines of work for evaluating predictive confidence: confidence ranking and confidence calibration. Works on confidence ranking focuses on ranking confidence scores such that the lower-ranked samples are more likely to misclassified while the higher-ranked samples are more likely to be correctly classified. Confidence ranking is useful in applications such as misclassification detection (Hendrycks & Gimpel, 2017; Jiang et al., 2018; Corbière et al., 2019; Jaeger et al., 2023), out-of-distribution detection (Hendrycks & Gimpel, 2017; DeVries & Taylor, 2018; Liang et al., 2018) and selective classification (Geifman & El-Yaniv, 2017; Feng et al., 2023; Galil et al., 2023). On the other hand, research on confidence calibration aims to provide confidence scores that accurately reflect the likelihood of a prediction being correct (Guo et al., 2017; Nixon et al., 2019; Tao et al., 2024). This requires the confidence scores to be probabilities within the range of 0 to 1.

## A.2. General Properties of Information Measures and Their Pointwise Variants

In this section, we describe some general properties of mutual information (MI), $\mathcal{V}$-information ($\mathcal{V}$I), and Sliced Mutual Information (SMI), and their pointwise variants.

A.2.1. GENERAL PROPERTIES OF MI AND PMI

We first restate the definition of MI and PMI from the main paper:

**Definition 1** (**MI and PMI**). *Let* $(X, Y) \sim P_{XY}$. *The MI between* $X$ *and* $Y$ *is:*

$$I(X;Y) := \mathbb{E}_{X,Y} \left[ \log \frac{P(X,Y)}{P(X)P(Y)} \right] \tag{12}$$

*Let* $(x, y) \sim (X, Y)$. *The PMI of an instance* $(x, y)$ *is:*

$$pmi(x;y) := \log \frac{p(x,y)}{p(x)p(y)} \tag{13}$$

MI satisfies the following properties:

1. **Non-negativity:** $I(X;Y) \geq 0$.
2. **Independence:** $I(X;Y) = 0$ iff $X$ and $Y$ are independent.
3. **Entropy decomposition:** $I(X;Y) = H(X) + H(Y) - H(X,Y) = H(X) - H(X|Y) = H(Y) - H(Y|X)$ where $H(\cdot)$ and $H(\cdot|\cdot)$ are the entropy and conditional entropy respectively.
4. **Chain rule:** $I(X,Y;Z) = I(X;Z) + I(Y;Z|X)$. More generally, for $X_1, \cdots, X_n$, we have $I(X_1, \cdots, X_n; Y) = I(X_1; Y) + \sum_{i=2}^{n} I(X_i; Y | X_1, \cdots, X_{i-1})$.

**Remark 4** (**Data processing inequality**). *MI also satisfies the data processing inequality which means that* $I(X;Y) \geq I(f(X);Y)$ *for any deterministic function* $f$. *This is in contrast to* $\mathcal{V}I$ *and SMI which can grow with more processing of the random variables.*

We list some properties of PMI as follows:

1. **Range:**
   - Continuous $X$ and $Y$: $-\infty \leq pmi(x;y) \leq \infty$.
   - Discrete $Y$: $-\infty \leq pmi(x;y) \leq -\log p(y)$.
   - Discrete $X$ and $Y$: $-\infty \leq pmi(x;y) \leq \min[-\log p(x), -\log p(y)]$.
2. **Independence:** If $X$ and $Y$ are independent, then $pmi(x;y) = 0 \; \forall (x,y) \in \mathcal{X} \times \mathcal{Y}$. Note that $pmi(x;y) = 0$ for a certain $(x, y) \sim P_{XY}$ does not imply $X$ and $Y$ are independent.
3. **Entropy decomposition:** $pmi(x;y) = h(x) + h(y) - h(x,y) = h(x) - h(x|y) = h(y) - h(y|x)$ where $h(\cdot) = -\log p(\cdot)$ is called the self-information.
4. **Chain rule:** $pmi(x, y; z) = pmi(x; z) + pmi(y; z|x)$.

A.2.2. GENERAL PROPERTIES OF SMI AND PSI

We first restate the definition of SMI and PSI from the main paper:

**Definition 2** (SMI and PSI). *Let* $(X, Y) \sim P_{XY} \in \mathcal{P}(\mathbb{R}^{d_x} \times \mathbb{R}^{d_y})$. *Let* $\Theta \sim \text{Unif}(\mathbb{S}^{d_x - 1})$ *and* $\Phi \sim \text{Unif}(\mathbb{S}^{d_y - 1})$ *be independent of each other and of* $(X, Y)$. *The SMI between* $X$ *and* $Y$ *is:*

$$SI(X;Y) := \mathbb{E}_{\substack{\theta \in \Theta \\ \phi \in \Phi}} [I(\theta^T X; \phi^T Y)] \tag{14}$$

*Let* $(x, y) \sim X, Y$. *The PSI of an instance* $(x, y)$ *is:*

$$psi(x;y) := \mathbb{E}_{\substack{\theta \in \Theta \\ \phi \in \Phi}} \left[ pmi(\theta^T x; \phi^T y) \right] \tag{15}$$

SMI shares many similar properties with MI (Goldfeld & Greenewald, 2021, Proposition 1), including:

1. **Non-negativity:** $SI(X;Y) \geq 0$.
2. **Independence:** $SI(X;Y) = 0$ iff $X$ and $Y$ are independent.
3. **Entropy decomposition:** $SI(X;Y) = SH(X) + SH(Y) - SH(X,Y) = SH(X) - SH(X|Y) = SH(Y) - SH(Y|X)$ where $SH(\cdot)$ and $SH(\cdot|\cdot)$ are the sliced entropy and conditional sliced entropy respectively.

4. **Chain rule:** $SI(X,Y;Z) = SI(X;Z) + SI(Y;Z|X)$. More generally, for $X_1, \cdots, X_n$, we have $SI(X_1, \cdots, X_n; Y) = SI(X_1; Y) + \sum_{i=2}^{n} SI(X_i; Y|X_1, \cdots, X_{i-1})$.

**Remark 5** (**SMI can grow with processing**). *We note that unlike MI and similar to VI, SMI can increase with more processing of the random variables, i.e., we can have $SI(f(X); Y) \geq SI(X; Y)$ for any deterministic function $f$. (Goldfeld & Greenewald, 2021) argued that this property is desirable in the context of machine learning, where it is more intuitive to think that processing input features yields representations that are more useful for inferring the labels.*

We list some properties of PSI as follows:

1. **Range:**
   - Continuous $X$ and $Y$: $-\infty \leq psi(x; y) \leq \infty$.
   - Discrete $Y$: $-\infty \leq psi(x; y) \leq -\log p(y)$.
2. **Independence:** If $X$ and $Y$ are independent, then $psi(x; y) = 0, \forall (x, y) \in \mathcal{X} \times \mathcal{Y}$. Note that $psi(x; y) = 0$ for a certain $(x, y) \sim P_{XY}$ does not imply $X$ and $Y$ are independent.
3. **Entropy decomposition:** $psi(x; y) = sh(x) + sh(y) - sh(x, y) = sh(x) - sh(x|y) = sh(y) - sh(y|x)$ where $sh(x) := -\mathbb{E}_{\theta \in \Theta} \log p(\theta^T x)$ and $sh(x|y) := -\mathbb{E}_{\theta \in \Theta, \phi \in \Phi} \log p(\theta^T x | \phi^T y)$ are the pointwise sliced entropy and pointwise conditional sliced entropy respectively.
4. **Chain rule:** $psi(x, y; z) = psi(x; z) + psi(y; z|x)$.

### A.2.3. GENERAL PROPERTIES OF $\mathcal{V}$I AND PVI

We first provide more detailed definitions for predictive family and conditional $\mathcal{V}$-entropy, and subsequently restate the definition of $\mathcal{V}$I and PVI from the main paper.

**Definition 4** (**Predictive Family**). *Let $\Omega = \{f : \mathcal{X} \cup \varnothing \to P(\mathcal{Y})\}$. The predictive family $\mathcal{V} \subseteq$ is defined such that it satisfies:*

$$\forall f \in \mathcal{V}, \forall P \in \text{range}(f), \ \exists f' \in \mathcal{V}, \ s.t. \ \forall x \in \mathcal{X}, f'[x] = P, f'[\varnothing] = P \tag{16}$$

**Remark 6** (**Optional ignorance**). *In words, a predictive family is a set of predictive models the agent can use, often limited by computational or statistical constraints. The additional condition $f'[x] = P, f'[\varnothing] = P$ is called the optional ignorance, which gives the agent an option to ignore the side information $x$ and still be able to predict get $P$. As shown in (Xu et al., 2020), this condition is necessary to obtain the desirable properties of $\mathcal{V}$I.*

**Definition 5** (**Predictive Conditional $\mathcal{V}$-entropy**). *Let $(X, Y) \sim P_{XY} \in \mathcal{P}(\mathcal{X} \times \mathcal{Y})$. We use $\varnothing$ to represent a null input that provides no information about $Y$. Given a predictive family $\mathcal{V}$, we can define the predictive conditional V-entropy as:*

$$H_{\mathcal{V}}(Y|X) := \inf_{f \in \mathcal{V}} \mathbb{E}_{X,Y}[-\log f[X](Y)] \tag{17}$$

$$H_{\mathcal{V}}(Y|\varnothing) := \inf_{f \in \mathcal{V}} \mathbb{E}_{Y}[-\log f[\varnothing](Y)] \tag{18}$$

*$H_{\mathcal{V}}(Y|\varnothing)$ is also called the $\mathcal{V}$-entropy and denoted as $H_{\mathcal{V}}(Y)$ for simplicity.*

**Definition 3** (**$\mathcal{V}$I and PVI**). *Let $\mathcal{V}, H_{\mathcal{V}}(Y), H_{\mathcal{V}}(Y|X)$ and $(X, Y)$ be defined as in Def. 4 and Def. 5. We are given predictive family. Then the $\mathcal{V}$I from $X$ to $Y$ is:*

$$I_{\mathcal{V}}(X \to Y) := H_{\mathcal{V}}(Y) - H_{\mathcal{V}}(Y|X) \tag{19}$$

*Let $g = \arg\min_{f \in \mathcal{V}} \mathbb{E}_{Y}[-\log f[\varnothing](Y)]$ and $g' = \arg\min_{f \in \mathcal{V}} \mathbb{E}_{X,Y}[-\log f[X](Y)]$. Given $(x, y) \sim (X, Y)$, the PVI from $x$ to $y$ is:*

$$pvi(x \to y) := -\log f[\varnothing](y) + \log f'[x](y) \tag{20}$$

$\mathcal{V}$I satisfies the following properties:

1. **Non-negativity:** $I_{\mathcal{V}}(X \to Y) \geq 0$.
2. **Independence:** $I_{\mathcal{V}}(X \to Y) = I_{\mathcal{V}}(Y \to X) = 0$ iff $X$ and $Y$ are independent.
3. **Entropy decomposition:** $I_{\mathcal{V}}(X \to Y) = H_{\mathcal{V}}(Y) - H_{\mathcal{V}}(Y|X)$.

**Remark 7** (**$\mathcal{V}$I can grow with processing**). *We note that unlike MI and similar to SMI, $\mathcal{V}$I can increase with more processing of the random variables, i.e., we can have $I_{\mathcal{V}}(f(X) \to Y) \geq I_{\mathcal{V}}(X \to Y)$ for any deterministic function $f$. (Xu et al., 2020) argued that this property is desirable in the context of machine learning, where it is more intuitive to think that processing input features yields more usable information about the label.*

**Remark 8** (**Asymmetry of $\mathcal{V}I$**). *We also note that unlike MI and SMI, $\mathcal{V}I$ is asymmetric in nature which is align with the intuition that sometimes, it is easier to predict $Y$ from $X$ than to predict $X$ from $Y$.*

We list some properties of PVI as follows:

1. **Range:**
   - Continuous $X$ and $Y$: $-\infty \leq pvi(x \to y) \leq \infty$
   - Discrete $Y$: $-\infty \leq pvi(x \to y) \leq -\log p(y)$ when $H_{\mathcal{V}}(Y) = H(Y)$. Note that this is true when $\mathcal{V}$ represents a function modelled by a neural network with trainable weights and biases.
2. **Independence:** If $X$ and $Y$ are independent, then we have $pvi(x \to y) = pvi(y \to x) = 0$. Note that $pvi(x \to y) = 0$ for some $(x, y) \sim P_{XY}$ does not imply that $X$ and $Y$ are independent.
3. **Entropy decomposition:** $pvi(x \to y) = h_{\mathcal{V}}(y) - h_{\mathcal{V}}(y|x)$, where $h_{\mathcal{V}}(y)$ is the pointwise $\mathcal{V}$-entropy of $y$ and $h_{\mathcal{V}}(y|x)$ is the pointwise conditional $\mathcal{V}$-entropy of $y$.

### A.3. Pointwise Information Estimators

In this section, we describe the estimators of PMI, PSI and PVI as well as provide the algorithms for each pointwise measure. We implemented these estimators in Python and use the Tensorflow library for neural networks.

#### A.3.1. PMI ESTIMATORS

In (Tsai et al., 2020), the authors proposed three different estimators for PMI: probabilistic classifier, density ratio fitting and variational Jensen-Shannon (JS) bound. All of these approaches estimate PMI using neural networks with distinct loss functions described below. We provide the pseudocode for the PMI estimator in Algorithm 1. Note that we presented the algorithm for any label $y$ but used predicted labels $\hat{y}$ in our experiments.

**Probabilistic Classifier (PC) Method.** In this approach, we assign class 1 to samples drawn from the joint density ($c = 1$ for $(x, y) \sim P_{XY}$) and class 0 to samples drawn from the product of marginal densities ($c = 0$ for $(x, y) \sim P_X P_Y$). Thus, we can rewrite the density ratio as:

$$\frac{p(x, y)}{p(x)p(y)} = \frac{p(x, y|c = 1)}{p(x, y|c = 0)} = \frac{p(c = 0)}{p(c = 1)} \frac{p(c = 1|x, y)}{p(c = 0|x, y)} \tag{21}$$

where we have used Bayes' Theorem for the second equality. Furthermore, we can approximate the ratio of class probabilities by the ratio of the sample size:

$$\frac{\hat{p}(c = 0)}{\hat{p}(c = 1)} = \frac{n_{P_X P_Y}/n_{P_X P_Y} + n_{P_{XY}}}{n_{P_{XY}}/n_{P_X P_Y} + n_{P_{XY}}} = \frac{n_{P_X P_Y}}{n_{P_{XY}}} \tag{22}$$

To approximate the class-posterior probabilities, we use a neural network $f$ parameterized by $\theta$ with the following binary cross-entropy loss function:

$$L_{\text{PC}}(\theta) = -\mathbb{E}_{P_{XY}}[\log f_\theta(c = 1|(x, y))] - \mathbb{E}_{P_X P_Y}[\log(1 - f_\theta(c = 1|(x, y)))] \tag{23}$$

For $b$ mini-batch samples, we can write the loss function as:

$$\hat{L}_{\text{PC}}(\theta) = -\frac{1}{b}\sum_{i=1}^{b}[\log f_\theta(c = 1|(x^{(i)}, y^{(i)}))] - \frac{1}{b}\sum_{i=1}^{b}[\log(1 - f_\theta(c = 1|(x^{(i)}, \bar{y}^{(i)})))] \tag{24}$$

where $(x, y) \sim P_{XY}$ and $\bar{y} \sim P_Y$.

(Tsai et al., 2020) also showed that when $\Theta$ is large enough, the optimal $f_\theta(c|x, y) = p(c|x, y)$.

**Density Ratio Fitting (DRF) Method.** This approach seeks to minimize the expected least-square difference between the true density ratio and the density ratio estimated using a neural network $f$ parameterized by $\theta$. By letting $r(x, y) = p(x, y)/p(x)p(y)$, the objective function can be written as:

$$\inf_{\theta \in \Theta} \mathbb{E}_{P_X P_Y}[(r(x, y) - f_\theta(x, y))^2] \Leftrightarrow \sup_{\theta \in \Theta} \mathbb{E}_{P_{XY}}[f_\theta(x, y)] - \frac{1}{2}\mathbb{E}_{P_X P_Y}[f_\theta^2(x, y)] \tag{25}$$

Thus, the loss function is:

$$L_{\text{DRF}}(\theta) = -\mathbb{E}_{P_{XY}}[f_\theta(x,y)] + \frac{1}{2}\mathbb{E}_{P_X P_Y}[f_\theta^2(x,y)] \tag{26}$$

For $b$ mini-batch samples, we can write the loss function as:

$$\hat{L}_{\text{DRF}}(\theta) = -\frac{1}{b}\sum_{i=1}^{b}[f_\theta(x^{(i)}, y^{(i)})] + \frac{1}{2b}\sum_{i=1}^{b}[f_\theta^2(x^{(i)}, \bar{y}^{(i)})] \tag{27}$$

where $(x,y) \sim P_{XY}$ and $\bar{y} \sim P_Y$.

(Tsai et al., 2020) also showed that when $\Theta$ is large enough, the optimal $f_\theta(x,y) = r(x,y) = \frac{p(x,y)}{p(x)p(y)}$.

**Variational Jensen-Shannon (JS) Bound Method.** This approach relies on the variational form of MI, and in particular the Jensen-Shannon divergence between $P_{XY}$ and $P_X P_Y$ (Poole et al., 2019). The Jensen-Shannon variational estimator is found to be more stable than the other proposed variational lower bounds. Similar to the density ratio fitting method, the density ratio is estimated using a neural network $f$ parameterized by $\theta$. The loss function can be written as:

$$L_{\text{JS}}(\theta) = \mathbb{E}_{P_{XY}}[\text{softplus}(-\log f_\theta(x,y))] + \mathbb{E}_{P_X P_Y}[\text{softplus}(\log f_\theta(x,y))] \tag{28}$$

where $\text{softplus}(x) = \log(1 + \exp(x))$.

For $b$ mini-batch samples, we can write the loss function as:

$$\hat{L}_{\text{JS}}(\theta) = \frac{1}{b}\sum_{i=1}^{b}[\text{softplus}(-\log f_\theta(x^{(i)}, y^{(i)}))] + \frac{1}{b}\sum_{i=1}^{b}[\text{softplus}(\log f_\theta(x^{(i)}, \bar{y}^{(i)}))] \tag{29}$$

where $(x,y) \sim P_{XY}$ and $\bar{y} \sim P_Y$.

(Tsai et al., 2020) also showed that when $\Theta$ is large enough, the optimal $f_\theta(x,y) = \frac{p(x,y)}{p(x)p(y)}$.

**Critic Model Architectures.** The neural networks used to estimate PMI are also commonly referred to as critic models. In the literature, there are two common structures for the critic models: separable and joint. They primarily differ in how $x$ and $y$ are considered in the neural network training. In separable critic design, $x$ and $y$ are being passed to two separate neural networks: $h(x)$ and $g(y)$. The final model then computes the dot product between the outputs of the two neural networks: $f(x,y) = h(x)^T g(y)$. In joint critic design, $x$ and $y$ are concatenated and fed as input to one neural network. In Appendix D.2.1, we compare the performance of the different critic architectures. We represent the neural network using a multi-layer perceptron, consisting of one hidden layer with 512 units and ReLU activation function. For separable critic, the outputs of neural network $h(x)$ and $g(y)$ have dimensions of 128, while for joint critic, the output has a dimension of 1. They are trained with Adam optimizer with learning rate of 0.001 for a maximum of 200 epochs. We employ early stopping if the maximum MI on the validation dataset fails to improve after 10 epochs. For the final PMI model, we use the one that yields the highest MI on the validation dataset.

**Note on Implementation.** We followed the implementation by (Tsai et al., 2020), adapting their original PyTorch code to Tensorflow. In their implementation, rather than shuffling samples from the joint distribution to obtain samples drawn from product of marginal densities, they manipulate the output of the critic model to have a shape of $b \times b$, where $b$ represents the batch size. To achieve this for the joint critic, they introduce a new axis and replicate the input $b$ times along that axis. Consequently, the diagonal elements naturally correspond to samples drawn from the joint density, while the off-diagonal elements represent the product of marginal densities. In this setup, there are $b$ mini-batch joint samples and $b^2 - b$ mini-batch marginal samples. When using the PC method, there is an additional term $n_{P_X P_Y}/n_{P_{XY}}$ which computes the ratio of samples from the different distributions. In line with this implementation, given the unequal number of samples from the different distributions, an additional term of $\log[(b^2 - b)/b] = \log[b-1]$ must be included in the final PMI estimation.

---

**Algorithm 1** PMI Estimator

---

**Require:** $(X^n, Y^n) \sim P_{XY} \in \mathcal{P}(\mathbb{R}^{d_x} \times \mathbb{R})$ where $Y \in \{1, .., k\}$, a chosen pair of sample $(x, y) \sim (X^n, Y^n)$, critic model $f$, and $E$ number of epochs to train the critic model.

$\theta \leftarrow$ initialize parameters of $f$

$e \leftarrow 0$

**while** $e < E$ **do**

    Draw $b$ mini-batch samples from the joint density: $(x^{(1)}, y^{(1)}), \cdots, (x^{(b)}, y^{(b)}) \sim (X^n, Y^n)$

    Draw $b$ mini-batch samples from the marginal density[1]: $\bar{y}^{(1)}, \cdots, \bar{y}^{(b)} \sim P_Y$

    Compute the loss function $L(\theta)$ on the mini-batch samples:

    (Eq. (24) for PC, Eq. (27) for DRF, or Eq. (29) for variational JS bound)

    Update the critic model parameters $\theta$ based on $L(\theta)$

    $e \leftarrow e + 1$

**end while**

**return** $\widehat{pmi}(x; y) \leftarrow f(x, y)$ for PC and variational JS bound or

    $\widehat{pmi}(x; y) \leftarrow \log f(x, y)$ for DRF

---

For all our experiments, we choose the variational JS bound (with separable critic) as the default PMI estimator as we show in Appendix D.2.1, it yields the best performance.

### A.3.2. PSI ESTIMATORS

We followed the implementation by (Wongso et al., 2023b) and considered an additional method: binning. We provide the pseudocode for the PSI estimator for the binning method in Algorithm 2 and for the Gaussian method in Algorithm 3. Note that we presented the algorithm for any label $y$ but used predicted labels $\hat{y}$ in our experiments. For our problems, we only project $X$ since $Y$ is discrete. For both methods, we clip the probability to a minimum of 1e-5 to prevent division by zero. We did not consider kernel density and neural network estimation in this work due to its high computational cost, which is not practical for confidence estimation.

---

**Algorithm 2** PSI Estimator (Binning Method)

---

**Require:** $(X^n, Y^n) \sim P_{XY} \in \mathcal{P}(\mathbb{R}^{d_x} \times \mathbb{R})$ where $Y \in \{1, .., c\}$ ($c$ classes), a chosen pair of sample $(x, y) \sim (X^n, Y^n)$, a chosen number of slices (projections) $m$, and a chosen number of bins $n_{\text{bins}}$.

Initialize $\theta_i$ by sampling uniformly on the sphere $\mathbb{S}^{d_x - 1}$ for $i = 1, \ldots, m$.

**for** $i = 1$ to $m$ **do**

    Compute $\theta_i^T X$ and discretize it into $n_{\text{bins}}$ bins using training features $X^n$

    Compute joint counts of binned $\theta_i^T X$ and $Y$

    Normalize joint counts to obtain joint probabilities $P(\theta_i^T X, Y)$

    Compute marginal probabilities $P(\theta_i^T X)$ and $P(Y)$

    Find the bin index of $\theta_i^T x$ in the discretized $\theta_i^T X$ for the given sample $x$

    Retrieve $p(\theta_i^T x, y)$ from $P(\theta_i^T X, Y)$

    Retrieve $p(\theta_i^T x)$ from $P(\theta_i^T X)$

    Retrieve the marginal probability $p(y)$ from $P(Y)$

    Compute the term: $pmi_i(x; y) \leftarrow \log \frac{p(\theta_i^T x, y)}{p(\theta_i^T x) p(y)}$

**end for**

**return** $\widehat{psi}(x; y) \leftarrow \frac{1}{m} \sum_{i=1}^{m} pmi_i(x, y)$

---

[1]This can be done by shuffling the samples from the joint distribution along the batch axis.

---

**Algorithm 3** PSI Estimator (Gaussian Method)

---

**Require:** $(X^n, Y^n) \sim P_{XY} \in \mathcal{P}(\mathbb{R}^{d_x} \times \mathbb{R})$ where $Y \in \{1, \ldots, c\}$ (with $c$ classes), a chosen pair of sample $(x, y) \sim (X^n, Y^n)$, a chosen number of slices (projections) $m$, and a chosen number of bins $n_{\text{bins}}$.

Initialize $\theta_i$ by sampling uniformly on the sphere $\mathbb{S}^{d_x - 1}$ for $i = 1, \ldots, m$.

**for** $i = 1$ to $m$ **do**
    **for** $j = 1$ to $c$ **do**
        Find $\mu_{ij}, \sigma_{ij}^2$ with $P(\theta_i^T X | Y = j) \sim \mathcal{N}(\mu_{ij}, \sigma_{ij}^2)$.
    **end for**
**end for**
**for** $i = 1$ to $m$ **do**
    Compute $\theta_i^T x$ for the given sample $x$
    Retrieve $p(\theta_i^T x | y)$ from $P(\theta_i^T X | Y = y) \sim \mathcal{N}(\mu_{iy}, \sigma_{iy}^2)$
    Compute $p(\theta_i^T x) = \sum_{j=1}^c p(\theta_i^T x | y = j) p(y = j)$
    Compute the term: $pmi_i(x, y) \leftarrow \log \frac{p(\theta_i^T x | y)}{p(\theta_i^T x)}$
**end for**
**return** $\widehat{psi}(x; y) \leftarrow \frac{1}{m} \sum_{i=1}^m pmi_i(x, y)$

---

**Binning.** For each projection $i$, we bin $\theta_i^T X$ into $n_{\text{bins}}$. To compute the PSI, we estimate $P(\theta_i^T X, Y)$, $P(\theta_i^T X)$, and $P(Y)$ from the binned data. For a given sample, we can then find the $p(\theta_i^T x, y)$, $p(\theta_i^T x)$, and $p(y)$. The PSI is then given by:

$$\widehat{psi}(x; y) \leftarrow \frac{1}{m} \sum_{i=1}^m \log \frac{p(\theta_i^T x, y)}{p(\theta_i^T x) p(y)} \tag{30}$$

**Gaussian.** We assume that $\theta^T X$ for each class follows a Gaussian distribution. For each projection $i$ and for each class $j$, we estimate the mean ($\mu_{ij}$) and standard deviation ($\sigma_{ij}$). To compute the PSI, we estimate $P(\theta_i^T X | Y)$ and $P(\theta_i^T X)$ from $\mu_{ij}$ and $\sigma_{ij}$. For a given sample, we can then find the $p(\theta_i^T x | y)$ and $p(\theta_i^T x)$. The PSI is then given by:

$$\widehat{psi}(x; y) \leftarrow \frac{1}{m} \sum_{i=1}^m \log \frac{p(\theta_i^T x | y)}{p(\theta_i^T x)} \tag{31}$$

We choose the **Gaussian method (with 500 projections)** as the default estimator as we show in Appendix D.2.2, it consistently yields good performance.

### A.3.3. PVI ESTIMATORS

We followed the implementation by (Ethayarajh et al., 2022), adapting their original PyTorch code to Tensorflow. We provide the pseudocode for the PVI estimator in Algorithm 4. Note that we presented the algorithm for any label $y$ but used predicted labels $\hat{y}$ in our experiments. To estimate PVI, we are required to train two neural networks to obtain $f$ (for null inputs) and $f'$ (for training inputs). The null inputs can be obtained by setting the values of the input features to zero. Below we describe several methods we experiment to estimate the PVI.

---

**Algorithm 4** PVI Estimator

---

**Require:** $(X^n, Y^n)$ i.i.d. sampled according to $P_{XY} \in \mathcal{P}(\mathbb{R}^{d_x} \times \mathbb{R})$ where $Y \in \{1, .., k\}$, a chosen pair of sample $(x, y) \sim (X^n, Y^n)$, and a model $\mathcal{V}$.

$f' \leftarrow$ train $\mathcal{V}$ on $(X^n, Y^n)$
$\varnothing \leftarrow$ null input (array of zeros with the same shape as $X^n$)
$f \leftarrow$ train $\mathcal{V}$ on $(\varnothing, Y^n)$
**return** $\widehat{pvi}(x \to y) \leftarrow -\log f[\varnothing](y) + \log f'[x](y)$

---

[1]A uniform sample from $\mathbb{S}^{d-1}$ can be found by sampling a vector $Z$ from a $d$-dimensional isotropic Gaussian and forming $Z / ||Z||_2$.

**No Training.** To obtain $f'$, we use the model that has already been trained on the dataset. To obtain $f$, we train the (untrained) model on null inputs.

**Training from Scratch.** To obtain $f'$, we train another model (with different initialization but same architecture) on the training data. To obtain $f$, we train the (untrained) model on null inputs. In practice, instead of training a new model, we can use the model from the different run.

**Training MLP Penultimate.** To obtain $f'$, we use the penultimate layer features as input $x$ rather than the original inputs. We train a one-hidden-layer MLP with 512 units on $x$ to obtain $f'$. To obtain $f$ we train the untrained MLP model on null inputs with the same dimension as the penultimate layer features.

We choose the **Training from Scratch** method as the default estimator as we show in Appendix D.2.3, it consistently yields good performance. In addition, when computing the PVI, we can choose to first calibrate the probabilities with a simple temperature scaling. As we see in Appendix D.2.3, this improves the performance.

# B. Theoretical Analysis & Proofs

## B.1. Invariance Properties

Proof of Proposition 1

**Proposition 1** (**Invariance to shift, scale, and rotation**). *Let $\mathcal{T}x = \alpha \boldsymbol{R}x + \boldsymbol{p}$, where $\boldsymbol{p} \in \mathbb{R}^{d_x}$ represents the extent to which the distribution is shifted, $\alpha \in \mathbb{R}$ represents how much the distribution is scaled, and $\boldsymbol{R} \sim \mathbb{R}^{d_x \times d_x}$ is a rotation matrix such that $\boldsymbol{R}\boldsymbol{R}^T = I$ and $\det(\boldsymbol{R}) = 1$, where $I$ is the identity matrix and $\det$ represents the determinant operator. Then we have:*

$$pmi_P(x, y) = pmi_{\mathcal{T}P}(\alpha \boldsymbol{R}x + \boldsymbol{p}, y)$$
$$psi_P(x, y) = psi_{\mathcal{T}P}(\alpha \boldsymbol{R}x + \boldsymbol{p}, y)$$
$$pvi_P(x, y) = pvi_{\mathcal{T}P}(\alpha \boldsymbol{R}x + \boldsymbol{p}, y)$$

*Proof.* For simplicity of notation, we denote the probability distribution in the original domain by $P$ and the distribution in the transformed domain by $P_{\mathcal{T}}$.

For PMI, we have:

$$pmi_{\mathcal{T}P}(\alpha \boldsymbol{R}x + \boldsymbol{p}, y) = \log \frac{P_{\mathcal{T}}(\alpha \boldsymbol{R}x + \boldsymbol{p}|y)}{P_{\mathcal{T}}(\alpha \boldsymbol{R}x + \boldsymbol{p})} = \log \frac{p(x|y)/\det(\alpha \boldsymbol{R})}{p(x)/\det(\alpha \boldsymbol{R})} = \log \frac{p(x|y)}{p(x)} = pmi_P(x, y),$$

where $\det$ denotes the determinant operator.

For PSI, we first note that

$$psi_{\mathcal{T}P}(\alpha \boldsymbol{R}x + \boldsymbol{p}, y) = \mathbb{E}_{\theta} \left[ \log \frac{P_{\mathcal{T}}(\theta^T(\alpha \boldsymbol{R}x + \boldsymbol{p})|y)}{P_{\mathcal{T}}(\theta^T(\alpha \boldsymbol{R}x + \boldsymbol{p}))} \right] = \mathbb{E}_{\theta} \left[ \log \frac{P_{\mathcal{T}}(\theta'^T(\alpha x) + \theta^T \boldsymbol{p})|y)}{P_{\mathcal{T}}(\theta'^T(\alpha x) + \theta^T \boldsymbol{p}))} \right],$$

where $\theta' = \theta \boldsymbol{R}$. Notice that $\theta'$ will have a uniform distribution over the sphere, similar to $\theta$, because $\boldsymbol{R}$ is a rotation matrix. We also apply the fact that $P_{\mathcal{T}}(\alpha x + \boldsymbol{p}) = p(x)/\alpha$ to the numerator and denominator. This ultimately yields:

$$psi_{\mathcal{T}P}(\alpha \boldsymbol{R}x + \boldsymbol{p}, y) = \mathbb{E}_{\theta'} \left[ \log \frac{p(\theta'^T x)|y)}{p(\theta'^T x)} \right] = \mathbb{E}_{\theta} \left[ \log \frac{p(\theta'^T x)|y)}{p(\theta^T x)} \right] = psi_P(x, y).$$

For PVI, we first note that the first term of PVI, $-\log f[\varnothing](y)$, will remain unchanged as it only depends on $y$, and $f$ depends on the distribution of $y$, both of which do not change with $\mathcal{T}$ as it is a one-to-one transformation. Then, for the conditional entropy term, let

$$f' = \arg\min_{f \in \mathcal{V}} \mathbb{E}_P[-\log f[X](Y)].$$

As $f'$ is a fully connected neural network with weights and biases, let $W$ and $b$ represent the weights and biases of the first layer, respectively. When the distribution of $x$ changes in response to $\mathcal{T}$, let

$$g' = \arg\min_{f \in \mathcal{V}} \mathbb{E}_{P_{\mathcal{T}}}[-\log f[X](Y)].$$

Note that $g'$'s first layer weights $W'$ and biases $b'$ will be such that $W'^T(\alpha \boldsymbol{R}x + \boldsymbol{p}) + b' = W^T x + b$. We will simply have $W'^T \alpha \boldsymbol{R} = W$ and $b' = b - W'^T \boldsymbol{p}$. Therefore, $g'[\mathcal{T}x](y) = f'[x](y)$. The search space for the $\arg\min$ is the same in both cases, as the transformation is linear. We have that $W^T(\mathcal{T}X) = W'^T X$ such that the weights $W'$ and $W$ have a one-to-one correspondence (as $\mathcal{T}$ is invertible). Since $\log g'[\mathcal{T}x](y) = \log f'[x](y)$, we have the result:

$$pvi_{\mathcal{T}P}(\alpha \boldsymbol{R}x + \boldsymbol{p}, y) = pvi_P(x, y).$$

$\square$

Proof of Proposition 2

**Proposition 2** (**Invariance to general linear transformations**). *Let $\mathcal{T}x = \boldsymbol{M}x$, where $\boldsymbol{M} \sim \mathbb{R}^{d_x \times d_x}$ is an invertible matrix. Then we have,*

$$pmi_P(x, y) = pmi_{\mathcal{T}P}(\boldsymbol{M}x, y)$$
$$pvi_P(x, y) = pvi_{\mathcal{T}P}(\boldsymbol{M}x, y)$$

*Proof.* For PMI, as $\boldsymbol{M}$ is invertible, we have:

$$pmi_{\mathcal{T}P}(\boldsymbol{M}x, y) = \log \frac{P_{\mathcal{T}}(\boldsymbol{M}x|y)}{P_{\mathcal{T}}(\boldsymbol{M}x)} = \log \frac{p(x|y)/det(\boldsymbol{M})}{p(x)/det(\boldsymbol{M})} = \log \frac{p(x|y)}{p(x)} = pmi_P(x, y).$$

where $det$ denotes the determinant operator.

For PVI, we follow the same reasoning as the previous proof. Same as before, let

$$f' = \arg\min_{f \in \mathcal{V}} \mathbb{E}_P[-\log f[X](Y)] \qquad g' = \arg\min_{f \in \mathcal{V}} \mathbb{E}_{P_{\mathcal{T}}}[-\log f[X](Y)].$$

Let $W$ and $b$ be the weights and biases of $f'$, and let $W'$ and $b'$ be the weights and biases of $g'$. Then, we have, $W'^T = W\boldsymbol{M}^{-1}$ and $b' = b$. As $\boldsymbol{M}$ is invertible, this implies that $g'[\mathcal{T}x](y) = f'[x](y)$, which yields the result:

$$pvi_{\mathcal{T}P}(\boldsymbol{M}x, y) = pvi_P(x, y).$$

$\square$

Proof of Proposition 3

**Proposition 3** (**Invariance to homeomorphic transformations**). *Let $\mathcal{T}x = f(x)$, where $f : \mathbb{R}^{d_x} \to \mathbb{R}^{d_x}$ represents any continuous and invertible transformation (i.e. a homeomorphism). Then we have,*

$$pmi_P(x, y) = pmi_{\mathcal{T}P}(f(x), y)$$

*Proof.* For smooth and invertible maps, it is known that the probability density function $P_{\mathcal{T}}(f(x)) = P(x)/J_X$, where $J_X = |\frac{\partial x}{\partial f(x)}|$ is a scalar that only depends on x. The same rule would apply to conditional distributions $p(x|y)$ as well. Thus we have:

$$pmi_{\mathcal{T}P}(f(x), y) = \log \frac{P_{\mathcal{T}}(f(x)|y)}{P_{\mathcal{T}}(f(x))} = \log \frac{p(x|y)/J_X}{p(x)/J_X} = \log \frac{p(x|y)}{p(x)} = pmi_P(x, y)$$

$\square$

**Remark 9** (**On PMI and invariance**). *Note that the above property for PMI also implies invariance to general linear transformations, which is the extension to Proposition 2. What these results mainly indicate is that out of the three metrics, PMI has the most structure-preserving property, followed by PVI and then PSI. This makes sense as PMI is the most general and only depends on the distribution and does not rely on anything else. Note that MI is invariant to homeomorphisms as well, but the invariance property for PMI is stronger as it states that the aggregate invariance for MI can be mirrored at the pointwise level.*

**Remark 10** (**On PSI and invariance**). *Note that PSI need not be invariant to both general linear transformations and homeomorphisms. To see why, just consider a simple case where $\mathcal{T}$ represents general linear transformations which scale each dimension of the input separately. Then, a sphere in the original domain of the distribution $P$ gets transformed into an ellipse in the domain of the distribution $\mathcal{T}P$. As PSI uses a uniform distribution over all projections over the sphere, we cannot say with certainty that the PSI in the new domain of $\mathcal{T}P$ will be preserved, because it will prefer some directions more over others. To see this, consider a specific case of $\mathcal{T}$ where one of the dimensions is scaled significantly more than the rest, thereby resulting in a ellipse that is very flat. In that case, most projections will contain more of that dimension, and we cannot say that PSI will be surely preserved.*

## B.2. Convergence Rates

Proof of Proposition 4

**Proposition 4** (**PMI convergence rate**). *Let $P(X)$ be $\alpha$-Holder continuous and let $(x, y) \sim P_{XY}$ where $X \in \mathbb{R}^{d_x}$ and $Y \in \{0, 1\}$. Let $\widehat{pmi}_n$ represent the KDE estimate of PMI using $n$ samples. Assuming $\min\{P(Y = 0), P(Y = 1)\} \neq 0$ when the probabilities are estimated on the training data, for large enough $n$, we can bound the estimation error as*

$$\left| pmi(x; y) - \widehat{pmi}_n \right| \leq \mathcal{O}\left( \frac{n^{-\alpha/(2\alpha + d_x)}}{\min\{p(x), p(x|y)\}} \right) \tag{32}$$

*Proof.* For simplicity of notation, we represent all estimated probability terms by $\widehat{P}_n$, where $n$ represents the number of samples used to estimate the term. We have:

$$\left| pmi(x; y) - \widehat{pmi}_n \right| = \left| \log \frac{P(x|y)}{P(x)} - \log \frac{\widehat{P}_n(x|y)}{\widehat{P}_n(x)} \right| \tag{33}$$

$$\leq \left| \log P(x|y) - \log \widehat{P}_n(x|y) \right| + \left| \log P(x) - \log \widehat{P}_n(x) \right| \tag{34}$$

$$\leq \sup_{x \in \mathbb{R}^d} \left| \log P(x|y) - \log \widehat{P}_n(x|y) \right| + \sup_{x \in \mathbb{R}^d} \left| \log P(x) - \log \widehat{P}_n(x) \right| \tag{35}$$

Now, from (Jiang, 2017), we have the uniform bounds on $\widehat{P}_n(x)$ in Theorem 2, which yields:

$$\sup_{x \in \mathbb{R}^d} |P(x) - \widehat{P}_n(x)| \leq \mathcal{O}\left( n^{\frac{-\alpha}{2\alpha + d}} \right) \tag{36}$$

Note that we can apply these bounds to $P(x|y)$ as well, and in that case the sample complexity changes from $n$ to $\min\{P(Y = 0), P(Y = 1)\} \times n$, because the number of samples that now controls the convergence rate is reduced as these are class-wise distributions. In the case when $\min\{P(Y = 0), P(Y = 1)\} \neq 0$, note that this keeps the final convergence order unchanged, as it adds a fixed multiplicative term. As $\min\{P(Y = 0), P(Y = 1)\} \neq 0$ is assumed in the problem, we can directly apply the results from (Jiang, 2017) for $P(x|y)$ as well.

With this, we use the expansion of $\log$ to write:

$$\left| \log P(x) - \log \widehat{P}_n(x) \right| = \left| \log \frac{\widehat{P}_n(x)}{P(x)} \right| = \left| \log \left( 1 + \frac{\widehat{P}_n(x) - P(x)}{P(x)} \right) \right| \tag{37}$$

$$\leq \left| \frac{\widehat{P}_n(x) - P(x)}{P(x)} \right| - \frac{1}{2} \left| \frac{\widehat{P}_n(x) - P(x)}{P(x)} \right|^2 + \frac{1}{3} \left| \frac{\widehat{P}_n(x) - P(x)}{P(x)} \right|^3 - \dots \tag{38}$$

$$\leq \left| \frac{\widehat{P}_n(x) - P(x)}{P(x)} \right| - \frac{1}{2} \left| \frac{\widehat{P}_n(x) - P(x)}{P(x)} \right|^2 + \frac{1}{3} \left| \frac{\widehat{P}_n(x) - P(x)}{P(x)} \right|^3 - \dots \tag{39}$$

$$\leq \mathcal{O}\left( \frac{n^{-\alpha/(2\alpha + d)}}{p(x)} \right) - \frac{1}{2} \mathcal{O}\left( \frac{n^{-2\alpha/(2\alpha + d)}}{p(x)^2} \right) + \frac{1}{3} \mathcal{O}\left( \frac{n^{-3\alpha/(2\alpha + d)}}{p(x)^3} \right) - \dots \tag{40}$$

$$\leq \mathcal{O}\left( \frac{n^{-\alpha/(2\alpha + d)}}{p(x)} \right) \tag{41}$$

Here we assume that $n$ is large enough, such that the rest of the terms are insignificant compared to the first term. Combining this with (35), we have the result.

$\square$

Proof of Proposition 5

**Proposition 5** (**PSI convergence rate**). *Let $P(\theta^T X)$ be $\alpha$-Holder continuous for all $\theta$ and let $(x, y) \sim P_{XY}$ where $X \in \mathbb{R}^{d_x}$ and $Y \in \{0, 1\}$. Let $\widehat{psi}_{n,m}$ represent the KDE estimate of PSI using $n$ samples and $m$ projections. Furthermore, let $\min_\theta pmi(\theta^T x; y) \geq \rho$. Assuming $\min\{P(Y = 0), P(Y = 1)\} \neq 0$ when the probabilities are estimated on the training data, for large enough $n$, we can bound the estimation error as*

$$\mathbb{E}_{X,Y}\left[\left|psi(x; y) - \widehat{psi}_{n,m}\right|\right] \leq \frac{1 - \rho}{2\sqrt{m}} + \mathcal{O}\left(\frac{n^{-\alpha/(2\alpha+1)}}{\min_\theta \min\{p(\theta^T x), p(\theta^T x|y)\}}\right) \tag{42}$$

*Proof.* We apply the triangle inequality, similar to (Goldfeld & Greenewald, 2021) (Appendix A.4), to obtain:

$$\left|psi(x; y) - \widehat{psi}_{n,m}\right| \leq \left|psi(x; y) - \frac{1}{m}\sum_{i=1}^{m} pmi(\theta^T x; y)\right| + \left|\sum_{i=1}^{m} pmi(\theta^T x; y) - \widehat{psi}_{n,m}\right| \tag{43}$$

$$\tag{44}$$

Now, as $\theta_i$ are i.i.d, and $PSI(x; y)$ is essentially equal to $\sum_{i=1}^{m} PMI(\theta^T x; y)$ as $m \to \infty$, we can use a variance based bound to obtain:

$$\mathbb{E}\left[\left|psi(x; y) - \frac{1}{m}\sum_{i=1}^{m} pmi(\theta^T x; y)\right|\right] \leq \sqrt{\frac{var(pmi(\theta^T x; y))}{m}} \leq \frac{1 - \rho}{2\sqrt{m}} \tag{45}$$

Next, we have that:

$$\mathbb{E}\left[\left|\sum_{i=1}^{m} pmi(\theta^T x; y) - \widehat{psi}_{n,m}\right|\right] \leq \sum_{i=1}^{m} \mathbb{E}\left[\left|pmi(\theta^T x; y) - \widehat{pmi}_n(\theta^T x; y)\right|\right] \tag{46}$$

$$\leq \sup_\theta \mathbb{E}\left[\left|pmi(\theta^T x; y) - \widehat{pmi}_n(\theta^T x; y)\right|\right] \tag{47}$$

We then apply the previous result (Proposition 6), to obtain:

$$\sup_\theta \mathbb{E}\left[\left|pmi(\theta^T x; y) - \widehat{pmi}_n(\theta^T x; y)\right|\right] \leq \mathcal{O}\left(\frac{n^{-\alpha/(2\alpha+d)}}{\min_\theta \min\{p(\theta^T x), p(\theta^T x|y)\}}\right) \tag{48}$$

$$= \mathcal{O}\left(\frac{n^{-\alpha/(2\alpha+1)}}{p_{min}}\right) \tag{49}$$

This completes the proof. $\square$

**Remark 11** (**On KDE-based PSI estimator**). *The above result provides convergence bounds for the KDE-based PSI estimator, providing guarantees as a function of the number of projections $m$ and the number of datapoints $n$. The result makes use of the uniform convergence bounds for the KDE-based density estimator provided in (Jiang, 2017) The convergence rates would be tighter for larger values of $\alpha$, and larger values of $p_{min}$. Thus, we note that the convergence can be slower for datapoints $x$ for which $p_{min}$ is small, which will be true for datapoints in the edge of the distribution $P(X)$.*

**Remark 12** (**On PSI's faster convergence over PMI**). *Note that when $\alpha \to \infty$, we obtain the same rate of convergence as SMI itself, which is $O(m^{-1/2} + n^{-1/2})$ (Goldfeld & Greenewald, 2021). Also, note that PSI converges at a much faster rate than PMI, especially when considering data of large dimensionality $d$, as the convergence rate for PMI will be $O(n^{-\alpha/(2\alpha+d)})$, which follows from Theorem 2 and Remark 8 in (Jiang, 2017).*

Proof of Theorem 1

**Theorem 1** (**PVI convergence rate**). *Given $(x, y) \sim P_{XY}$ where $X \in \mathbb{R}^{d_x}$ and $Y \in \{0, 1\}$, we assume that $P(Y = 0) = P(Y = 1) = 0.5$. Assume $\mathcal{V}$ represents the set of all possible functions modelled by a neural network having some fixed architecture. Assume $\forall f \in \mathcal{V}$, $\log f[x](y) \in [-B, B]$. Also, let $f^* = \arg\min_{f \in \mathcal{V}} \mathbb{E}_{X,Y}[-\log f[X](Y)]$ represent the ground truth function for estimating conditional $\mathcal{V}$-entropy, and $\widehat{f}$ represent the trained function given $n$ datapoints $(x_1, y_1), ..., (x_n, y_n)$ sampled from $P_{XY}^n$. Let $M = \max\{var(f^*[x](y)), var(\widehat{f}[x](y))\}$ where $var$ denotes the variance.*

*Let $\widehat{pvi}_n$ represent the PVI estimated using this neural network with $n$ samples. Then, for any $\delta \in (0, 0.5)$, with probability $p \geq 1 - 2\delta$, we have*

$$\mathbb{E}_{X,Y}\left[\left|pvi(x \to y) - \widehat{pvi}_n\right|\right] \leq 2\mathcal{R}_n(\mathcal{G}_\mathcal{V}) + 2\sqrt{M} + 2B\sqrt{\frac{2\log(1/\delta)}{n}}, \tag{50}$$

*where the function family $\mathcal{G}_\mathcal{V} = \{g|g(x,y) = \log f[x](y), f \in \mathcal{V}\}$ and $\mathcal{R}_n$ denotes the Rademacher complexity with $n$ sampled points.*

*Proof.* The result is a consequence of the generalization bound for $\mathcal{V}$-information proposed in (Xu et al., 2020) (Lemma 3 of their Appendix). First, note that we can express $PVI(x \to y) = I_\mathcal{V}(X \to Y) + \epsilon$, where $\epsilon$ is a random variable, and similarly for $\widehat{pvi}(x \to y)$. Also, note that as $f^*$ and $\widehat{f}$ are neural networks, the first term in the estimation of PVI will be fixed to 1, as the network can simply assign biases to the last layer such that $f^*[\varnothing](y) = 0.5$, and similarly for $\widehat{f}$ as the training set is balanced. In that case, we can write:

$$\mathbb{E}_{P_{XY}}\left[\left|pvi(x \to y) - \widehat{pvi}(x \to y)\right|\right] = \mathbb{E}_{P_{XY}}\left[\left|I_\mathcal{V}(X \to Y) + \epsilon_1 - \widehat{I}_\mathcal{V}(X \to Y) - \epsilon_2\right|\right] \tag{51}$$

$$\leq \mathbb{E}_{P_{XY}}\left[\left|I_\mathcal{V}(X \to Y) - \widehat{I}_\mathcal{V}(X \to Y)\right|\right] \tag{52}$$

$$+ \mathbb{E}[|\epsilon_1|] + \mathbb{E}[|\epsilon_2|]$$

$$\leq \left|I_\mathcal{V}(X \to Y) - \widehat{I}_\mathcal{V}(X \to Y)\right| + 2\sqrt{M}, \tag{53}$$

where the last step follows from noting that the absolute difference between the true and estimated $\mathcal{V}$-information doesn't depend on the individual instances, and that the L1-norm is bounded using the variance via the application of the Cauchy-Schwarz inequality. Next, we directly apply Lemma 3 of (Xu et al., 2020), after the additional observation that in this case $H_\mathcal{V}(Y) = \widehat{H}_\mathcal{V}(Y)$. We then have, with probability $p \geq 1 - 2\delta$,

$$\left|I_\mathcal{V}(X \to Y) - \widehat{I}_\mathcal{V}(X \to Y)\right| \leq 2\mathcal{R}_n(\mathcal{G}_\mathcal{V}) + 2B\sqrt{\frac{2\log(1/\delta)}{n}} \tag{54}$$

Applying this to (53) yields the result. $\square$

**Remark 13** (**On the convergence of PVI**). *We note that the result provides a bound on the average error w.r.t the PVI estimation over datapoints, and thus are not uniform convergence bounds. Next, we also note that the result depends on the upper bound on the variance of the neural networks ($M$), which is not trivial to bound. However, overall, the convergence result for PVI still shows us a few important differences w.r.t the convergence bounds for PSI and PMI. First, we note that here, the convergence depends heavily on the choice of $\mathcal{V}$. Choosing very deep and complex neural networks for estimating PVI will lead to a large Rademacher complexity $\mathcal{R}_n(\mathcal{G}_\mathcal{V})$, which will lead to slower convergence. Also, ideally, we want networks to have a smaller variance over its output logits, which will eventually also reduce the value of $M$ and make convergence stronger. This can be achieved by regularizing the outputs of the network to have low variance, and there are approaches in literature which have studied this kind of regularization (Littwin & Wolf, 2018).*

Table 3: Convergence Rate of PMI and PSI using KDE estimator (Averaged over 50 Runs with Standard Deviations Included).

| $n$ | 100 | 1,000 | 10,000 | 100,000 | 1,000,000 |
|---|---|---|---|---|---|
| $|pmi(x;y) - \widehat{pmi}_n|$ | 6.075±8.881 | 1.684±1.209 | 1.268±0.714 | 0.911±0.473 | 0.809±0.301 |
| $n$ | 100 | 200 | 1,000 | 2,000 | 10,000 |
| $\mathbb{E}_{x,y}[|psi(x;y) - \widehat{psi}_{n,m}|]$ | 0.292±0.037 | 0.282±0.036 | 0.270±0.034 | 0.270±0.027 | 0.269±0.028 |

**Experiment on Convergence Rate:** We conduct a simple experiment on Gaussian mixture distributions to test the convergence rates of PMI and PSI. Based on the results shown in Table 3, we have two main observations. First, we find that both the trends of PMI and PSI are within the predicted convergence trends in Proposition 4 and Theorem 5 (we set $\alpha = 1$

as our mixtures are Lipschitz continuous). This re-affirms the convergence bounds being an upper bound on the observed trend with the number of samples $n$. Second, we find that the predicted convergence rate for PMI and PSI are reflective of the theoretical results. Our theoretical results stated that PMI should converge slowly compared to PSI, and the difference is amplified with greater dimensionality. After adjusting for scale and bias (error as $n$ goes to very large values), we find that the observed convergence rate for PSI is indeed greater than that for PMI. Note that for PVI, we found it hard to estimate Rademacher complexity measures for neural network classifiers, so we cannot directly test our convergence rates.

### B.3. Geometric Properties

In the following results, we mainly explore whether geometric properties of the feature distribution, such as the sample-wise margin and the subspace intrinsic dimensionality, can affect the different PI measures. We define the notion of sample-wise margin as the distance of a datapoint $x$ to the other class distribution, when it is encapsulated by a sphere.

First, we provide the general idea of sample-wise margin. In the results that follow, we adopt more specific definitions that are motivated from the general principle in the following definition.

**Definition 6 (Sample-Wise Margin).** *Given $x, y \sim P_{XY}$ and $Y \in \{0, 1\}$ such that $P(X|Y = 0)$ and $P(X|Y = 1)$. The sample-wise margin refers to the distance of the sample $x$ from the distribution $P(X|Y = 1 - y)$, when $P(X|Y = 0)$ and $P(X|Y = 1)$ are non-overlapping. When $P(X|Y = 0)$ and $P(X|Y = 1)$ are overlapping, first we can create non-overlapping probability masses $Q(X|Y = 0)$ and $Q(X|Y = 1)$ which encapsulate most of $P(X|Y = 0)$ and $P(X|Y = 1)$ (fraction of $1 - \epsilon$) respectively. Next, we estimate sample-wise margin as the distance of $x$ from the distribution $Q(X|Y = 1 - y)$.*

We have the following result for PMI, in the context of non-overlapping feature distributions.

**Proposition 6 (PMI and sample-wise margin).** *Let $x, y \sim P_{X,Y}$ and $Y \in \{0, 1\}$ such that $P(X|Y = 0)$ and $P(X|Y = 1)$ are non-overlapping and $P(Y = 0) = P(Y = 1) = 0.5$. Then, we have that $pmi(x; y) = 1$.*

*Proof.* Since $P(X|Y = 0)$ and $P(X|Y = 1)$ are non-overlapping, for a certain sampled $y$, we will have $p(x|y) = 1$ and $p(x|y = 1 - y) = 0$. Thus, we have:

$$pmi(x; y) = \log_2 \frac{p(x|y)}{p(x)} = \log_2 \frac{p(x|y)}{0.5 \left( p(x|y = 0) + p(x|y = 1) \right)} = \log_2 2 = 1$$

$\square$

**Remark 14 (On PMI and sample-wise margin).** *Note that the above result implies that when the distributions $P(X|Y = 0)$ and $P(X|Y = 1)$ are non-overlapping, then $pmi(x; y)$ is always 1 irrespective of the distance of $x$ from the decision boundary. Thus, in this case, sample-wise margin does not affect the PMI at all.*

Next, we highlight the conditions when PSI can be related to both sample-wise margin and intrinsic dimensionality (ID) of the data. First, we define the subspace ID:

**Definition 7 (Subspace Intrinsic Dimensionality).** *The subspace intrinsic dimensionality (ID), denoted by $K_P$, is the dimensionality of the smallest subspace $W$ that contains the support of $P(X)$.*

We have the following result for PSI that relates it to sample-wise margin and the intrinsic dimensionality (ID) of the data. Note that for the overlapping case, there is no unique notion of sample-wise margin, as it depends on how $Q$ is constructed, and also depends on the fraction $(1 - \epsilon)$ of the distribution involved in encapsulating the class-wise distributions. For the following result, we use spheres to construct $Q$, for each class-wise distribution.

**Theorem 2 (PSI and sample-wise margin and ID).** *Given $x, y \sim P_{XY}$ with $Y \in \{0, 1\}$, and assuming $y = 0$ without loss of generality, we consider two non-overlapping spheres $S_1$ and $S_2$ with radii $R_1$ and $R_2$, and centers $C_1$ and $C_2$ such that $x \in S_1$. Here, the sample-wise margin, denoted by $d(x, S_2)$, refers to the distance between $x$ and the surface of $S_2$. The subspace intrinsic dimensionality of $P(X)$ is denoted by $K_P$. Let $\{\theta^T x | S_2\} = \{\theta^T x : x \in S_2\}$ represent the set of points in the real line of the $\theta$ projection of $S_2$. Let $\epsilon = \max_{\theta, x} P(\theta^T x | y = 1, x \in \mathbb{R} - \{\theta^T x : x \in S_2\})$, where $\{\theta^T x : x \in S_2\}$. We also define the following two quantities:*

$$p_{\max} = \max \left\{ \max_{\theta, x \in S_2} p(\theta^T x | y = 1), \max_{\theta, x \in S_1} p(\theta^T x | y = 0) \right\}, \qquad p_{\min} = \min_{\theta, x \in S_1} p(\theta^T x | y = 0).$$

*Then, we have the following lower bound:*

$$psi(x; y) \geq \log \frac{p_{\min}}{p_{\max}} + \left(1 + \log \frac{p_{\max}}{p_{\min} + \epsilon}\right) B_{\gamma(d(x, S_2), R_2)} \left(\frac{K_P - 1}{2}, \frac{1}{2}\right), \tag{55}$$

*where $B_x(a, b)$ denotes the regularized incomplete beta function (Oldham et al., 2008), and $\gamma(a, b) = \frac{a}{a+b}\left(2 - \frac{a}{a+b}\right)$.*

*Proof.* The proof follows from the proof elements of Theorem 1 and 2 of (Wongso et al., 2023a), and Theorem 1 of (Wongso et al., 2023b). First, using the proof of Theorem 1 of (Wongso et al., 2023a), it follows that $\Pr(\theta^T x \in \{\theta^T x | S_2\} | x) = B_{\gamma(d(x, S_2), R_2)}\left(\frac{d_x - 1}{2}\right)$. We arrive at this result by considering two spheres in the context of Theorem 1 of (Wongso et al., 2023a), $S_1'$ being a zero-radius sphere centered at $x$, and $S_2'$ being the same as $S_2$ here.

Given that $y = 0$, we then can write:

$$psi(x; y) = \mathbb{E}_\theta \left[\log \frac{p(y = 0 | \theta^T x)}{p(y = 0)}\right] \tag{56}$$

$$= \Pr(\theta^T x \in \{\theta^T x | S_2\} | x) \cdot \mathbb{E}_{\theta: \theta^T x \in \{\theta^T x | S_2\}} \left[\log \frac{p(y = 0 | \theta^T x)}{p(y = 0)}\right]$$

$$+ \Pr(\theta^T x \notin \{\theta^T x | S_2\} | x) \cdot \mathbb{E}_{\theta: \theta^T x \notin \{\theta^T x | S_2\}} \left[\log \frac{p(y = 0 | \theta^T x)}{p(y = 0)}\right] \tag{57}$$

When $\theta^T x \notin \{\theta^T x | S_2\}$, note that $S_2$ does not play a role in estimating the probabilities. In this cases, we have:

$$p(y = 0 | \theta^T x) = \frac{P(\theta^T x | y = 0)}{p(\theta^T x | y = 0) + p(\theta^T x | y = 1)} \geq \frac{p_{\min}}{p_{\min} + \epsilon}$$

The $\epsilon$ term is a consequence of the fact that only the probability outside the set $\{\theta^T x | S_2\}$ contributes to $p(\theta^T x | y = 1)$ in this case.

When $\theta^T x \in \{\theta^T x | S_2\}$, both $S_1$ and $S_2$ will contribute to estimating the probabilities. In this case, we have:

$$p(y = 0 | \theta^T x) = \frac{p(\theta^T x | y = 0)}{p(\theta^T x | y = 0) + p(\theta^T x | y = 1)} \geq \frac{p_{\min}}{2p_{\max}}$$

This, combined with the fact that $p(y = 0) = p(y = 1) = 0.5$ and $\Pr(\theta^T x \in \{\theta^T x | S_2\} | x) = B_{\gamma(d(x, S_2), R_2)}\left(\frac{d_x - 1}{2}\right)$, then yields:

$$psi(x; y) \geq \left(1 + \log \frac{p_{\min}}{p_{\min} + \epsilon}\right) B_{\gamma(d(x, S_2), R_2)}\left(\frac{d_x - 1}{2}\right) \tag{58}$$

$$+ \log \frac{p_{\min}}{p_{\max}} \left(1 - B_{\gamma(d(x, S_2), R_2)}\left(\frac{d_x - 1}{2}\right)\right) \tag{59}$$

$$= \log \frac{p_{\min}}{p_{\max}} + \left(1 + \log \frac{p_{\max}}{p_{\min} + \epsilon}\right) B_{\gamma(d(x, S_2), R_2)}\left(\frac{d_x - 1}{2}, \frac{1}{2}\right) \tag{60}$$

Furthermore, given that all of $P(X)$ lies within a subspace of dimensionality $K_P$, we can convert our analysis into a space of dimensionality $K_P$ instead, as implied from Theorem 2 of (Wongso et al., 2023a). Note that in doing so, the distances do not change, and the measures $\epsilon, p_{\max}, p_{\min}$ all stay the same, because the dimensionality of the null-space within the projections has zero measure. This yields the final result:

$$psi(x; y) \geq \log \frac{p_{\min}}{p_{\max}} + \left(1 + \log \frac{p_{\max}}{p_{\min} + \epsilon}\right) B_{\gamma(d(x, S_2), R_2)}\left(\frac{K_P - 1}{2}, \frac{1}{2}\right), \tag{61}$$

$\square$

**Remark 15** (**On the lower bound of PSI**). *Note that when $x$ is further away from $S_2$, i.e. a larger sample-wise margin, it leads to a larger lower bound on the PSI. Thus, in this case, PSI will likely be larger. This generalizes the result in (Wongso et al., 2023b), which was only for symmetric non-overlapping distributions $P(X|Y=0)$ and $P(X|Y=1)$. As Theorem 2 shows, PSI can be sensitive to both soft and hard margins. Furthermore, in three scenarios we expect the bound to be tight. (i) For distributions where $\epsilon$ is small, and $p_{max} >> p_{min}$. (ii) When the radius $R_2$ is large, or the distance $d(x, S_2)$ is large. (iii) When the intrinsic dimensionality $K_P$ is small. Thus, for high-dimensional data, if it lies on a low dimensional manifold, we will get a significantly tighter result. Furthermore, we note that none of the terms $p_{min}, p_{max}, \epsilon, R_2, K_P$ are dependent on the sample-wise margin $d(x, S_2)$. Thus, the only term affected by sample-wise margin is the regularized incomplete beta function $B_{\gamma(d(x,S_2),R_2)}\left(\frac{K_P-1}{2}, \frac{1}{2}\right)$. Therefore, our hypothesis that the lower bound of $psi(x; y)$ increases as the sample-wise margin increases is valid.*

**Remark 16** (**On the sample-wise margin definition in Theorem 2**). *Note that the definition of sample-wise margin here $d(x, S_2)$ converges to the classical definition of margin w.r.t a linear decision boundary when $R_2 \to \infty$.*

**Remark 17** (**On the choice of $S_1$, $S_2$, and $\epsilon$**). *As mentioned in the main paper, here we provide some more context to the choice of the spheres $S_1$ and $S_2$, and the nature of $\epsilon$. Note that the choice of $S_1$ does not affect the result much, as the only main constraint for $S_1$ is that $x$ must be contained within it. As such, the radius $R_1$ also does not directly impact the result. However, $S_2$ should be ideally chosen such that it contains as much of the distribution $P(X|Y=1)$. To see this, we mainly look at how the choice of $S_2$ impacts $\epsilon$. Note that if $\epsilon$ is very large, such that $p_{max} < p_{min} + \epsilon$, then the dependence on sample-wise margin reverses (less margin leads to more psi). To avoid this, we can always choose $S_2$ such that $\epsilon$ is small. Let $S_2$ be chosen such that $p(x \in S_2|y=1) = \rho$. Furthermore, let us assume that there is another bigger sphere $S_3$ such that $S_2$ is contained in $S_3$, such that $p(x \in S_3|y=1) = 1$. Let the radius of $S_3$ be $R_3$. Then, we can approximate $\epsilon$ as $(1-\rho)/(2(R_3 - R_2))$. This is because the projection of $S_2$ will have a length of $2R_2$ and similarly for $S_3$. Thus, if we choose $S_2$ such that $\rho$ is made arbitrarily close to 1, we can make $\epsilon$ arbitrarily close to zero. However, do note that although this can be done when the distributions $P(X|Y=0)$ and $P(X|Y=1)$ have less overlap, for the case where $P(X|Y=0)$ and $P(X|Y=1)$ are highly overlapping, this may not be possible. As in most of our experiments $x$ is taken from the penultimate layer of neural networks which have separable features, the assumption will hold with high probability.*

**Remark 18** (**On sensitivity to hard margins**). *When $P(X|Y=0)$ and $P(X|Y=1)$ are clearly separated, one should ideally have maximum confidence estimates everywhere. But the fact that we do not know the ground truth distribution $P(X, Y)$ implies that even when the estimate of $P$, denoted by $Q(X, Y)$, from the training data, is perfectly separated, the separation of the true unknown $P(X, Y)$ will be most likely smaller with potential overlap. This is because $Q(X, Y)$ clearly has a significant chance of overfitting the true distributions, as the objective of the classifier is always to separate the training feature distributions anyway. Due to this potential overestimation of the real margin, encoding additional geometric information about $Q(X, Y)$, such as the hard margin involved in the perfect separation, can inform about the probability of $P(X|Y=0)$ and $P(X|Y=1)$ being perfectly separated as well. If $Q(X, Y)$ has a very small hard margin, then it is possible that $P(X, Y)$ ends up with overlapping class-wise feature distributions, and if it has a very large hard margin, then the opposite is likely. Lastly, correlation between the hard margin between the class-wise feature distributions and generalization has indeed been observed in literature (Grønlund et al., 2020), showcasing the significance of this issue.*

Finally, we have the following result to relate PVI to the sample-wise margin.

**Proposition 7** (**PVI and sample-wise margin**). *We are given a neural network with function $f : \mathbb{R}^d \to \mathbb{R}^2$ for classifying points $X$ into two labels $Y \in \{0, 1\}$, and we are given that $P(Y=0) = P(Y=1) = 0.5$. We assume that the final outputs of $f$ are passed through a softmax operator with temperature $T = 1$, to yield the output $softmax(f(X))$. We are given an instance $(x, y) \sim P(X, Y)$. Given $x$ as the input, we define margin $\tau$ as in (Vemuri, 2020), where*

$$\tau = \frac{f(x)_y - f(x)_{1-y}}{\|\nabla_x(f(x)_y) - \nabla_x(f(x)_{1-y})\|_2}. \tag{62}$$

*If $M = \max_x\{||\nabla_x(f(x)_y)||, ||\nabla_x(f(x)_{1-y})||\}$, then we have: $pvi(x \to y) \leq 1 - \log\left(1 + e^{-2M\tau}\right)$.*

*Proof.* As we consider the function outputs before the softmax here, we re-represent the two terms of PVI. The first term of PVI, is now represented as $-\log softmax(f)[\varnothing](y)$, which will be equal to 1, as the neural network can simply learn the biases of the last layer and set them such that $softmax(f)[\varnothing](y) = softmax(f)[\varnothing](1 - y) = 0.5$. Note that, as $\tau = \frac{f(x)_y - f(x)_{1-y}}{\|\nabla_x(f(x)_y) - \nabla_x(f(x)_{1-y})\|_2}$ and $M = \max_x\{|\nabla_x(f(x)_y)|, |\nabla_x(f(x)_{1-y})|\}$, we can write

$$f[x](1 - y) - f[x](y) \leq \sqrt{(2M^2 + 2M^2)}\tau = 2M\tau \tag{63}$$

Table 4: Correlation of PMI/PSI/PVI with Margin (Averaged over 5 Runs with Std Dev shown). Best results are highlighted in bold.

| Method | MLP, MNIST | CNN, F-MNIST | VGG16, STL-10 | ResNet50, CIFAR-10 |
|---|---|---|---|---|
| PMI | 0.398±0.029 | 0.429±0.034 | 0.619±0.011 | 0.637±0.019 |
| PSI | **0.657±0.022** | **0.846±0.006** | **0.809±0.006** | **0.758±0.033** |
| PVI | 0.327±0.025 | 0.368±0.008 | 0.604±0.010 | 0.563±0.011 |

Like before, let $f' = \arg\min_{g \in \mathcal{V}} \mathbb{E}_{(X,Y) \sim P_{XY}}[-\log g[X](Y)]$, denote the trained neural network that estimates conditional $\mathcal{V}$-entropy. Now, the second term for PVI will be represented as $\log \text{softmax}(f')[x](y)$. Then, given $x, y \sim P_{XY}$, we have:

$$\log\left(\text{softmax}(f')[x](y)\right) = \log\left(\frac{e^{f'[x](y)}}{e^{f'[x](y)} + e^{f'[x](1-y)}}\right) \tag{64}$$

$$= \log\left(\frac{1}{e^{f'[x](1-y)-f'[x](y)} + 1}\right) \tag{65}$$

$$\leq -\log\left(1 + e^{-2M\tau}\right) \tag{66}$$

Then, the result direct follows from the expression of PVI. $\square$

**Remark 19** (**On PVI and sample-wise margin**). *As the above result shows, PVI can indeed be sensitive to the sample-wise margin, and thus datapoints $x$ which are near to the decision boundary can be expected to have a lower PVI and vice versa. However, the raw PVI values may not be very sensitive to margin. For $\tau >> 1$, we can approximate $pvi(x \rightarrow y) \leq 1 - e^{-4M^2\tau}$, which converges to 1 quickly as $\tau$ increases and the differences become smaller for larger $\tau$. Thus, if one were to replace the PVI values by their relative rank, we could potentially see a higher correlation. As our experiments use the pointwise measures to rank confidence scores relatively among samples, samples with larger PVI will likely correspond to the samples with larger sample-wise margins.*

**Experiment on Correlation to Margin:** We perform an experiment to examine whether samples closer to the decision boundary (smaller margin) are assigned lower confidence scores by the various measures compared to those located further away (higher margin). We aim to test our hypothesis that PSI is the most sensitive to sample-wise margin. We approximate the sample-wise margin using the method provided in (Elsayed et al., 2018), which approximates the smallest distance of a datapoint **x** to the decision boundary by:

$$d_{i,j}(\mathbf{x}) \approx \frac{f(\mathbf{x})_i - f(\mathbf{x})_j}{\|\nabla_{\mathbf{x}}(f(\mathbf{x})_i) - \nabla_{\mathbf{x}}(f(x)_j)\|_2} \tag{67}$$

where we choose $f(\mathbf{x})_i$ and $f(\mathbf{x})_j$ to be the highest and second highest logits of the neural network $f$ (also used in Proposition 7) and $\nabla$ represents the gradient operator. Then, we compute the Pearson correlation between the margin and the confidence estimates returned by the different PI measures. The results are shown in Table 4. In addition, we use Uniform Manifold Approximation and Projection (UMAP) to visualize the features of the penultimate layer on the test dataset. We rank the PMI, PSI, and PVI for each sample and visualize these rankings using color bars in the UMAP plots. As shown in Table 4, we find that PSI is the most correlated with margin, followed by PMI and then PVI, supporting the theory. The higher correlation of PMI with margin compared to PVI could be attributed to the decrease of sensitivity of PVI when $M$ (related to the complexity of the network) is large. In Figure 1, we find that for all measures, as the samples get closer to the decision boundary, the values generally decrease. We generally observe that PSI tends to rank highly misclassified classes lower than those that are often classified correctly. For example, in the STL-10 dataset, animal categories generally receive lower-ranked confidence scores overall (showing more blue than pink).

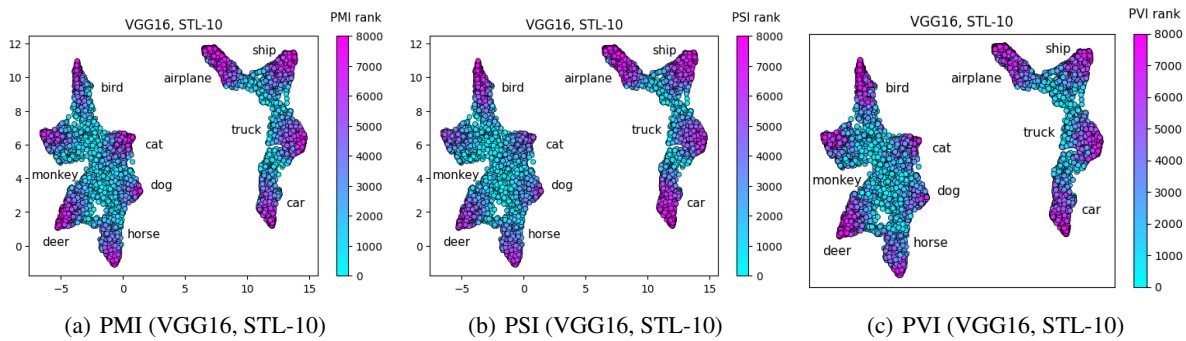

(a) PMI (VGG16, STL-10)          (b) PSI (VGG16, STL-10)          (c) PVI (VGG16, STL-10)

Figure 1: UMAP visualization of the penultimate layer features.

## C. Benchmarks & Experimental Details

### C.1. Benchmark Datasets & Architectures

Below is a list of the benchmark datasets we use in our experiments:

1. **MNIST** is a dataset comprising of 28×28 grayscale images of handwritten digits from 0 to 9.
2. **Fashion MNIST** is a dataset comprising of 28×28 grayscale images of fashion products from 10 classes: T-shirt, trouser, pullover, dress, coat, sandal, shirt, sneaker, bag, ankle boot.
3. **STL-10** is a subset of the ImageNet dataset, consisting of 96×96 color images from 10 classes: airplane, bird, car, cat, deer, dog, horse, monkey, ship, truck. It was primarily developed for unsupervised learning, and thus most of the samples are unlabelled.
4. **CIFAR-10** is a dataset consisting of 32×32 color images from 10 classes: airplane, automobile, bird, cat, deer, dog, frog, horse, ship, and truck.
5. **CIFAR-100** is an expanded version of CIFAR-10, comprising 100 classes.
6. **Stanford Dogs** is a dataset consisting of color images of various sizes, spanning 120 different dog breeds.
7. **Tiny-ImageNet** is a dataset consisting of 64×64 color images from 200 classes. It is a smaller subset of the original ImageNet dataset.

All of these datasets are publicly accessible through the TensorFlow Datasets catalog at https://tensorflow.org/datasets/catalog/overview. All images, except those from MNIST and Fashion-MNIST, are resized to 224×224. We split the dataset into training, validation and test sets.

Table 5: The architecture of the basic CNN.

| Layer Type | Parameters |
| --- | --- |
| Convolutional | 32 filters, kernel_size=3×3, strides=1, padding=same, ReLU |
| Convolutional | 32 filters, kernel_size=3×3, strides=1, padding=same, ReLU |
| Max Pooling | pool_size=2×2 |
| Dropout | rate=0.3 |
| Convolutional | 64 filters, kernel_size=3×3, strides=1, padding=same, ReLU |
| Convolutional | 64 filters, kernel_size=3×3, strides=1, padding=same, ReLU |
| Max Pooling | pool_size=2×2 |
| Dropout | rate=0.3 |
| Convolutional | 128 filters, kernel_size=3×3, strides=1, padding=same, ReLU |
| Convolutional | 128 filters, kernel_size=3×3, strides=1, padding=same, ReLU |
| Max Pooling | pool_size=2×2 |
| Dropout | rate=0.3 |
| Fully-Connected | 128 units, ReLU |
| Fully-Connected | $K$ units (where $K$ is the number of classes), softmax |

Below is a list of neural network architectures that we use in our experiments:

Table 6: The top layers of the benchmark model's architecture.

| Layer Type | Parameters |
|---|---|
| Base Network | Weights are pre-trained on ImageNet dataset |
| Fully-Connected | 256 units, ReLU |
| Dropout | rate=0.3 |
| Fully-Connected | 128 units, ReLU |
| Fully-Connected | $K$ units (where $K$ is the number of classes), softmax |

1. **Multi-layer Perceptron (MLP)**: We implemented a simple MLP, consisting of three hidden layers with 512 units and ReLU activation each.
2. **Convolutional Neural Network (CNN)**: We implemented a simple CNN with a detailed architecture as illustrated in Table 5. We refer to this as the Basic CNN.
3. **VGG16**: We loaded the base model of VGG16 from Tensorflow and excluded the three fully-connected layers at the top of the network.
4. **ResNet50**: We loaded the base model of ResNet50V2 from Tensorflow and excluded the three fully-connected layers at the top of the network.
5. **ResNet101**: We loaded the base model of ResNet101V2 from Tensorflow and excluded the three fully-connected layers at the top of the network.
6. **InceptionV3**: We loaded the base model of InceptionV3 from Tensorflow and excluded the three fully-connected layers at the top of the network.
7. **DenseNet121**: We loaded the base model of DenseNet121 from Tensorflow and excluded the three fully-connected layers at the top of the network.

For all the pre-trained networks, we incorporated four new layers on top of the base network, as detailed in Table 6. All the pre-trained network modules are publicly accessible through the Tensorflow Keras Applications catalog at https://tensorflow.org/api_docs/python/tf/keras/applications.

### C.2. Hyperparameters

We use the AdamW optimizer with a learning rate of 1e-4 and weight decay of 1e-4, with a batch size of 128. A learning rate scheduler (ReduceLROnPlateau) is employed, which monitors validation accuracy and reduces the learning rate by a factor of 0.5 if it does not improve for 5 consecutive epochs. Early stopping is also applied, monitoring validation accuracy with a patience of 10 epochs, and restoring the best weights upon stopping. We train for a maximum of 100 epochs, except for Fashion-MNIST, where we use 300 epochs. All experiments were performed using a single NVIDIA A100 (80GB SXM) GPU. We report the classification errors for the different model-dataset pairs in our experiments in Table 7.

Table 7: Train, Validation and Test Classification Error in Percentage for the Different Model-Dataset Pairs (Averaged over 5 Runs with Standard Deviations Included)

| MODEL, DATASET | Train Error | Validation Error | Test Error |
|---|---|---|---|
| MLP, MNIST | $0.00 \pm 0.00$ | $1.56 \pm 0.04$ | $1.53 \pm 0.03$ |
| BASIC CNN, FASHION MNIST | $0.01 \pm 0.01$ | $6.21 \pm 0.09$ | $6.52 \pm 0.19$ |
| VGG16, STL-10 | $0.00 \pm 0.01$ | $8.73 \pm 0.51$ | $8.94 \pm 0.41$ |
| RESNET50, CIFAR-10 | $0.00 \pm 0.01$ | $4.39 \pm 0.21$ | $4.78 \pm 0.15$ |
| RESNET101, CIFAR-100 | $0.06 \pm 0.07$ | $21.62 \pm 0.53$ | $21.60 \pm 0.58$ |
| INCEPTIONV3, STANFORD DOGS | $0.14 \pm 0.04$ | $24.47 \pm 0.94$ | $23.46 \pm 0.65$ |
| DENSENET121, TINY-IMAGENET | $0.02 \pm 0.01$ | $27.89 \pm 0.44$ | $27.95 \pm 0.31$ |

### C.3. Details for Experiments in Main Paper

Below, we provide more details on the experiments presented in the main paper.

C.3.1. DETAILS FOR EXPERIMENT IN SECTION 4.1 (FAILURE PREDICTION)

In this experiment, the goal is to compare the effectiveness of the three PI measures for misclassification detection and selective prediction. We formulate the problem as a binary classification task where we have a binary failure label:

$$y_f = \mathbb{1}(y \neq \hat{y}) \tag{68}$$

In other words, we assign label 1 for misclassified samples and 0 for correctly classified samples. Let $c$ be the confidence scores quantified by different approaches. For a threshold value $\tau$, we can compute:

$$\text{TP}_f(\tau) = \sum_{i=1}^{N}(1 - y_{f,i}) \cdot \mathbb{1}(c \geq \tau) \qquad \text{FP}_f(\tau) = \sum_{i=1}^{N} y_{f,i} \cdot \mathbb{1}(c \geq \tau) \tag{69}$$

$$\text{FN}_f(\tau) = \sum_{i=1}^{N}(1 - y_{f,i}) \cdot \mathbb{1}(c < \tau) \qquad \text{TN}_f(\tau) = \sum_{i=1}^{N} y_{f,i} \cdot \mathbb{1}(c < \tau) \tag{70}$$

From these, we can compute the following:

$$\text{Sensitivity}_f(\tau) = \frac{\text{TP}_f(\tau)}{\text{TP}_f(\tau) + \text{FN}_f(\tau)} \tag{71}$$

$$\text{Precision}_f(\tau) = \frac{\text{TP}_f(\tau)}{\text{TP}_f(\tau) + \text{FP}_f(\tau)} \tag{72}$$

$$\text{FPR}_f(\tau) = \frac{\text{FP}_f(\tau)}{\text{TN}_f(\tau) + \text{FP}_f(\tau)} \tag{73}$$

$$\tag{74}$$

In misclassification detection, the two commonly used metrics are AUROC (Area under Receiver Operating Curve) and AUPRC (Area under Precision-Recall Curve) to evaluate performance on a multi-threshold list $\{\tau_t\}_{t=0}^{T}$ of length $T$.

The **AUROC** is defined as:

$$\text{AUROC}_f = \sum_{t=1}^{T}(\text{FPR}_f(\tau_t) - \text{FPR}_f(\tau_{t-1})) \cdot \frac{(\text{Sensitivity}_f(\tau_t) + \text{Sensitivity}_f(\tau_{t-1}))}{2} \tag{75}$$

$$= \sum_{t=1}^{T} \frac{\sum_{i=1}^{N} y_{f,i} \cdot (\mathbb{1}(c \geq \tau_t) - \mathbb{1}(c \geq \tau_{t-1}))}{\sum_{i=1}^{N} y_{f,i}} \cdot \frac{\sum_{i=1}^{N}(1 - y_{f,i}) \cdot (\mathbb{1}(c \geq \tau_t) + \mathbb{1}(c \geq \tau_{t-1}))}{2 \cdot \sum_{i=1}^{N}(1 - y_{f,i})} \tag{76}$$

The **AUPR** is defined as:

$$\text{AUPR}_{f,\text{success}} = \sum_{t=1}^{T}(\text{Sensitivity}_f(\tau_t) + \text{Sensitivity}_f(\tau_{t-1})) \cdot \text{Precision}_f(\tau_t) \tag{77}$$

$$= \sum_{t=1}^{T} \frac{\sum_{i=1}^{N}(1 - y_{f,i}) \cdot (\mathbb{1}(c \geq \tau_t) - \mathbb{1}(c \geq \tau_{t-1}))}{\sum_{i=1}^{N}(1 - y_{f,i})} \cdot \frac{\sum_{i=1}^{N}(1 - y_{f,i}) \cdot \mathbb{1}(c \geq \tau_t)}{\sum_{i=1}^{N} \mathbb{1}(c \geq \tau_t)} \tag{78}$$

AUPRC is more informative than AUROC when there is a significant difference between the positive and negative class base rates. However, AUPRC is heavily influenced by the base rate of the positive class. Therefore, as suggested by (Hendrycks & Gimpel, 2017), we present two types of AUPRC results: **AUPR$_{f,\text{success}}$**, where the success class is treated as positive, and **AUPR$_{f,\text{error}}$**, where the error class is treated as positive. The error classes can be treated as positive by labeling them positive and multiplying the confidence scores $c$ by -1. The AUPR$_{f,\text{error}}$ is defined as:

$$\text{AUPR}_{f,\text{error}} = \sum_{t=1}^{T}(\text{Sensitivity}_f(\tau_t) + \text{Sensitivity}_f(\tau_{t-1})) \cdot \text{Precision}_f(\tau_t) \tag{79}$$

$$= \sum_{t=1}^{T} \frac{\sum_{i=1}^{N} y_{f,i} \cdot (\mathbb{1}(c < \tau_t) - \mathbb{1}(c < \tau_{t-1}))}{\sum_{i=1}^{N} y_{f,i}} \cdot \frac{\sum_{i=1}^{N} y_{f,i} \cdot \mathbb{1}(c < \tau_t)}{\sum_{i=1}^{N} \mathbb{1}(c < \tau_t)} \tag{80}$$

Additionally, we report the **FPR95** metric, which measures the false positive rate when the true positive rate is at 95%.

In selective prediction, given a threshold $\tau$, we filter out the samples with confidence $c < \tau$, and compute the performance on the remaining samples $c \geq \tau$. In this context, the risk is defined as the error rate of the remaining samples after selection:

$$\text{Risk}(\tau) = 1 - \text{Precision}_f(\tau) = \frac{\sum_{i=1}^{N} y_{f,i} \cdot \mathbb{1}(c \geq \tau)}{\sum_{i=1}^{N} \mathbb{1}(c \geq \tau_t)} \tag{81}$$

Coverage is defined as the proportion of samples remaining after selection:

$$\text{Coverage}(\tau) = \frac{\sum_{i=1}^{N} \mathbb{1}(c \geq \tau_t)}{N} \tag{82}$$

The most common metric used in selective prediction is the AURC (Area under Risk-Coverage Curve) which evaluates performance on a multi-threshold list $\{\tau_t\}_{t=0}^{T}$ of length $T$. The **AURC** is defined as:

$$\text{AURC} = \sum_{t=1}^{T} (\text{Coverage}(\tau_t) - \text{Coverage}(\tau_{t-1})) \cdot \frac{(\text{Risk}(\tau_t) + \text{Risk}(\tau_{t-1}))}{2} \tag{83}$$

$$= \sum_{t=1}^{T} \frac{\sum_{i=1}^{N} (\mathbb{1}(c \geq \tau_t) - \mathbb{1}(c \geq \tau_{t-1}))}{N} \cdot \left( \frac{\sum_{i=1}^{N} y_{f,i} \cdot \mathbb{1}(c \geq \tau)}{2 \cdot \sum_{i=1}^{N} \mathbb{1}(c \geq \tau_t)} + \frac{\sum_{i=1}^{N} y_{f,i} \cdot \mathbb{1}(c \geq \tau_{t-1})}{2 \cdot \sum_{i=1}^{N} \mathbb{1}(c \geq \tau_{t-1})} \right) \tag{84}$$

A common criticism of the AURC metric is that it does not allow for meaningful comparisons across problems (Geifman et al., 2019). The same AURC value can correspond to an ideal confidence estimator for one classifier (with high overall risk) and to a completely random confidence estimator for another classifier (with low overall risk). To address this limitation, Excess-AURC (**E-AURC**) normalizes the AURC against an oracle confidence estimator $g^*$ that perfectly orders samples in decreasing loss. Formally, for a classifier $h$ and confidence estimator $g$, E-AURC is defined as:

$$\text{E-AURC}(h, g) = \text{AURC}(h, g) - \text{AURC}(h, g^*) \tag{85}$$

where $\text{AURC}(h, g^*)$ represents the minimal achievable AURC for $h$. The oracle $g^*$ achieves perfect ranking, ensuring the smallest possible risk for every coverage level. By construction, E-AURC = 0 for an ideal estimator, and positive values indicate excess risk due to imperfect ranking of samples by confidence.

Next, we provide details on the benchmark methods against which we compare our methods. Let $\mathbf{z}$ represent the logits of the network (output of the last layer before the softmax function). For $K$ number of classes, the softmax function $\sigma$ is defined as:

$$\sigma_k(\mathbf{z}) = \frac{e^{\mathbf{z}_i}}{\sum_{j=1}^{K} e^{\mathbf{z}_j}} \tag{86}$$

where $\sigma_k(\mathbf{z})$ denotes the $k$-th element of $\sigma(\mathbf{z})$.

We define the maximum softmax probability (MSP), the softmax margin (SM), the max logit (ML), the logits margin (LM), the negative entropy (NE), and the negative Gini (NG) as follows:

$$\text{MSP}(\mathbf{z}) := \sigma_{\hat{y}}(\mathbf{z}) \tag{87}$$

$$\text{SM}(\mathbf{z}) := \sigma_{\hat{y}}(\mathbf{z}) - \max_{k \in \mathcal{Y}: k \neq \hat{y}} \sigma_k(\mathbf{z}) \tag{88}$$

$$\text{ML}(\mathbf{z}) := z_{\hat{y}} \tag{89}$$

$$\text{LM}(\mathbf{z}) := z_{\hat{y}} - \max_{k \in \mathcal{Y}: k \neq \hat{y}} z_k \tag{90}$$

$$\text{NE}(\mathbf{z}) := \sum_{k \in \mathcal{Y}} \sigma_k(\mathbf{z}) \log \sigma_k(\mathbf{z}) \tag{91}$$

$$\text{NG}(\mathbf{z}) := \sum_{k \in \mathcal{Y}} \sigma_k(\mathbf{z})^2 - 1 \tag{92}$$

where $\hat{y} = \arg\max_{k \in \mathcal{Y}} z_k$ is the predicted label.

Aside from ML and LM, the remaining methods apply temperature scaling to the logits or PI values, with the scaling parameter optimized using negative log-likelihood on the validation set.

C.3.2. DETAILS FOR EXPERIMENT IN SECTION 4.2 (CONFIDENCE CALIBRATION)

In this experiment, the goal is to determine to what extent the confidence scores estimated by the three PI measures reflect the true correctness likelihood (well-calibrated). A commonly used calibration metric is **Expected Calibration Error (ECE)** which bins the predictions in $[0, 1]$ under $M$ equally-spaced intervals (we choose $M = 10$), and then averages the accuracy/confidence in each bin. ECE is defined as follows:

$$ECE = \sum_{m=1}^{M} \frac{|B_m|}{n} |\text{acc}(B_m) - \text{conf}(B_m)| \tag{93}$$

However, the ECE has noteworthy limitations: it relies on a binning approach that introduces bias–variance trade-offs (fewer bins increase bias, more bins escalate variance), considers only the maximum predicted probabilities (ignoring potentially meaningful information in lower-confidence estimates), and may mislead by yielding low calibration error even in poorly accurate models (Pavlovic, 2025; Nixon et al., 2019). Nevertheless, we still present the results for ECE in Table 8 in Appendix C.4.1.

Instead, we consider the metrics introduced by (Nixon et al., 2019):

Static Calibration Error (**SCE**) is a multiclass extension of the Expected Calibration Error (ECE) that evaluates calibration across all predicted class probabilities rather than only the top prediction. For each class, predicted probabilities are partitioned into bins, and the absolute difference between the bin's average confidence and accuracy is computed, weighted by the number of samples in that bin. The SCE is the average of these weighted errors over all classes, providing a more comprehensive measure of calibration. Unlike ECE, SCE can be zero only if the model is perfectly calibrated for every class, making it a stricter and more informative metric for multiclass settings.

Class-Conditional Static Calibration Error (**CC-SCE**) is a multiclass calibration metric that extends SCE by computing calibration error separately for each class and then averaging over all classes. For each class, predicted probabilities for that class (regardless of whether it is the top prediction) are partitioned into bins, and the absolute difference between the bin's average confidence and accuracy is calculated, weighted by the proportion of samples in that bin. Averaging these weighted errors across all classes yields the final score. This class-wise approach captures calibration behaviour for every class individually, avoiding the bias of standard ECE, which only considers the most confident class per sample.

Adaptive Calibration Error (**Ada-SCE**) is a multiclass calibration metric that, like SCE, measures calibration across all predicted class probabilities but avoids the fixed-bin bias of SCE by using adaptive binning. Instead of dividing the probability range [0,1] into equal-width bins, ACE sorts the predicted probabilities for each class and partitions them so that each bin contains approximately the same number of samples. This ensures that even low-probability regions with few samples are adequately represented, reducing bias in sparse regions. For each class and bin, the absolute difference between average confidence and accuracy is computed and weighted equally across bins and classes. Averaging these values yields ACE, which offers a more reliable estimate of calibration—especially for skewed probability distributions or imbalanced datasets.

Root Mean Squared Calibration Error (**RMS-SCE**) extends SCE by computing the square root of the mean squared difference between predicted confidence and empirical accuracy. It applies the L2 error to penalize larger calibration deviations more strongly than absolute-error metrics. Each bin holds roughly 100 predictions, ensuring stable accuracy estimates within bins. RMSCE is particularly useful when large miscalibrations are of greater concern, as the squaring step magnifies their impact before averaging and taking the square root.

We additionally consider 4 other common metrics in Appendix C.4.1:

The Maximum Calibration Error (**MCE**) measures the worst-case deviation between predicted confidence and empirical accuracy across all bins. Unlike ECE, which averages the bin-wise calibration errors weighted by bin size, MCE only considers the largest absolute gap. Formally, it is defined as:

$$MCE = \max_{m \in \{1,...,M\}} |\text{acc}(B_m) - \text{conf}(B_m)| \tag{94}$$

where $B_m$ denotes the set of samples whose predicted confidence falls into the $m$-th bin, $\text{acc}(B_m)$ is the accuracy within bin $m$, and $\text{conf}(B_m)$ is the average predicted confidence in bin $m$.

The Average Calibration Error (**ACE**) measures the mean absolute difference between predicted confidence and empirical accuracy across all bins, without weighting by bin size. Unlike ECE, which weights each bin by its number of samples, ACE treats all bins equally, providing an unweighted estimate of calibration error. Formally, it is defined as:

$$\text{ACE} = \frac{1}{M} \sum_{m=1}^{M} |\text{acc}(B_m) - \text{conf}(B_m)| \tag{95}$$

where $B_m$ denotes the set of samples whose predicted confidence falls into the $m$-th bin, $\text{acc}(B_m)$ is the accuracy within bin $m$, and $\text{conf}(B_m)$ is the average predicted confidence in bin $m$.

The Negative Log Likelihood (**NLL**) measures the quality of the predicted probability distribution by penalizing incorrect or overconfident predictions. Formally, for $N$ samples, it is defined as:

$$\text{NLL} = -\frac{1}{N} \sum_{i=1}^{N} \log p_\theta(y_i \mid x_i) \tag{96}$$

where $p_\theta(y_i \mid x_i)$ denotes the predicted probability assigned to the true class $y_i$ for input $x_i$ under model parameters $\theta$.

The Brier Score (**BS**) measures the mean squared difference between the predicted probability distribution and the actual outcome, providing an overall assessment of the accuracy of probabilistic predictions. It penalizes both overconfidence and underconfidence in predictions. Formally, for $N$ samples and $K$ classes, it is defined as:

$$\text{BS} = \frac{1}{N} \sum_{i=1}^{N} \sum_{k=1}^{K} \left( p_\theta(y_i = k \mid x_i) - \mathbb{1}(y_i = k) \right)^2 \tag{97}$$

where $p_\theta(y_i = k \mid x_i)$ is the predicted probability that sample $i$ belongs to class $k$, and $\mathbb{1}(y_i = k)$ is the indicator function that equals 1 if the true label is $k$ and 0 otherwise.

Similar to the failure prediction experiment, we apply temperature scaling to all the logits and PI values, with the scaling parameter optimized using negative log-likelihood on the validation set.

## C.4. Additional Results for Experiments in Main Paper

### C.4.1. ADDITIONAL RESULTS FOR EXPERIMENT IN SECTION 4.2 (CONFIDENCE CALIBRATION)

For completeness, in Table 8, we also report results for 5 additional common calibration metrics: Maximum Calibration Error (MCE), Average Calibration Error (ACE), Negative Log-Likelihood (NLL), and Brier Score (BS). Generally, MSP performs well on ECE, MCE, and ACE; however, as noted earlier, these metrics have known limitations, such as bias–variance trade-offs and reliance solely on the maximum predicted probability. For NLL and BS, both MSP and PVI achieve competitive results.

Table 8: Comparison of Confidence Estimation Methods for Confidence Calibration (Averaged over 10 Runs).

| Model, Dataset | Method | ECE ↓ | MCE ↓ | ACE ↓ | NLL ↓ | BS ↓ |
|---|---|---|---|---|---|---|
| MLP, MNIST | MSP | **0.34 ± 0.07** | 0.10 ± 0.04 | 9.46 ± 2.28 | 6.45 ± 0.18 | 3.02 ± 0.07 |
| | PMI | 0.31 ± 0.06 | **0.08 ± 0.03** | **7.33 ± 1.39** | **6.32 ± 0.17** | **2.97 ± 0.07** |
| | PSI | 1.75 ± 0.16 | 0.28 ± 0.04 | 23.26 ± 1.43 | 9.56 ± 0.37 | 3.99 ± 0.13 |
| | PVI | 0.59 ± 0.08 | **0.13 ± 0.03** | 11.63 ± 1.81 | 6.47 ± 0.18 | 3.02 ± 0.07 |
| CNN, F-MNIST | MSP | **0.72 ± 0.10** | **0.18 ± 0.02** | **5.12 ± 1.47** | **19.72 ± 0.54** | **10.23 ± 0.34** |
| | PMI | 1.27 ± 0.10 | 0.30 ± 0.06 | 8.17 ± 1.14 | 21.93 ± 0.33 | 10.56 ± 0.29 |
| | PSI | 2.85 ± 0.28 | 0.32 ± 0.03 | 18.40 ± 0.96 | 24.54 ± 1.12 | 12.10 ± 0.56 |
| | PVI | 1.48 ± 0.30 | 0.18 ± 0.03 | 10.28 ± 2.65 | 19.71 ± 0.54 | 10.22 ± 0.34 |
| VGG16, STL-10 | MSP | 1.51 ± 0.32 | 0.46 ± 0.15 | **5.40 ± 0.53** | **27.62 ± 0.84** | **13.09 ± 0.40** |
| | PMI | **1.07 ± 0.13** | **0.21 ± 0.04** | 5.51 ± 1.18 | 28.71 ± 0.77 | 13.25 ± 0.39 |
| | PSI | 1.40 ± 0.20 | **0.21 ± 0.03** | 6.85 ± 1.16 | 28.26 ± 0.86 | 13.21 ± 0.39 |
| | PVI | 2.34 ± 0.35 | 0.31 ± 0.06 | 11.27 ± 1.71 | 27.44 ± 0.81 | 13.02 ± 0.40 |
| ResNet50, CIFAR-10 | MSP | **1.06 ± 0.06** | 0.45 ± 0.06 | **6.15 ± 0.81** | **15.51 ± 0.45** | **7.24 ± 0.20** |
| | PMI | 1.11 ± 0.13 | 0.48 ± 0.08 | 6.82 ± 1.01 | 16.44 ± 0.56 | 7.31 ± 0.22 |
| | PSI | 1.01 ± 0.11 | **0.33 ± 0.07** | 9.52 ± 2.26 | 16.05 ± 0.53 | 7.33 ± 0.21 |
| | PVI | 2.53 ± 0.21 | 0.39 ± 0.02 | 22.42 ± 1.06 | 15.49 ± 0.44 | 7.24 ± 0.20 |
| ResNet101, CIFAR-100 | MSP | **4.76 ± 0.28** | 1.70 ± 0.16 | **6.63 ± 0.46** | **91.24 ± 1.78** | **31.56 ± 0.59** |
| | PSI | 5.73 ± 0.32 | **0.80 ± 0.10** | 10.75 ± 0.41 | 102.67 ± 1.65 | 33.14 ± 0.53 |
| | PVI | 10.29 ± 0.56 | 2.56 ± 0.15 | 20.99 ± 0.72 | 91.24 ± 1.78 | 31.56 ± 0.59 |
| InceptionV3, Stanford Dogs | MSP | **4.80 ± 0.19** | 1.66 ± 0.14 | **5.87 ± 0.39** | **95.27 ± 2.44** | **34.12 ± 0.69** |
| | PSI | 6.84 ± 0.62 | **1.11 ± 0.22** | 11.83 ± 0.71 | 112.53 ± 1.73 | 36.71 ± 0.51 |
| | PVI | 8.79 ± 0.63 | 1.78 ± 0.20 | 15.27 ± 1.07 | 95.27 ± 2.44 | 34.12 ± 0.69 |
| DenseNet121, TinyImageNet | MSP | **3.45 ± 0.51** | 1.04 ± 0.24 | **4.03 ± 0.61** | **125.17 ± 0.95** | **38.62 ± 0.20** |
| | PSI | 6.73 ± 0.46 | **1.03 ± 0.12** | 9.87 ± 0.45 | 147.91 ± 2.01 | 41.53 ± 0.37 |
| | PVI | 9.80 ± 0.28 | 2.60 ± 0.14 | 17.47 ± 0.33 | 125.05 ± 0.98 | 38.58 ± 0.22 |

# D. Additional Experiments

## D.1. Normalization: Effects of Softmax and Temperature Scaling

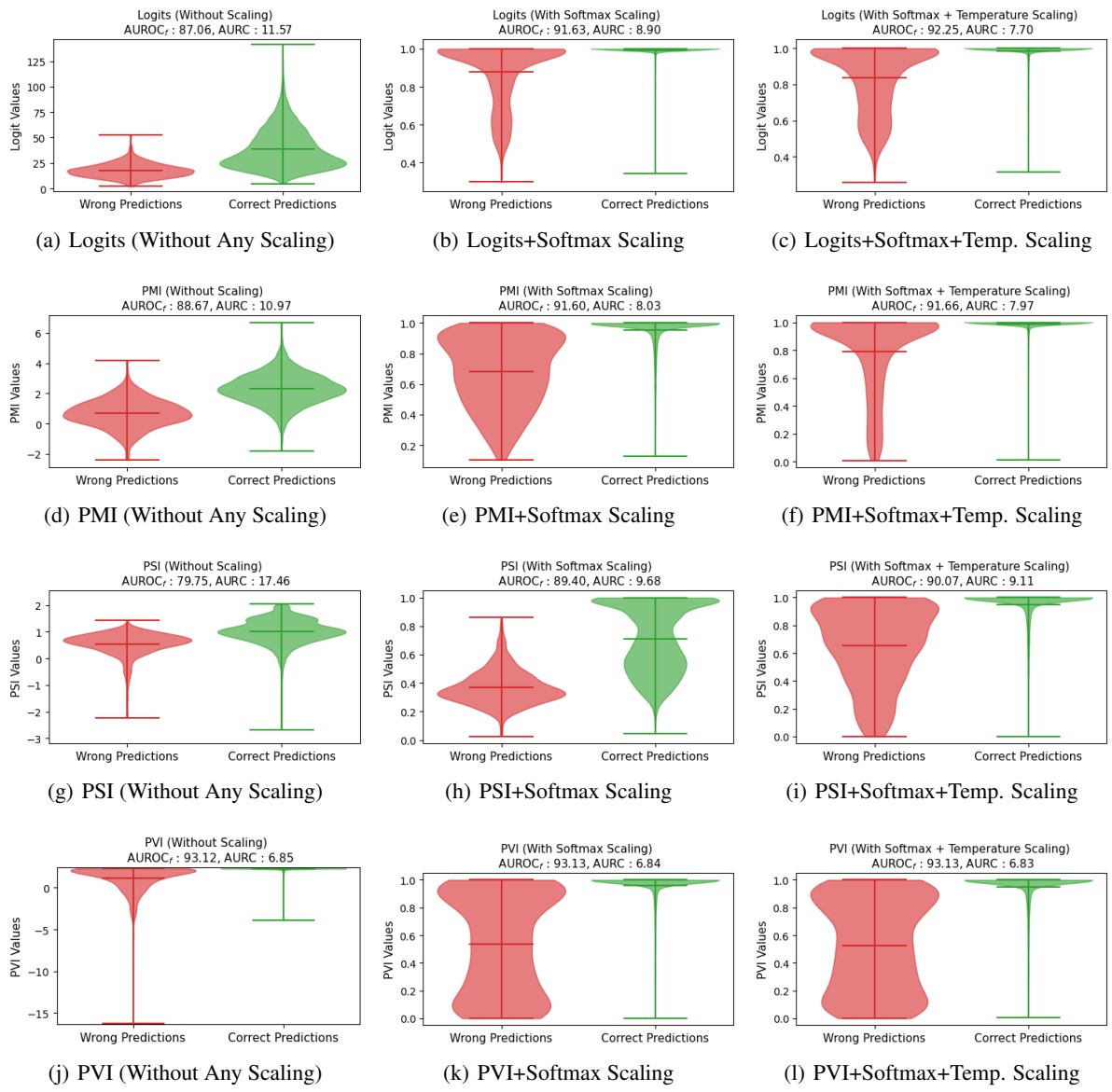

Figure 2: The distributions of confidence values estimated using logits (figures a-c), PMI (figures d-f), PSI (figures g-i), and PVI (figures j-l) for incorrect and correct test predictions (model: CNN, dataset: Fashion MNIST). First column (left figures): raw values; Second column (middle figures): with softmax scaling; Third column (right figures): with softmax and temperature scaling. The respective AUROC (scaled by 100, higher is better) and AURC (scaled by 1000, lower is better) are also reported.

In this section, we analyze the effects of softmax and temperature scaling on the raw (unnormalized) confidence values. For a given vector of unnormalized confidence values $\boldsymbol{\tau}$ of length $K$ (number of classes), the softmax function $\sigma$ is given by:

$$\sigma_k(\boldsymbol{\tau}) = \frac{e^{\tau_k}}{\sum_{j=1}^{K} e^{\tau_j}} \tag{98}$$

where $\sigma_k(\boldsymbol{\tau})$ denotes the $k$-th element of $\sigma(\boldsymbol{\tau})$.

The softmax function with temperature scaling is:

$$\sigma_k(\boldsymbol{\tau}, T) = \frac{e^{\tau_k/T}}{\sum_{j=1}^{K} e^{\tau_j/T}} \tag{99}$$

where $T$ is the temperature parameter. By adjusting the temperature $T$, we can control the sharpness or smoothness of the resulting probability distribution. When $T = 1$, the temperature-scaled softmax reduces to the standard softmax function. Since the same $T$ is used for all classes, it does not change the maximum of the softmax function, which means that the predictions of the network remain the same. To obtain the optimal temperature for a trained network, we select the temperature from the range 0.01, 0.02, ..., 4.99, 5.00 that maximizes the AURC on the validation dataset.

We compare the effects of softmax and temperature scaling for the four approaches (trained network logits, PMI, PSI, and PVI). We show the results for CNN model with Fashion MNIST dataset in Figure 2 The figure shows the violin plots of the confidence scores for wrong and correct predictions. We also report the AUROC and AURC for each method in the respective figure.

**Takeaway.** Without any scaling, the raw confidence estimates can span a wide range of values, resulting in a distribution that may be highly skewed or exhibit large variance, with significant overlap between the distributions of wrong and correct predictions. Applying softmax normalizes the logits into a probability distribution, which transforms the range of values and adjusts the distribution, often leading to more concentrated confidence scores for correct predictions at high values, while wrong predictions tend to have a broader distribution. This can help reduce the overlap between the two distributions. Temperature scaling further increases this separation by either compressing the confidence scores of correct predictions or broadening those of wrong predictions.

### D.2. Comparison of Various Pointwise Information Estimators

In this section, we compare the different methods of estimating each PI measure to obtain the best results. For comparison, we report the results for MLP trained with MNIST dataset as well as CNN trained with Fashion MNIST dataset for 5 runs. The hyperparameters for the training are reported in Appendix C.2 and the model classification errors are reported in Table 7. To show the improvement of these estimators, we also include the results for softmax (without temperature scaling).

D.2.1. COMPARISON OF PMI ESTIMATORS

For this experiment, we consider two different types of critic design: joint critic and separable critic. We also consider the three estimators: probabilistic classifier, density ratio fitting and variational JS bound. More details on these critic designs and estimators can be found in Appendix A.3.1. We first look at the convergence behaviour of these estimators by computing the $I(T; \hat{Y})$ where $T$ is the penultimate layer for MLP model trained on MNIST dataset. We train each critic model for 100 epochs with batch size of 512 and Adam optimizer (with learning rate of 0.001). We present the results (averaged over 5 runs) in Figure 3, with the shaded regions representing the standard deviations. We observe that the probabilistic classifier estimator converges more slowly and exhibits higher variance compared to the other two estimators. As a result, we exclude it from subsequent comparisons.

We then evaluate the performance of confidence estimates returned by both the density ratio fitting and variational JS bound estimators (using both joint and separable critics) on MLP and MNIST, as well as CNN and Fashion MNIST. The confidence ranking metrics are $\text{AUROC}_f$ and AURC. In addition, we assess their performance based on whether softmax scaling is used and whether confidence estimates are derived from the penultimate layer features or output layer features (before the softmax function). We report the results in Table 9.

**Takeaway.** We observe that using the output layer features, rather than the penultimate layer features, yields significantly better results. Additionally, applying softmax scaling enhances the performance of the variational JS bound estimator but degrades the results for the density ratio fitting estimator. Among the estimators, the variational JS bound estimator, combined with softmax scaling, surpasses the density ratio fitting estimator. Regarding critic design, the separable critic slightly outperforms the joint critic. We find that the best configuration includes using **output layer features** with a **separable critic** and the **variational JS bound estimator** with **softmax scaling**, which we adopt for all subsequent experiments.

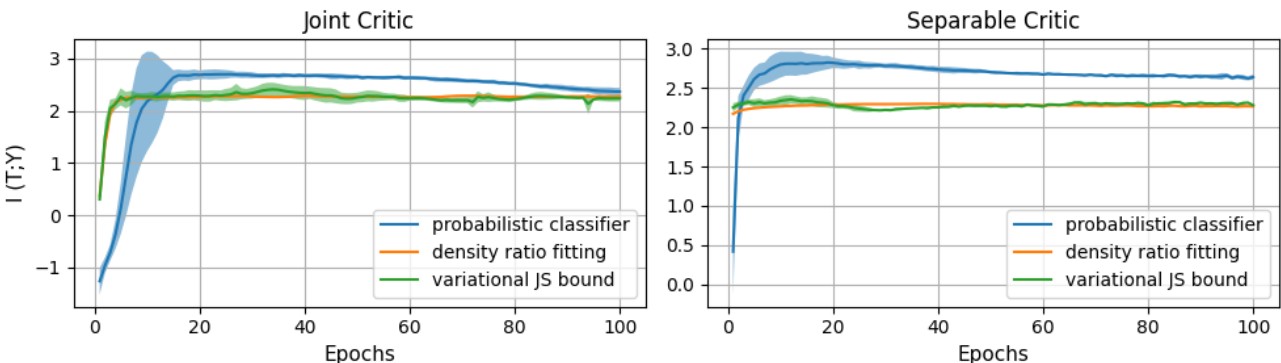

Figure 3: Estimation of $I(T; \hat{Y})$ where $T$ is the penultimate layer for the MLP model trained on the MNIST dataset. Three estimators are considered: probabilistic classifier (blue line), density ratio fitting (orange line), and variational JS bound (green line). Two critic designs are considered: joint (left) and separable (right). Here, the epochs refer to the training of the critic, not the original network. The shaded regions represent the standard deviations.

Table 9: Comparison of Different PMI Estimators (Averaged over 5 Runs with Standard Deviations Included). The best results are highlighted in bold.

| Critic, Estimator | AUROC$_f \times 10^2 \uparrow$ | | AURC $\times 10^3 \downarrow$ | |
|---|---|---|---|---|
| | MLP, MNIST | CNN, F-MNIST | MLP, MNIST | CNN, F-MNIST |
| Without Softmax Scaling, Penultimate Layer | | | | |
| Joint Critic, Density Ratio Fitting | $90.47 \pm 1.16$ | $89.09 \pm 0.96$ | $2.54 \pm 0.35$ | $11.81 \pm 1.37$ |
| Joint Critic, Variational JS Bound | $87.30 \pm 1.38$ | $78.53 \pm 1.16$ | $3.58 \pm 0.51$ | $25.89 \pm 3.77$ |
| Separable Critic, Density Ratio Fitting | $78.51 \pm 2.43$ | $85.09 \pm 1.04$ | $8.24 \pm 1.73$ | $16.94 \pm 1.61$ |
| Separable Critic, Variational JS Bound | $76.45 \pm 3.08$ | $85.56 \pm 0.81$ | $6.27 \pm 0.88$ | $16.10 \pm 1.18$ |
| With Softmax Scaling, Penultimate Layer | | | | |
| Joint Critic, Density Ratio Fitting | $89.35 \pm 1.87$ | $79.00 \pm 3.38$ | $2.62 \pm 0.66$ | $22.14 \pm 3.93$ |
| Joint Critic, Variational JS Bound | $85.51 \pm 5.38$ | $90.31 \pm 0.46$ | $4.88 \pm 2.44$ | $10.19 \pm 0.48$ |
| Separable Critic, Density Ratio Fitting | $89.02 \pm 1.79$ | $87.45 \pm 0.82$ | $3.14 \pm 0.94$ | $15.11 \pm 1.33$ |
| Separable Critic, Variational JS Bound | $90.12 \pm 1.44$ | $91.67 \pm 0.28$ | $2.55 \pm 0.52$ | $8.40 \pm 0.16$ |
| Without Softmax Scaling, Output Layer | | | | |
| Joint Critic, Density Ratio Fitting | $95.36 \pm 0.38$ | $91.49 \pm 0.39$ | $1.01 \pm 0.09$ | $8.68 \pm 0.44$ |
| Joint Critic, Variational JS Bound | $91.63 \pm 0.62$ | $87.39 \pm 1.01$ | $2.19 \pm 0.34$ | $13.95 \pm 1.82$ |
| Separable Critic, Density Ratio Fitting | $95.55 \pm 0.59$ | $86.65 \pm 0.61$ | $1.03 \pm 0.20$ | $14.55 \pm 0.93$ |
| Separable Critic, Variational JS Bound | $92.95 \pm 1.94$ | $88.22 \pm 0.62$ | $1.57 \pm 0.45$ | $12.00 \pm 0.81$ |
| With Softmax Scaling, Output Layer | | | | |
| Joint Critic, Density Ratio Fitting | $93.32 \pm 1.42$ | $87.37 \pm 1.25$ | $1.441 \pm 0.377$ | $13.39 \pm 1.59$ |
| Joint Critic, Variational JS Bound | $\mathbf{97.35 \pm 0.36}$ | $\mathbf{91.54 \pm 0.22}$ | $\mathbf{0.57 \pm 0.08}$ | $\mathbf{8.52 \pm 0.43}$ |
| Separable Critic, Density Ratio Fitting | $\mathbf{97.13 \pm 0.33}$ | $88.44 \pm 0.55$ | $\mathbf{0.67 \pm 0.12}$ | $13.94 \pm 0.73$ |
| Separable Critic, Variational JS Bound | $\mathbf{97.24 \pm 0.18}$ | $\mathbf{91.97 \pm 0.35}$ | $\mathbf{0.57 \pm 0.05}$ | $\mathbf{8.11 \pm 0.09}$ |
| Softmax | $95.11 \pm 0.48$ | $92.03 \pm 0.23$ | $1.38 \pm 0.16$ | $8.75 \pm 0.34$ |

### D.2.2. COMPARISON OF PSI ESTIMATORS

For this experiment, we consider the two methods: binning and Gaussian described in Section A.3.2. First, we validate the accuracy of these estimators by comparing their estimates with those obtained using the KSG estimator (Kraskov et al., 2004). Note that SMI is the average of PSI over all samples. We compute the SMI between the penultimate layer and the predicted labels during training (100 epochs) for MLP model and MNIST validation dataset. We use 500 projections for both SMI and PSI estimation and 20 bins for the binning method. The results are shown in Figure 4. We observed that the SMI estimates derived from the PSI binning method align more closely with the direct SMI estimates from the KSG estimator than those from the Gaussian method. However, both methods exhibit the same overall trend.

We then evaluate the performance of confidence estimates returned by both the binning and Gaussian estimators (with different number of projections $m$) on MLP and MNIST, as well as CNN and Fashion MNIST. The confidence ranking

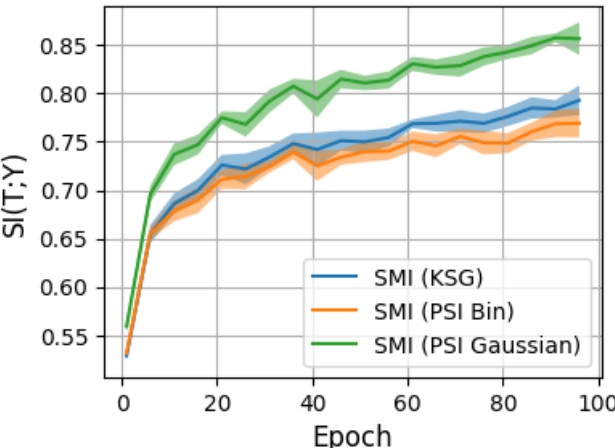

Figure 4: SMI between penultimate layer and predicted labels during training. The plot shows the average SMI over 5 runs for three different estimation methods (KSG, PSI Bin, and PSI Gaussian) across epochs. The shaded regions represent the 95% confidence intervals.

metrics are AUROC$_f$ and AURC. In addition, we assess their performance based on whether softmax scaling is used and whether confidence estimates are derived from the penultimate layer features or output layer features (before the softmax function). We report the results in Table 10.

**Takeaway.** We observe that using the output layer features, rather than the penultimate layer features, yields significantly better results. We observe that the Gaussian method yields poor performance for the CNN with Fashion MNIST case, but this could be remedied by a simple softmax scaling. We also observe that increasing the number of projections $m$ beyond 500 leads to little or no improvement in the results. We find that the best configuration includes using **output layer features** with the **Gaussian** estimator and **softmax scaling**, which we adopt for all subsequent experiments.

### D.2.3. COMPARISON OF PVI ESTIMATORS

We evaluate the performance of confidence estimates returned by the various PVI estimation methods described in Section A.3.3 on MLP and MNIST, as well as CNN and Fashion MNIST. The confidence ranking metrics are AUROC$_f$ and AURC. In addition, we also consider calibrating the softmax probabilities used to compute the PVI. Similar to PMI and PSI, we assess if the performance improves with softmax scaling. We report the results in Table 10.

In addition, we assess their performance based on whether softmax scaling is used and whether the associated probabilities are calibrated with temperature scaling before computing the PVI.

**Takeaway.** We find that calibrating the softmax probabilities before computing PVI, along with applying softmax scaling, significantly improves performance. The best result is achieved by "training from scratch," which means using another trained network with a different initialization. We use this as the default estimator for PVI in all experiments.

Table 10: Comparison of Different PSI Estimators (Averaged over 5 Runs with Standard Deviations Included). The best results are highlighted in bold.

| Estimator | $\text{AUROC}_f \times 10^2 \uparrow$ | | $\text{AURC} \times 10^3 \downarrow$ | |
|---|---|---|---|---|
| | MLP, MNIST | CNN, F-MNIST | MLP, MNIST | CNN, F-MNIST |
| Without Softmax Scaling, Penultimate Layer | | | | |
| Binning ($m = 250$) | $96.85 \pm 0.25$ | $85.82 \pm 0.33$ | $0.63 \pm 0.06$ | $12.78 \pm 0.25$ |
| Binning ($m = 500$) | $96.89 \pm 0.12$ | $86.13 \pm 0.48$ | $0.62 \pm 0.03$ | $12.55 \pm 0.34$ |
| Binning ($m = 750$) | $96.92 \pm 0.18$ | $86.11 \pm 0.51$ | $0.61 \pm 0.04$ | $12.56 \pm 0.10$ |
| Binning ($m = 1000$) | $96.90 \pm 0.15$ | $86.19 \pm 0.39$ | $0.62 \pm 0.03$ | $12.49 \pm 0.25$ |
| Gaussian ($m = 250$) | $96.38 \pm 0.40$ | $81.74 \pm 0.68$ | $0.71 \pm 0.09$ | $16.20 \pm 0.40$ |
| Gaussian ($m = 500$) | $96.46 \pm 0.32$ | $82.13 \pm 0.87$ | $0.69 \pm 0.07$ | $15.90 \pm 0.39$ |
| Gaussian ($m = 750$) | $96.45 \pm 0.25$ | $82.12 \pm 0.89$ | $0.69 \pm 0.05$ | $15.87 \pm 0.40$ |
| Gaussian ($m = 1000$) | $96.43 \pm 0.28$ | $82.21 \pm 0.72$ | $0.70 \pm 0.06$ | $15.80 \pm 0.27$ |
| With Softmax Scaling, Penultimate Layer | | | | |
| Binning ($m = 250$) | $96.08 \pm 0.54$ | $88.11 \pm 0.12$ | $0.78 \pm 0.14$ | $10.94 \pm 0.33$ |
| Binning ($m = 500$) | $96.89 \pm 0.12$ | $88.34 \pm 0.26$ | $0.73 \pm 0.08$ | $10.76 \pm 0.34$ |
| Binning ($m = 750$) | $96.28 \pm 0.26$ | $88.35 \pm 0.30$ | $0.73 \pm 0.06$ | $10.73 \pm 0.19$ |
| Binning ($m = 1000$) | $96.17 \pm 0.32$ | $88.47 \pm 0.19$ | $0.76 \pm 0.08$ | $10.65 \pm 0.31$ |
| Gaussian ($m = 250$) | $96.05 \pm 0.28$ | $88.29 \pm 0.22$ | $0.79 \pm 0.06$ | $10.79 \pm 0.34$ |
| Gaussian ($m = 500$) | $95.95 \pm 0.16$ | $88.23 \pm 0.49$ | $0.81 \pm 0.04$ | $10.84 \pm 0.39$ |
| Gaussian ($m = 750$) | $96.00 \pm 0.21$ | $88.45 \pm 0.35$ | $0.81 \pm 0.05$ | $10.66 \pm 0.26$ |
| Gaussian ($m = 1000$) | $96.07 \pm 0.18$ | $88.46 \pm 0.39$ | $0.79 \pm 0.03$ | $10.65 \pm 0.34$ |
| Without Softmax Scaling, Output Layer | | | | |
| Binning ($m = 250$) | $\mathbf{97.01 \pm 0.11}$ | $85.88 \pm 0.75$ | $\mathbf{0.60 \pm 0.03}$ | $12.89 \pm 0.24$ |
| Binning ($m = 500$) | $\mathbf{97.05 \pm 0.20}$ | $85.77 \pm 0.68$ | $\mathbf{0.59 \pm 0.05}$ | $13.00 \pm 0.25$ |
| Binning ($m = 750$) | $\mathbf{97.05 \pm 0.13}$ | $85.96 \pm 0.66$ | $\mathbf{0.60 \pm 0.03}$ | $12.84 \pm 0.14$ |
| Binning ($m = 1000$) | $\mathbf{97.06 \pm 0.11}$ | $85.87 \pm 0.75$ | $\mathbf{0.59 \pm 0.03}$ | $12.91 \pm 0.25$ |
| Gaussian ($m = 250$) | $96.68 \pm 0.23$ | $80.40 \pm 1.02$ | $0.66 \pm 0.05$ | $17.62 \pm 0.67$ |
| Gaussian ($m = 500$) | $96.73 \pm 0.30$ | $80.38 \pm 0.69$ | $0.65 \pm 0.07$ | $17.63 \pm 0.58$ |
| Gaussian ($m = 750$) | $96.72 \pm 0.26$ | $80.55 \pm 0.62$ | $0.65 \pm 0.06$ | $17.46 \pm 0.31$ |
| Gaussian ($m = 1000$) | $96.71 \pm 0.23$ | $80.54 \pm 0.63$ | $0.66 \pm 0.05$ | $17.47 \pm 0.36$ |
| With Softmax Scaling, Output Layer | | | | |
| Binning ($m = 250$) | $96.15 \pm 0.43$ | $85.96 \pm 1.06$ | $0.79 \pm 0.11$ | $13.31 \pm 0.84$ |
| Binning ($m = 500$) | $96.24 \pm 0.52$ | $85.89 \pm 0.86$ | $0.76 \pm 0.13$ | $13.38 \pm 0.67$ |
| Binning ($m = 750$) | $96.31 \pm 0.39$ | $86.23 \pm 0.85$ | $0.75 \pm 0.10$ | $13.10 \pm 0.69$ |
| Binning ($m = 1000$) | $96.31 \pm 0.41$ | $86.12 \pm 0.94$ | $0.75 \pm 0.10$ | $13.19 \pm 0.70$ |
| Gaussian ($m = 250$) | $96.58 \pm 0.21$ | $\mathbf{89.46 \pm 0.46}$ | $0.69 \pm 0.05$ | $\mathbf{9.97 \pm 0.23}$ |
| Gaussian ($m = 500$) | $96.61 \pm 0.18$ | $\mathbf{89.42 \pm 0.42}$ | $0.69 \pm 0.04$ | $\mathbf{10.00 \pm 0.25}$ |
| Gaussian ($m = 750$) | $96.59 \pm 0.17$ | $\mathbf{89.44 \pm 0.49}$ | $0.69 \pm 0.04$ | $\mathbf{10.00 \pm 0.25}$ |
| Gaussian ($m = 1000$) | $96.62 \pm 0.22$ | $\mathbf{89.36 \pm 0.50}$ | $0.68 \pm 0.05$ | $\mathbf{10.05 \pm 0.23}$ |
| Softmax | $95.11 \pm 0.48$ | $92.03 \pm 0.23$ | $1.38 \pm 0.16$ | $8.75 \pm 0.34$ |

Table 11: Comparison of Different PVI Estimators (Averaged over 5 Runs with Standard Deviations Included). The best results are highlighted in bold.

| Estimator | $\text{AUROC}_f \times 10^2 \uparrow$ | | $\text{AURC} \times 10^3 \downarrow$ | |
|---|---|---|---|---|
| | MLP, MNIST | CNN, F-MNIST | MLP, MNIST | CNN, F-MNIST |
| *Uncalibrated, Without Softmax Scaling* | | | | |
| No training | $75.65 \pm 1.93$ | $83.11 \pm 0.77$ | $6.33 \pm 0.56$ | $22.16 \pm 0.26$ |
| Training from scratch | $82.79 \pm 1.20$ | $89.07 \pm 0.06$ | $4.52 \pm 0.42$ | $12.07 \pm 0.48$ |
| Training MLP penultimate | $65.45 \pm 1.23$ | $70.53 \pm 0.61$ | $9.39 \pm 0.44$ | $32.45 \pm 1.42$ |
| *Uncalibrated, With Softmax Scaling* | | | | |
| No training | $95.12 \pm 0.48$ | $92.03 \pm 0.23$ | $1.38 \pm 0.16$ | $8.75 \pm 0.34$ |
| Training from scratch | $95.85 \pm 0.43$ | $92.75 \pm 0.28$ | $1.19 \pm 0.14$ | $8.24 \pm 0.37$ |
| Training MLP penultimate | $88.31 \pm 1.37$ | $85.20 \pm 0.38$ | $3.56 \pm 0.45$ | $19.51 \pm 0.44$ |
| *Calibrated, Without Softmax Scaling* | | | | |
| No training | $88.13 \pm 0.79$ | $92.37 \pm 0.19$ | $2.82 \pm 0.29$ | $8.10 \pm 0.18$ |
| Training from scratch | $90.76 \pm 1.06$ | $93.28 \pm 0.26$ | $2.20 \pm 0.22$ | $7.10 \pm 0.20$ |
| Training MLP penultimate | $80.57 \pm 1.73$ | $84.51 \pm 0.31$ | $5.09 \pm 0.48$ | $17.13 \pm 0.55$ |
| *Calibrated, With Softmax Scaling* | | | | |
| No training | $97.12 \pm 0.23$ | $92.68 \pm 0.22$ | $0.60 \pm 0.04$ | $7.43 \pm 0.17$ |
| Training from scratch | $\mathbf{97.53 \pm 0.23}$ | $\mathbf{93.33 \pm 0.25}$ | $\mathbf{0.54 \pm 0.03}$ | $\mathbf{6.99 \pm 0.15}$ |
| Training MLP penultimate | $96.84 \pm 0.27$ | $91.82 \pm 0.20$ | $0.72 \pm 0.06$ | $8.29 \pm 0.18$ |
| Softmax | $95.11 \pm 0.48$ | $92.03 \pm 0.23$ | $1.38 \pm 0.16$ | $8.75 \pm 0.34$ |

