# OpenReview forum: "Pointwise Information Measures as Confidence Estimators in Deep Neural Networks: A Comparative Study"
_ICML.cc/2025/Conference — ICML 2025 poster_

### Official Review · Reviewer_Wh94 · 2025-03-11

**Overall Recommendation:** 4

**Summary:**

This paper presents a comparative analysis of three point-wise information measures—PMI, PVI, and PSI—from both theoretical and empirical perspectives. The study theoretically analyzes the sensitivity of these measures to margin effects and intrinsic dimensionality, along with their convergence rates. Empirical validation is conducted through numerical experiments using benchmark computer vision models and datasets, evaluating the measures' performance in failure prediction and confidence calibration tasks.

## Update after rebuttal

I keep my score unchanged, and I think the authors addressed my concerns well.

**Claims And Evidence:**

The claim that PVI is the "most well-rounded" measure cannot be justified solely by analyzing invariance and margin sensitivity. By definition, PVI critically depends on the predictive family $\mathcal{V}$. Improper specification of $\mathcal{V}$ may significantly degrade PVI's performance, necessitating rigorous theoretical and empirical investigation into $\mathcal{V}$'s role. While Tables 2-3 demonstrate PVI's superiority over post-hoc baselines, two critical questions remain unresolved:

1. How severely does misspecification of $\mathcal{V}$ impact PVI's effectiveness?

2. What principles should guide the selection of $\mathcal{V}$ for optimal PVI performance?

**Essential References Not Discussed:**

None.

**Experimental Designs Or Analyses:**

The experimental design and analytical validity were checked. The experiments in the paper properly evaluated PMI, PSI, PVI, and other post-hoc methods under controlled experimental parameters. Detailed supplementary documentation delineates performance variations across operational conditions for PMI, PSI, and PVI. Despite this, further controlled ablation studies remain necessary to elucidate the relationship between PVI and the predictive family $\mathcal{V}$.

**Methods And Evaluation Criteria:**

While the proposed experimental framework appears methodologically sound, three critical enhancements are required:

1.	Systematic investigation of PVI's dependence on $\mathcal{V}$ across parameter configurations.

2.	Quantitative benchmarking against baseline methods under controlled $\mathcal{V}$ variations.

3.	Causal analysis elucidating why PVI outperforms post-hoc alternatives in specific $\mathcal{V}$ regimes.

**Other Comments Or Suggestions:**

Typos:

1. Lines 066-069 in right column:  By doing so, .... in the data.
2. line 1212: sa1me

**Other Strengths And Weaknesses:**

Strengths:

1.	The comparison of point-wise versions of MI, SI, and VI provides an insightful and significant contribution to quantifying uncertainty in deep learning models. The paper offers both theoretical and empirical comparisons of these measures.
2. The paper demonstrates that PVI outperforms other post-hoc methods in most failure prediction and confidence calibration scenarios. This suggests PVI could be a promising approach for exploring adversarial robustness.

Weaknesses:

1. The theoretical framework requires further development by relaxing the stringent assumption in Section 3.1 that requires $\mathcal{V}$ to be a fully connected neural network.
2. The role of $\mathcal{V}$ warrants more rigorous theoretical analysis and empirical ablation investigation.

**Questions For Authors:**

1.	To what extent does misspecification of $\mathcal{V}$ influence the efficacy of PVI?
2.	What guiding principles should inform the selection of $\mathcal{V}$ to optimize PVI performance?
3.	In Table 2's third numerical column, why does PVI exhibit consistently larger confidence intervals when AUPR$_{f,error}$ serves as the evaluation metric?

**Relation To Broader Scientific Literature:**

The study systematically compared three uncertainty measures (PMI, PSI, and PVI) through theoretical analysis and empirical validation, demonstrating PVI's promising performance in uncertainty quantification for deep learning tasks. These findings not only advance key applications like failure prediction and misclassification detection, but also provide novel insights for enhancing model robustness against adversarial attacks.

**Theoretical Claims:**

I have reviewed the proofs provided in the Supplementary Materials. The proofs demonstrate scientific validity under the specified theoretical assumptions outlined in this study.

---

> ### Author Rebuttal · Authors · 2025-04-01
>
> We thank the reviewer for their positive remarks and valuable suggestions on the paper. Below, we address the concerns and questions raised by the reviewer.
>
> **Weakness 1 (On the Assumption for PVI's Theoretical Properties)**
>
> This is a valid point, and we will explicitly specify the limitations of this assumption for our theoretical results relating to PVI. However, having said that, we note that our results can be extended to non fully-connected networks by considering the input as the feature layer that eventually feeds into a fully-connected architecture. For instance, we can consider the feature output after the convolution layers in a CNN, which is input to a fully connected set of layers before yielding the final output. For those cases, our results are still applicable.
>
> **Question 1 (On the Misspecification of $\mathcal{V}$)**
>
> Yes, misspecification of $\mathcal{V}$ can reduce the applicability of both the theoretical and empirical results in our work. However, in our work we assume that the knowledge of the model’s architecture is available. In the cases when the model is entirely a black box, we won’t be able to arrive at our theoretical and empirical results in this paper. To that end, we are including additional experiments where we either over or under-specify the complexity of $\mathcal{V}$, and see how it affects the results.
>
> **Question 2 (On the Selection of $\mathcal{V}$)**
>
> Ideally, $\mathcal{V}$ should match the architecture of the model used to generate the predicted outputs. This alignment ensures that the PVI estimation accurately reflects the confidence characteristics of the original model.
>
> **Question 3 (On the $AUPR_{f,error}$)**
>
> Since $AUPR_{f,error}$ emphasizes the accurate ranking of rare error cases (i.e., treating incorrect predictions as positives and using low confidence scores as indicators), methods that can precisely distinguish which specific samples are likely to be errors will naturally perform better. It is essentially asking “How well can we rank and detect errors using the model’s uncertainty/confidence score?” PVI outperforms others for this metric reflects its ability to identify misclassification, even when errors are rare. We hypothesize that this may be attributed to PVI's incorporation of both the network architecture and the prior probabilities of the output classes, which helps mitigate overconfidence on misclassified samples.

---

### Official Review · Reviewer_Kkv7 · 2025-03-12

**Overall Recommendation:** 2

**Summary:**

The authors propose a new indicator for quantifying the confidence of a neural network: the pointwise information between an input’s features and the corresponding output. When the information is high (corresponding to a large reduction in the entropy of the true label when conditioned on the input), we expect that the input features are especially useful for making predictions (thus confidence should be higher).

Three measures of pointwise information are compared: mutual information, V-information, and sliced mutual information. The authors first investigate these measures along the dimensions of transformation invariance, sensitivity to geometric properties of the feature distribution, and the convergence rates of their estimators. They then conduct a series of experiments that demonstrate the usefulness of these PI measures for failure/selective prediction and confidence calibration.

### Update after rebuttal
My main concerns about this work still remain — namely, that the empirical results are not strongly motivated or justified by the theoretical discussion (as currently presented), and that the experiments in their current form are not convincing enough for practitioners to consider adoption of this method.

This paper would be significantly strengthened by the use of more-realistic datasets (not toy ones like MNIST, Fashion-MNIST, CIFAR-10, etc.), especially if such experiments can demonstrate a clear and consistent advantage to PI estimation methods over existing UQ techniques. I maintain my score.

**Claims And Evidence:**

The authors claim that pointwise information (PI) is a useful indicator of predictive uncertainty and that PVI outperforms other PI estimation methods and UQ benchmarks. While experiments offer partial support, the connection between empirical evidence and theoretical analysis is poorly demonstrated.

In Section 3, invariance is emphasized as crucial for confidence estimation, yet PMI—the most invariant method—does not perform best. The justification for this (that excessive invariance may be counterproductive) is unconvincing and inconsistent. Remark 1 states that “it is important to be invariant to bijective transformations T in the context of confidence estimation”, and that the “ideal scenario is when the above is true for any invertible, and thus information-preserving transformation T”. However, Remark 9 suggests “the fact that PMI is invariant to a much larger degree of non-linear homeomorphisms may not always be advantageous”. These claims directly contradict each other. If invariance matters for confidence estimation, why does it not lead to observably better empirical performance? I would have appreciated an explicit demonstration of this principle, perhaps in an experiment where each method is applied to intentionally transformed data and results are compared.

The discussion of sample-wise margin (which seems to refer to class separation in feature space) lacks a unifying theory across PMI, PSI, and PVI, making Propositions 4 & 5 and Theorem 1 difficult to contextualize. Consequently, takeaway T3 appears unconvincing. Moreover, the correlation-to-margin experiment, combined with the results of Section 4, suggests that margin sensitivity does not directly drive confidence estimation performance, calling into question its relevance to the main analysis. This section might be better placed in the appendix.

Takeaway T4, on convergence rates, is the most practically relevant, as a PI estimator’s performance should correlate with the usefulness of its information scores for UQ. However, the theoretical discussion is buried in the appendix—it should be in the main body.

Section 4 presents mixed empirical results, with no method universally superior (as evidenced by the plethora of statistical ties in Tables 2 and 3). While PVI is the most consistent top performer, it is not dominant. The claim that PVI is “the most well-rounded” across invariance, margin sensitivity, and convergence rate is unconvincing—these factors are neither comprehensive nor clearly predictive of empirical success. Theoretical takeaways seem driven by empirical results rather than the other way around, weakening the paper’s core claims.

Overall, while the paper presents interesting theoretical insights, the lack of a clear and consistent connection between theory and empirical results weakens its central claims. Key theoretical principles, such as invariance and margin sensitivity, are not convincingly linked to empirical performance, and in some cases, the arguments appear contradictory. Additionally, the empirical results do not strongly support the claim that PVI is the best method, as its advantages seem marginal rather than definitive. A more thorough justification of the theoretical takeaways, along with targeted experiments to validate key assumptions, would significantly strengthen the paper’s contributions.

**Essential References Not Discussed:**

None that I am aware of.

**Experimental Designs Or Analyses:**

The experimental designs appear sound, but the analysis in Section 4 offers little beyond summarizing tables in words. Several claims also warrant scrutiny:

- For F-MNIST, the authors attribute PVI’s superiority over PMI and PSI to its “well-rounded” nature, citing invariance and margin sensitivity. However, it’s unclear why this explanation applies specifically to F-MNIST but not to CIFAR-10, where PVI and PSI are statistically tied, or to MNIST, where PVI and PMI perform similarly.
- In Section 4.2, the claim that “PVI significantly outperforms [all benchmarks] when assessing the average ECE… by a large amount” is questionable. While the means in Table 3 support this, the large standard deviations undermine the statistical significance of the difference.
- On page 8, the authors assert that “for average ECE, it seems that the improvement [of PVI over PMI and PSI] for more complex datasets and architectures is more significant.” This claim would be far more convincing if PMI/PSI results were reported for Tiny-ImageNet and DS-ImageNet. Without them, the argument for PVI’s superiority on complex datasets remains unsubstantiated.

**Methods And Evaluation Criteria:**

I am concerned with the authors’ use of ECE for calibration, given its known issues with discontinuity [1] and binning dependence [2].

The absence of transformer-based models leaves open the question of whether the reported PI method improvements hold for more modern architectures. This is particularly relevant since transformers' softmax probabilities have been shown to calibrate better than those of ConvNets [3].

The dataset choices are also underwhelming. The lack of standard-resolution (224x224) images raises concerns about scalability, as results on toy datasets like MNIST and F-MNIST may not generalize to realistic settings. It is disappointing that the largest datasets (Tiny-ImageNet, DS-ImageNet) were relegated to the appendix, with DS-ImageNet seemingly missing benchmarks for several algorithms.

[1] Błasiok, J., Gopalan, P., Hu, L., & Nakkiran, P. (2023, June). A unifying theory of distance from calibration. In Proceedings of the 55th Annual ACM Symposium on Theory of Computing (pp. 1727-1740).
[2] Nixon, J., Dusenberry, M. W., Zhang, L., Jerfel, G., & Tran, D. (2019, June). Measuring Calibration in Deep Learning. In CVPR workshops (Vol. 2, No. 7).
[3] Minderer, M., Djolonga, J., Romijnders, R., Hubis, F., Zhai, X., Houlsby, N., ... & Lucic, M. (2021). Revisiting the calibration of modern neural networks. Advances in neural information processing systems, 34, 15682-15694.

**Other Comments Or Suggestions:**

It is atypical to see the ECE reported in (what I think) are 100x amounts. ECE is typically bounded by [0, 1].

Table 9 reports exceptionally high error rates for ResNet101 trained on DS-ImageNet (80% train error, 85-86% generalization error). Are you sure that your models are adequately trained here? If not, we should question the validity of any confidence calibration results obtained on this dataset. I am also concerned about the 52% test error rate for DenseNet121 trained on Tiny ImageNet (while the validation error is 11%).

I noticed a few typos / grammatical errors while reading. Here is a list:
- Item 3 in “Motivation”, first paragraph: “We note that this measure is rooted in probability, and estimates priors and posterior probability measures” (did you mean to put prior here? The plural doesn’t make as much sense)
- Item 3 in “Motivation”, first paragraph: “This is unlike the typical neural network output, which, although *it* is supposed to model the conditional probabilities of each class p(y|x), often turn*s* out to be not a good indicator of the true uncertainty” (changes marked with stars)
- Item 3 in “Motivation”, first paragraph: “By doing so, *they* can potentially reduce inherent bias…” (changes marked with stars)
- Item 1 in “Contributions”: “We found that PVI outperforms PMI and PSI”. Did you mean to switch into past tense here?
- In equation 9, I believe you replaced $psi$ with $\psi$. You do not define $\psi$ anywhere in the paper.
- Item T3 in Section 3.3: “we see PMI to be invariant to hard margin, and yet PSI being sensitive to hard margin” (the italicized portion should be replaced with something like “while PSI is sensitive to hard margin”).

**Other Strengths And Weaknesses:**

The application of pointwise information (PI) for confidence estimation is a novel and promising research direction. I commend the authors for this innovative approach and hope future work will further validate their hypothesis.

However, the manuscript would benefit from improved clarity and coherence, particularly in its argumentative structure. The theoretical and empirical sections feel somewhat disjointed, with the developed theory not clearly informing or justifying the experimental results. As a result, the theoretical contributions appear underutilized. Since the empirical results are middling, this lessens the overall significance of the current contribution.

A key limitation not yet discussed is the computational cost of the proposed approach. The authors frame their method as a post-hoc alternative, emphasizing the practical challenges of modifying network architectures or retraining models. While technically post-hoc (in that it does not require retraining the original model), the proposed approach is highly computationally expensive. This is evident from the study’s restriction to smaller datasets and lower-resolution images: “we do not report the results [on Tiny -ImageNet and DS-ImageNet] for PMI and PSI as they are very computationally expensive for large-scale datasets”. If my understanding is correct, each PI estimate requires training a separate neural network—sometimes with the same architecture, which may not always be accessible in black-box settings. Given the mixed empirical results, the computational overhead may not be justified, raising concerns about the method’s practicality.

Overall, while the paper introduces an interesting application of PI for confidence estimation, its practical viability remains uncertain due to the high computational cost and the lack of a clear theoretical-to-empirical connection. Strengthening the coherence between these aspects and further justifying the trade-offs involved would significantly enhance the paper’s impact.

**Questions For Authors:**

1. The authors discuss the potential counterproductivity of excessive invariance, yet the theoretical importance of invariance is emphasized. Could the authors elaborate on how invariance relates to predictive performance?
2. In Section 4, the authors claim that the superior performance of PVI over PMI and PSI on F-MNIST is due to PVI’s “well-roundedness.” Why do the authors believe this characteristic is particularly beneficial for F-MNIST, and not for datasets like CIFAR-10 or MNIST, where PVI does not show the same degree of superiority?
3. The concept of "sample-wise margin" appears to be inconsistent across the methods (PMI, PSI, PVI). Can the authors provide further clarity on how this property should be interpreted or applied in practice, especially in light of the conflict between the correlation-to-margin results and Section 4?
4. Given the computational cost of PI methods, do the authors see any feasible ways to reduce the complexity of PI estimation while retaining its benefits?
5. Does PVI assume access to the model architecture, and if so, how does that align with the authors’ proposed use in black-box settings?

**Relation To Broader Scientific Literature:**

This study builds directly on Zhu et al. (2022), which demonstrated that common calibration techniques—such as label smoothing, mixup, focal loss, and temperature scaling—can inadvertently exacerbate the confidence gap between correct and incorrect predictions. Since these methods primarily operate by adjusting softmax probabilities, the authors explore whether alternative confidence estimation approaches, such as pointwise information (PI), can mitigate this issue. While the paper does not introduce new methods for estimating PI, its key contribution lies in the novel application of these techniques to confidence calibration and uncertainty quantification.

**Theoretical Claims:**

I did not check the correctness of any proofs.

---

> ### Author Rebuttal · Authors · 2025-04-01
>
> We thank the reviewer for their positive remarks and valuable suggestions on the paper. Below, we address the major concerns and questions raised by the reviewer. We will address all other points in the revision.
>
> **Question 1 (On Invariance)**
>
> We would like to clarify that we meant bijective linear transformations (in Remark 1). If the PI measures are not invariant to linear transformations, then it could pose an issue. We highlight this point with the following example. Let us consider PMI between a neural network layer $T$ and the output labels $Y$, and assume that $T’$ denotes another rendition of $T$ which has the same information but arises from a different initialization of the network. If the relationship between $T$ and $T’$ is linear, then PMI’s invariance is helpful. However, if the relationship is non-linear, as PMI’s invariance to non-linear invertible and continuous transformations also means that it outputs the same degree of uncertainty estimation when $T’$ is related to $T$ in a non-linear manner. If the function is highly non-linear, then the estimated label for $T’$ should have a different level of confidence compared to $T$, as the neural network’s remaining layers are limited in the ways it can transform the features $T’$ (considering finite networks). Therefore, in this case, the heavily invariant nature of PMI can be counterproductive for reflecting the uncertainty of predictions. Note that if PMI’s invariances were limited to linear transformations, this won’t be an issue.
>
> **Question 2 (On PVI Superiority)**
>
> For MNIST, where classifiers achieve very high accuracy and most examples are correctly classified, AUROC can give an overly optimistic view of performance, resulting in smaller observed differences across methods. For STL-10, we plan to run additional repetitions to reduce performance variability. In the case of CIFAR-10, when considering confidence intervals rather than standard deviations, PVI clearly outperforms all other methods for AUROC. Furthermore, PVI consistently achieves the best results in terms of AUPR (error), except for STL-10, which - as noted - may require more repetitions for stable evaluation. This indicates that PVI reliably assigns lower confidence to misclassified examples.
>
> **Question 3 (On Sample-Wise Margin)**
>
> A direct, 1-to-1 comparison with a unified sample-wise margin framework is challenging due to the significantly different nature of each measure. However, we believe there is much to be learned from their individual contributions, and we will emphasize how each result adds value to understanding uncertainty and confidence estimation. Prop. 4 shows that PMI is insensitive to margin when the classes are well-separated. This observation is motivated by prior work (e.g., Grønlund et al., 2020), which connects margin to generalization and prediction confidence. Thus, PMI’s inability to encode margin under such conditions is particularly relevant in the context of confidence estimation. In contrast, PSI and PVI retain meaningful notions of sample-wise margin. These more general measures can naturally extend to the case where class-conditional distributions $P(x|y=0)$ and $P(x|y=1)$ are non-overlapping—for example, by setting $\epsilon =0$ in Eq. (9), Prop. 5 holds without further assumptions.
>
> Empirically, we observe that PSI correlates most strongly with margin (Table 1), aligning with this theoretical intuition. However, it's important to distinguish between tasks: in the correlation-to-margin experiment, confidence reflects a sample’s sensitivity to decision boundaries, regardless of correctness. In contrast, tasks like misclassification detection, selective prediction, and calibration emphasize predictive reliability, where confidence is tied to accuracy. This difference in interpretation explains why PSI excels in margin-based correlation while PVI outperforms in accuracy-driven tasks. The results are not inconsistent but instead highlight the differing emphases of these evaluation criteria.
>
> **Question 4 (On Computational Cost)**
>
> While training additional models required for PI estimation can be computationally expensive, this training is performed only once. Inference, on the other hand, is efficient. For PVI, for example, the inference time is comparable to standard label prediction. Moreover, if multiple models from different runs are available, they can be directly reused for PVI estimation without retraining. We hope our findings can motivate future research toward developing more computationally efficient approaches for pointwise information estimation.
>
> **Question 5 (On PVI & Model Architecture)**
>
> While PVI assumes access to the model architecture, it does not require any modifications to the network architecture (as in MC Dropout) or training procedure (as in focal loss). Such alterations change the model’s prediction behavior, whereas PI methods operate post hoc do not alter the original predictive outputs.

---

> > ### Comment · Reviewer_Kkv7 · 2025-04-02
> >
> > I thank the authors for the thorough response. I will begin my reply by going through each enumerated point.
> >
> > ### Question 1
> >
> > This is a helpful clarification, and I would encourage the authors to further refine their manuscript to ensure this confusion does not arise with other readers.
> >
> > ### Question 2
> >
> > If the results on MNIST are truly not to be trusted / will not adequately indicate separation between methods, I wonder if they should be included in the main body of the paper. As it stands, multiple methods on the MNIST task statistically tie for all but AUPR, which weakens any claims you make about the superiority of PVI.
> >
> > For CIFAR-10, the confidence intervals (+- 1 sd) still indicate that PVI is in a series of statistical ties across metrics (unless I am missing something obvious).
> >
> > If STL-10 may require more repetitions for stable evaluation, I would encourage the authors to run those experiments and report the revised results. As it stands, the reader is still presented with a series of statistical ties across metrics.
> >
> > ### Question 3
> >
> > I appreciate the additional clarification regarding "sample-wise margin" as it is used to mean various things for each method. My initial opinion regarding this analysis still stands: as presented, sensitivity to margin does not appear to correlate with better uncertainty quantification. This discussion, while informative, does not belong in the main body of the paper and may be distracting from your overall claims.
> >
> > ### Question 4
> >
> > I agree with the authors that training, not inference, is the computationally expensive component of PI estimation. I also hope that more computationally efficient approaches can emerge to make PI estimation more tractable. However, there is not a lot of signal from the empirical results suggesting that such a direction would be worthwhile for researchers, since other, far cheaper methods of uncertainty quantification perform similarly.
> >
> > ### Question 5
> >
> > Thank you for the clarification.
> >
> > ## Overall
> >
> > My main concerns about this work still remain — namely, that the empirical results are not strongly motivated or justified by the theoretical discussion (as currently presented), and that the experiments in their current form are not convincing enough for practitioners to consider adoption of this method.
> >
> > This paper would be significantly strengthened by the use of more-realistic datasets (not toy ones like MNIST, Fashion-MNIST, CIFAR-10, etc.), especially if such experiments can demonstrate a clear and consistent advantage to PI estimation methods over existing UQ techniques. I maintain my score.

---

> > > ### Author Response · Authors · 2025-04-09
> > >
> > > We thank the reviewer and present additional results.
> > >
> > > **Q2 and Q4:**  We have re-trained VGG-16 on STL-10 and ResNet-50 on CIFAR-10 with added regularization to ensure low test errors (8.94% and 4.78%). Furthermore, as recommended, we have incorporated more realistic datasets: ResNet-101 on CIFAR-100, InceptionV3 on Stanford Dogs, and DenseNet121 on TinyImageNet. Additionally, we include 3 more commonly used metrics for failure prediction (Zhu et al., 2023): FPR at 95% TPR and EAURC, and 3 robust calibration metrics (Nixon et al., 2020): Static Calibration Error (SCE), Adaptive SCE (Ada-SCE), Class Conditional Adaptive SCE (CC-Ada-SCE), and RMS Class Conditional Adaptive SCE (CC-Ada-SCE-RMS).
> > >
> > > We report only MSP, PVI, and the top-performing benchmark method. 95% confidence intervals are provided in brackets (10 trials each). Note that STL-10 performance has overlaps due to higher sd, because of very small data size.
> > > Additional benchmarks are available at: https://shorturl.at/CWIqp.
> > >
> > > |Model|Dataset|Method|AUROC $\uparrow$|AUPR (succ) $\uparrow$|AUPR (err) $\uparrow$|FPR@95TPR $\downarrow$|AURC$\downarrow$|E-AURC$\downarrow$|
> > > |-|-|-|-|-|-|-|-|-|
> > > |VGG16|STL10|MSP|**91.43 (0.26)**|**99.07 (0.06)**|49.19 (1.13)|49.64 (1.46)|**12.72 (0.79)**|**8.59 (0.52)**|
> > > |||SM|**91.89 (0.25)**|**99.13 (0.05)**|48.70 (1.63)|48.92 (1.72)|**12.22 (0.76)**|**8.09 (0.48)**|
> > > |||**PVI**|**92.33 (0.81)**|**99.16 (0.08)**|**55.09 (3.46)**|**42.79 (3.28)**|**11.95 (0.67)**|**7.73 (0.84)**|
> > > |ResNet50|CIFAR10|MSP|93.33 (0.32)|99.62 (0.03)|42.21 (1.18)|37.70 (1.34)|4.83 (0.31)|3.67 (0.27)|
> > > |||LM|93.80 (0.23)|99.65 (0.02)|42.74 (1.23)|35.85 (1.23)|4.54 (0.26)|3.38 (0.21)|
> > > |||**PVI**|**94.82 (0.39)**|**99.70 (0.02)**|**60.45 (1.87)**|**26.86 (1.59)**|**4.08 (0.20)**|**2.92 (0.22)**|
> > > |ResNet101|CIFAR-100|MSP|84.98 (0.97)|95.02 (0.36)|59.93 (1.68)|64.09 (1.39)|66.32 (3.08)|41.08 (3.00)|
> > > |||LM|86.27 (0.28)|**95.65 (0.14)**|59.40 (0.55)|65.23 (0.94)|**61.13 (1.82)**|**35.89 (1.07)**|
> > > |||**PVI**|**87.74 (0.69)**|**95.72 (0.23)**|**69.71 (1.80)**|**49.65 (2.14)**|**60.13 (2.13)**|**34.89 (1.91)**|
> > > |InceptionV3|Stanford Dogs|MSP|81.54 (0.34)|92.88 (0.33)|57.22 (0.69)|68.58 (1.10)|87.73 (3.61)|57.74 (2.42)|
> > > |||LM|**83.51 (0.44)**|**94.05 (0.32)**|56.73 (0.69)|70.46 (1.44)|**78.31 (3.55)**|**48.33 (2.40)**|
> > > |||**PVI**|**84.32 (0.75)**|**93.80 (0.27)**|**64.86 (2.14)**|**59.89 (2.28)**|**79.88 (2.11)**|**49.87 (2.33)**|
> > > |DenseNet121|TinyImageNet|MSP|86.56 (0.47)|94.03 (0.31)|70.25 (0.65)|58.97 (1.17)|89.21 (2.80)|45.88 (2.26)|
> > > |||LM|86.50 (0.35)|**94.37 (0.18)**|66.60 (0.83)|65.44 (1.33)|**86.91 (1.68)**|43.58 (1.38)|
> > > |||**PVI**|**88.83 (0.58)**|**94.78 (0.34)**|**76.87 (0.93)**|**47.95 (0.83)**|**83.00 (2.52)**|**39.67 (2.62)**|
> > > |**Model**|**Dataset**|**Method**|**SCE**|**Ada-SCE**|**CC-Ada-SCE**|**CC-Ada-SCE-RMS**|
> > > |VGG16|STL10|MSP|0.59 (0.03)|0.52 (0.04)|**1.15 (0.06)**|**8.11 (0.18)**|
> > > |||**PVI**|**0.55 (0.00)**|**0.44 (0.01)**|**1.20 (0.00)**|**8.19 (0.00)**|
> > > |ResNet50|CIFAR10|MSP|0.35 (0.02)|0.28 (0.02)|0.60 (0.02)|5.41 (0.10)|
> > > |||**PVI**|**0.33 (0.00)**|**0.25 (0.00)**|**0.58 (0.00)**|**5.25 (0.02)**|
> > > |ResNet101|CIFAR-100|MSP|0.20 (0.00)|0.17 (0.00)|0.24 (0.00)|4.04 (0.05)|
> > > |||**PVI**|**0.19 (0.00)**|**0.16 (0.00)**|**0.23 (0.00)**|**3.98 (0.01)**|
> > > |InceptionV3|Stanford Dogs|MSP|0.20 (0.01)|0.18 (0.00)|**0.21 (0.01)**|3.75 (0.05)|
> > > |||**PVI**|**0.19 (0.00)**|**0.17 (0.00)**|**0.21 (0.00)**|**3.66 (0.01)**|
> > > |DenseNet121|TinyImageNet|MSP|**0.12 (0.00)**|**0.11 (0.00)**|**0.14 (0.00)**|**3.09 (0.04)**|
> > > |||**PVI**|**0.12 (0.00)**|**0.11 (0.00)**|**0.14 (0.00)**|**3.10 (0.01)**|
> > >
> > > **Theoretical Concerns (Q3)**: Yes, from a theory perspective, convergence rates and invariance analyses are more relevant for explaining the empirical observations, as noted in T4 and T2. As such, we will move the sample-wise margin analysis to the Appendix.
> > >
> > > Furthermore, after reviewing margin bounds in generalization theory, we can now clarify why margin sensitivity doesn’t correlate well with performance.
> > >
> > > Intuitively, while a larger margin $d$ (distance to decision boundary) for a sample $X$ may initially suggest higher confidence, the prediction's reliability also depends on how rapidly the classifier's output changes at $X$. If it changes quickly, a large $d$ does not guarantee safer/confident predictions; likewise, a smaller $d$ then does not immediately imply lower confidence if the rate of change is slow. This variability in the rate of function change limits the connection between sample-wise margin and true confidence.
> > >
> > > Furthermore, margin-bounds in generalization theory reinforce this (Theorem 1.1 of [1]), showing that the generalization gap depends on both the margin and the network's Lipschitz constant (rate of change). Thus, confidence and generalization are influenced by both factors, not margin size alone.
> > >
> > > [1] Bartlett, P. L. et al. Spectrally-normalized margin bounds for neural networks. Advances in neural information processing systems, 30 (2017).

---

### Official Review · Reviewer_3NDm · 2025-03-14

**Overall Recommendation:** 3

**Summary:**

This paper proposes to use three information-theoretic measures—Pointwise Mutual Information, Pointwise V-Information, and Pointwise Sliced Mutual Information, as post-hoc confidence estimators for neural network predictions. It theoretically analyzes their invariance properties, sensitivity to geometric features, and convergence rates, finding that PVI performs best for confidence calibration and failure prediction tasks

**Claims And Evidence:**

No problematic claims made.

**Essential References Not Discussed:**

No critical previous works seem to be omitted except the aforementioned lines of work.

**Experimental Designs Or Analyses:**

The experimental design seems convincing, except two issues:
- Results are reported only over 5 repetitions
- Comparison to prominent approaches of uncertainty estimation, such as Bayesian neural networks, is missing

**Methods And Evaluation Criteria:**

Some of the results (especially bold notation, for example in table 3) are unclear.
However, if I understand the results correctly, the empirical evaluation supports the made claims.

**Other Comments Or Suggestions:**

See above.

**Other Strengths And Weaknesses:**

The paper provides a theoretically solid and interesting contribution that enhances confidence estimation using information-theoretic measures. It provides thorough theoretical analysis, however empirical evaluation lacks comparison to the aforementioned lines of work.
I am willing to raise my score if the authors provide sufficient evidence that their method at least provides comparable results to these previous methods.

**Questions For Authors:**

It seems that PMI and PSI work well only on simple tasks. My guess is that this is due to estimation complexities. Could the authors clarify whether PMI and PSI indeed encounter those and why?

**Relation To Broader Scientific Literature:**

Connection and comparison to multiple lines of work Approaches for uncertainty estimation is missing:
-  Ensembles based methods
[e.g., Lakshminarayanan et al.]
- methods based on noise addition [e.g., Maddox et al., 2019].
- Bayesian neural networks [e.g., Tishby et al., 1989, Denker and LeCun, 1990, Ovadia et al., 2019, Kingma et al., 2015, Gal and Ghahramani, 2016, Graves, 2011, Louizos and Welling, 2016]

**Theoretical Claims:**

Definitions and assumptions regarding the information theoretic measures are clearly presented.
The theoretical analysis (invariance, sensitivity to margin, intrinsic dimensionality, and convergence) is well-reasoned and appears mathematically sound.

---

> ### Author Rebuttal · Authors · 2025-04-01
>
> We thank the reviewer for their positive remarks and valuable suggestions on the paper. Below, we address the concerns and questions raised by the reviewer.
>
> > Empirical evaluation lacks comparison to the aforementioned lines of work.
>
> The aforementioned lines of work are not post-hoc calibration methods. These methods do not compute confidence level on the original network outputs and thus will provide unfair comparison. Instead, we are expanding our comparisons to include additional post-hoc calibration methods, such as Ensemble Temperature Scaling (Zhang et al., 2020), Parameterized Temperature Scaling (Tomani et al., 2022), Class-based Temperature Scaling (Frenkel et al., 2021), Group Calibration (Yang et al., 2024), and Consistency Calibration (Tao et al., 2024). We are currently running these experiments and will include the results as soon as they become available.
>
> > It seems that PMI and PSI work well only on simple tasks. My guess is that this is due to estimation complexities. Could the authors clarify whether PMI and PSI indeed encounter those and why?
>
> The performance gap can be partially explained by the convergence properties of PMI and PSI. Specifically, the convergence rate of PMI depends inversely on the minimum of $p(x)$ and $p(x|y)$ (refer to Eq. (43) in Appendix B.3). When either of these probabilities is low, estimation becomes unreliable due to increased variance. In simpler datasets such as MNIST, class-conditional distributions are more concentrated, resulting in higher values of $p(x)$ and $p(x|y)$, which in turn leads to better convergence and performance for PMI. In contrast, complex datasets often exhibit more dispersed and overlapping class distributions, making these probabilities smaller and estimation more difficult. The same reasoning applies to PSI, which relies on similar terms across projected spaces. This explains why PMI and PSI perform better on simpler datasets and lag behind on more complex ones.
>
> References:
> - Zhang et al. (2020). Mix-N-Match: Ensemble and Compositional Methods for Uncertainty Calibration in Deep Learning.
> - Tomani et al. (2022). Parameterized Temperature Scaling For Boosting the Expressive Power in Post-hoc Uncertainty Calibration.
> - Frenkel et al. (2021). Network Calibration By Class-Based Temperature Scaling.
> - Yang et al. (2024). Beyond Probability Partitions: Calibrating Neural Networks With Semantic Aware Grouping.
> - Tao et al. (2024). Consistency Calibration: Improving Uncertainty Calibration via Consistency Among Perturbed Neighbors.

---

### Official Review · Reviewer_F6RX · 2025-03-16

**Overall Recommendation:** 3

**Summary:**

The paper explores the use of information-theoretic measures—specifically pointwise mutual information (PMI), pointwise V-information (PVI), and pointwise sliced mutual information (PSI)—to estimate prediction confidence in deep neural networks (DNNs) post-hoc, without modifying network architecture or training. It investigates their theoretical properties (invariance, margin sensitivity, convergence rates) and evaluates their effectiveness in failure prediction and confidence calibration using benchmark computer vision datasets (e.g., MNIST, CIFAR-10, STL-10). Main findings include PVI outperforming PMI, PSI, and existing baselines (e.g., maximum softmax probability) in both tasks, attributed to its balanced invariance and margin sensitivity. Key contributions are: (1) a comparative analysis of PMI, PSI, and PVI, with PVI showing superior performance; (2) theoretical insights into their invariance and geometric properties; (3) sensitivity analysis to sample-wise margin, with PSI correlating most with margin; and (4) convergence rate derivations, suggesting PSI’s advantage over PMI. Experiments also demonstrate PSI’s effectiveness in generating localized saliency maps.

**Claims And Evidence:**

Claims are well-supported by experiments (e.g., Tables 2, 11) and proofs (Appendix B), showing PVI’s superiority and PSI’s margin sensitivity. The PVI convergence claim lacks empirical backing beyond theory.

**Essential References Not Discussed:**

N/A

**Experimental Designs Or Analyses:**

Failure prediction (4.1) and calibration (4.2) experiments are sound, with robust metrics and baselines. Saliency map design (D.3) is valid.

**Methods And Evaluation Criteria:**

Using PMI, PSI, and PVI post-hoc is practical; evaluation metrics (AUROC_f, ECE) and datasets (MNIST, CIFAR-10) are appropriate for confidence estimation.

**Other Comments Or Suggestions:**

1. line 764 has a reference missing problem

**Other Strengths And Weaknesses:**

Strengths:
1. thorough study on point wise  Confidence Estimation

weakness:
1. did not include the OoD experiment as usual uncertainty estimation papers.

2. Despite the thorough study on PI measures, the paper lack the recommendation or suggestions for readers to apply these measures. I see some improvement upon some of the tasks, but looks like not consistent. I am familiar with the calibration task, but the results improvement seems trivial and the results of MSP not align well with my experience

3. A recent paper[1] also discuss a way to compute sample wise confidence, it would be better to compare with.

4. comparison many other post hoc calibration method is missed table 3.


[1] Consistency Calibration: Improving Uncertainty Calibration via Consistency among Perturbed Neighbors

**Questions For Authors:**

1. Since the paper normalize these PI measures  using a softmax function, does the softmax lose the ability of PI to express uncertainty?

2. For table 1, I wonder what is the correlation between vanilla neural network output(softmax value) and margin? How does it compared to  these PI measure?

3. Can you give more details about "estimate the PVI between X and Y in the experiment "?

4. I am a calibration person, for table 3, a normal ECE for a temperature scaled MSP (resent for cifar10 column) is around 2, but your results is 10, which is confusing to me. Maybe using a more standard, well accepted baseline[1] can give a fair comparison.

[1] Calibrating Deep Neural Networks using Focal Loss

**Relation To Broader Scientific Literature:**

It contribute the understanding of point wise estimation of confidence calibration based on information theory.

**Theoretical Claims:**

Checked Propositions 1-5 and Theorem 1 (Appendix B); they’re correct, though Theorem 1’s spherical assumption simplifies real-world applicability.

---

> ### Author Rebuttal · Authors · 2025-04-01
>
> We thank the reviewer for their positive remarks and valuable suggestions on the paper. Below, we address the concerns and questions raised by the reviewer.
>
> **Weakness 1 (On OoD Experiment)**
>
> The primary motivation of our work stems from the observation that existing calibration methods can adversely affect failure prediction performance. This motivates our focus on both calibration and failure prediction as key evaluation tasks. We may explore the extension to OoD in future work.
>
> **Weakness 2 (On Takeaway of Results)**
>
> Our results demonstrate that PVI consistently provides strong performance across both failure prediction (Table 2 and Table 11) and calibration (Table 3 and Table 15) tasks.
>
> **Weakness 3 and 4 (On Comparison to Other Methods)**
>
> We will be expanding our comparisons to include additional post-hoc calibration methods, such as Ensemble Temperature Scaling (ETS), Parameterized Temperature Scaling (PTS), Class-based Temperature Scaling (CTS), Group Calibration (GC), and Consistency Calibration (CC) as suggested by the reviewer. We are currently running these experiments and will include the results as soon as they become available.
>
> **Question 1 (On Softmax Scaling)**
>
> The softmax function is applied to re-normalize the PI measures for interpretability as probabilities. Unlike standard logits, PI measures are derived from a principled probabilistic framework and thus inherently capture pointwise relationships in the data. In contrast, logits are unnormalized scores lacking direct probabilistic meaning. Importantly, applying softmax to PI values does not remove the uncertainty signal they encode - it preserves the relative structure of uncertainty while transforming the values into confidence-like scores. High PI values (reflecting high confidence or low uncertainty) remain high after softmax normalization, and low values (reflecting low confidence or high uncertainty) remain low.
>
> **Question 2 (On Softmax Correlation to Margin)**
>
> Below are the correlation between softmax value and margin:
> | Method   | MLP, MNIST | CNN, F-MNIST | VGG16, STL-10 | ResNet50, CIFAR-10 |
> |---|---|----|---|----|
> | softmax  | 0.954±0.011 | 0.935±0.026 | 0.913±0.004 | 0.882±0.009 |
>
> Softmax output is most sensitive to the gap between the top two logits (assuming all other logits are much smaller), which is what the margin approximation is also capturing. Naturally, it will have the highest correlation with margin. That being said, the goal of this experiment is to empirically verify the properties of PI measures.
>
> Also note that a measure can have high correlation with margin because it closely tracks the model's logit gap - indicating how "close" a sample is to the decision boundary - but this doesn’t guarantee good performance in failure prediction or calibration. Margin-based measures like softmax may appear confident even on out-of-distribution or adversarial inputs, leading to overconfidence. In contrast, other measures might be less tightly coupled to margin but are more sensitive to ambiguity, distributional shifts, and model failures, making them more reliable for detecting errors and producing well-calibrated confidence.
>
>
> **Question 3 (On PVI Estimation)**
>
> We provided more details on PVI estimation in Appendix A.3.3. Following the approach of Ethayarajh et al. (2022), we estimate the PVI between $X$ and $Y$ by training an auxiliary model with the same architecture. This measures how well $Y$ can be predicted from $X$ using $\mathcal{V}$. Thus, in a way it’s capturing the confidence of the model $\mathcal{V}$.
>
> **Question 4 (On ECE Results)**
>
> In Mukhoti et al. (2020), data augmentation was used to improve the performance of ResNet-50 on CIFAR-10, resulting in a test error of 4.95. In contrast, our test error is 13.72 (Table 9), which accounts for the observed difference in ECE. Nevertheless, we are currently running experiments to reduce the test errors and will share the updated results once available.
>
> References:
> - Ethayarajh, K., et al. (2022). Understanding Dataset Difficulty with V-Usable Information.
> - Mukhoti, J., et al. (2020). Calibrating Deep Neural Networks using Focal Loss.

---

### Decision · Program_Chairs · 2025-05-01

**Decision:**

Accept (poster)

**Comment:**

This work considers three information-theoretic measures (Pointwise Mutual Information, Pointwise V-Information, and Pointwise Sliced Mutual Information) to estimate the confidence of predictions from trained neural networks after training. Through a thorough theoretical analysis, it is found that Pointwise V-Information provides the most reliable measure for confidence calibration and failure/selective prediction.

This is an interesting and thorough study, reflected in generally positive scores from the reviewers. Initially, reviewers had concerns over the simplicity of the models and datasets considered in the experiments, although this has been thoroughly addressed in the author rebuttal to Kkv7. This is a _major_ improvement to the manuscript. I strongly suggest that these results are included in the final revision, along with the results for the additional calibration approaches. There is a wide array of theoretical results on display, and all of the major concerns about the theoretical claims raised in the reviews have been addressed during the rebuttal period as well.

In my estimation, the authors have responded well to each concern and have produced a solid contribution to ICML. I recommend acceptance.